# EXPLAINABLE GRAPH REPRESENTATION LEARNING VIA GRAPH PATTERN ANALYSIS

## ABSTRACT

Explainable artificial intelligence (XAI) is an important area in the AI community, and interpretability is crucial for building robust and trustworthy AI models. While previous work has explored model-level and instance-level explainable graph learning, there has been limited investigation into explainable graph representation learning. In this paper, we focus on representation-level explainable graph learning and answer a fundamental question: *What specific information about a graph is captured in graph representations?* Our approach is inspired by graph kernels, which evaluate graph similarities by counting substructures within specific graph patterns. First, we present an unsupervised ensemble graph kernel method for representation or similarity explanation, which however has limitations such as ignoring node features and being computationally expensive. To address these limitations, we introduce a deep learning framework for learning and explaining graph representations through graph pattern analysis. We start by sampling graph substructures of various patterns. Then, we learn the representations of these patterns and combine them using a weighted sum, where the weights indicate the importance of each graph pattern's contribution. Note that our method can be both unsupervised and supervised and is a one-shot explanation, not specified to single samples or predictions. We also theoretically analyze the robustness and generalization ability of our models. Importantly, the generalization analysis shows that incorporating multiple graph patterns lowers the generalization error bound. In our experiments, we show how to learn and explain graph representations for real-world data using pattern analysis. Additionally, we compare our method against multiple baselines in both supervised and unsupervised learning tasks to demonstrate its superiority in terms of accuracy.

## 1 INTRODUCTION

The field of explainable artificial intelligence (XAI) (Došilović et al., 2018; Adadi & Berrada, 2018; Angelov et al., 2021; Hassija et al., 2024) is gaining significant attention in both AI and science communities. Interpretability is crucial for creating robust and trustworthy AI models, especially in critical domains like transportation, healthcare, law, and finance. Graph learning is an important area of AI that particularly focuses on graph-structured data widely exist in social science, biology, chemistry, etc. Explainable graph learning (XGL) (Kosan et al., 2023) can be generally classified into two categories: model-level methods and instance-level methods.

Model-level methods of XGL provide transparency by analyzing the model behavior. Examples include XGNN (Yuan et al., 2020), GLG-Explainer (Azzolin et al., 2022), and GCFExplainer (Huang et al., 2023). Instance-level methods of XGL offer explanations tailored to specific predictions, focusing on why particular instances are classified in a certain manner. For instance, GNNExplainer (Ying et al., 2019) identifies a compact subgraph structure crucial for a GNN's prediction. PGExplainer (Luo et al., 2020) trains a graph generator to incorporate global information and parameterize the explanation generation process. AutoGR (Wang et al., 2021) introduces an explainable AutoML approach for graph representation learning. MotifExplainer (Yu & Gao, 2022) identifies critical motifs (small subgraphs) in a graph. UNR-Explainer (Kang et al., 2024) identifies the top-k most important nodes in a graph to determine the most significant subgraph as the counterfactual explanation. More about XGL can be found in the Appendix C.1.

However, these works mainly focus on enhancing the transparency of GNN models or identifying the most important substructures that contribute to predictions. The exploration of representation-level explainable graph learning (XGL) is limited. We propose explainable graph representation learning and ask a fundamental question: **What specific information about a graph is captured in graph representations?** Formally, if we represent a graph $G$ as a $d$-dimensional vector $g$, our goal is to understand what specific information about the graph $G$ is embedded in the representation $g$. This problem is important and has practical applications. Some graph patterns are highly practical and crucial in various real-world tasks, and we want this information to be captured in representations. For instance, in molecular chemistry, bonds between atoms or functional groups often form cycles (rings), which indicate a molecule's properties and can be used to generate molecular fingerprints (Morgan, 1965; Alon et al., 2008; Rahman et al., 2009; O'Boyle & Sayle, 2016). Similarly, cliques characterize protein complexes in Protein-Protein Interaction networks and help identify community structures in social networks (Girvan & Newman, 2002; Jiang et al., 2010; Fox et al., 2020).

Although some previous works such as (Kosan et al., 2023) aimed to find the most critical subgraph $S$ by solving optimization problems based on perturbation-based reasoning, either factual or counterfactual, this kind of approach assumes that the most important subgraph $S$ mainly contributes to the representation $g$, neglecting other aspects of the graph, which doesn't align well with our goal of thoroughly understanding graph representations. Analyzing all subgraphs of a graph $G$ is impractical due to their vast number. To address the challenge, we propose to group the subgraphs into different graph patterns, like paths, trees, cycles, cliques, etc, and then analyze the contribution of each graph pattern to the graph representation $g$.

Our idea of pattern analysis is inspired by graph kernels, which compare substructures of specific graph patterns to evaluate the similarity between two graphs (Kriege et al., 2020). For example, random walk kernels (Borgwardt et al., 2005; Gärtner et al., 2003) use path patterns, sub-tree kernels (Da San Martino et al., 2012; Smola & Vishwanathan, 2002) examine tree patterns, and graphlet kernels (Pržulj, 2007; Shervashidze et al., 2009) focus on graphlet patterns. The graph kernel involves learning a pattern counting representation vector $h$, which counts the occurrences of substructures of a specific pattern within the graph $G$. While the pattern counting vector $h$ is an explainable representation, it has some limitations, such as the high dimensionality and ignorance of node features.

There also exist some representation methods based on subgraphs and substructures, such as Subgraph Neural Networks (SubGNN) (Kriege & Mutzel, 2012), Substructure Assembling Network (SAN) (Zhao et al., 2018), Substructure Aware Graph Neural Networks (SAGNN) (Zeng et al., 2023a), and Mutual Information (MI) Induced Substructure-aware GRL (Wang et al., 2020). However, these methods mainly focus on increasing expressiveness and do not provide explainability for representation learning. We will discuss the details in the Appendix C.2.

In this work, we propose a novel framework to learn and explain graph representations via graph pattern analysis. We start by sampling graph substructures of various patterns. Then, we learn the representations of these patterns and combine them adaptively, where the weights indicate the importance of each graph pattern's contribution. We also provide theoretical analyses of our methods, including robustness and generalization. Additionally, we compare our method against multiple baselines in both supervised and unsupervised learning tasks to demonstrate its effectiveness and superiority. Our contributions are summarized as follows:

- Unlike previous model-level and instance-level XGL, we introduce a new problem — representation-level explainable graph learning. This problem focuses on understanding what specific information about a graph is embedded within its representations in unsupervised learning.

- We propose two strategies to learn and explain graph representations, including a graph ensemble kernel method and a pattern analysis GNN method. The latter involves using GNNs to learn the representations of each pattern and evaluate its contribution to the ensemble graph representation.

- We provide robust analyses and generalization analysis for our methods theoretically. Particularly, our generalization analysis shows adding graph patterns lowers the generalization error bound.

## 2    NOTATIONS

In this work, we use $x$, $\boldsymbol{x}$, $\boldsymbol{X}$, and $\mathcal{X}$ (or $X$) to denote scalar, vector, matrix, and set, respectively. We denote $[n] = \{1, 2, ..., n\}$. Let $G = (V, E)$ be a graph with $n$ nodes and $d$-dimensional node features $\{\boldsymbol{x}_v \in \mathbb{R}^d \mid v \in V\}$. We denote $\boldsymbol{A} \in \{0, 1\}^{n \times n}$ the adjacency matrix and $\boldsymbol{X} = [\boldsymbol{x}_1, \ldots, \boldsymbol{x}_n]^\top \in \mathbb{R}^{n \times d}$ the node features matrix. Let $\mathcal{G} = \{G_1, \ldots, G_N\}$ be a dataset of $N$ graphs belonging $C$ classes, where $G_i = (V_i, E_i)$. For $G_i$, we denote its number of nodes as $n_i$, the one-hot graph label as $\boldsymbol{y}_i \in \{0, 1\}^C$, the graph-level representation as a vector $\boldsymbol{g}_i \in \mathbb{R}^d$, the adjacency matrix as $\boldsymbol{A}_i$, and the node feature matrix as $\boldsymbol{X}_i$. Let $S = (V_S, E_S)$ be a subgraph of graph $G = (V, E)$ such that $V_S \subseteq V$ and $E_S \subseteq E$. The the adjacency matrix of $S$ is denoted as $\boldsymbol{A}_S \in \{0, 1\}^{|V_S| \times |V_S|}$ and the node feature matrix of $S$ is sampled from the rows of $\boldsymbol{X}$, denoted as $\boldsymbol{X}_S \in \mathbb{R}^{|V_S| \times d}$.

The graph pattern is defined as a set of all graphs that share certain properties, denoted as $\mathcal{P} = \{P_1, P_2, \ldots, P_i, \ldots\}$, where $P_i$ is the $i$-th example of this pattern. In this work, the graph patterns are basic graph families such as paths, trees, cycles, cliques, etc. Detailed mathematical definitions for some of these patterns are provided in Appendix B. For example:

- $\mathcal{P}_{\text{path}} = \{\text{ph}_1, \text{ph}_2, \ldots, \text{ph}_i, \ldots\}$ is a path pattern with $\text{ph}_i$ as a path of length $i$.
- $\mathcal{P}_T = \{T_1, T_2, \ldots, T_i, \ldots\}$ is a tree pattern where $T_i$ is the $i$-th tree.
- $\mathcal{P}_{\text{gl}} = \{\text{gl}_1, \text{gl}_2, \ldots, \text{gl}_i, \ldots\}$ is a graphlet pattern where $\text{gl}_i$ is the $i$-th graphlet.

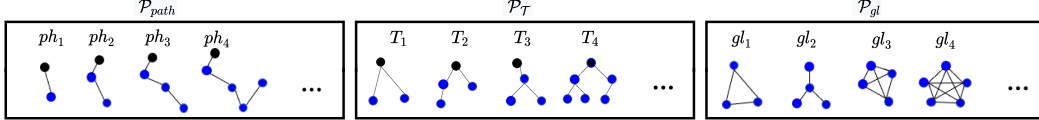

Figure 1: Examples of graph patterns: $\mathcal{P}_{\text{path}}$, $\mathcal{P}_T$ and $\mathcal{P}_{\text{gl}}$

Figure 1 illustrates some intuitive examples of graph patterns. Notably, there are overlaps among different patterns; for instance, the graph $T_3 \in \mathcal{P}_T$ and $\text{gl}_2 \in \mathcal{P}_{\text{gl}}$ are identical, being both a tree and a graphlet. Overlaps are inevitable due to the predefined nature of these basic graph families in graph theory. We denote a set of $M$ different patterns as $\{\mathcal{P}_1, \mathcal{P}_2, \ldots, \mathcal{P}_m, \ldots, \mathcal{P}_M\}$. Given the pattern $\mathcal{P}_m$ and the graph $G_i$, the pattern sampling set is denoted as $\mathcal{S}_i^{(m)}$ and the pattern representation is denoted as $\boldsymbol{z}_i^{(m)} \in \mathbb{R}^d$.

## 3    LEARNING EXPLAINABLE GRAPH REPRESENTATIONS VIA ENSEMBLE GRAPH KERNEL

In this section, we learn and explain the pattern counting graph representation via graph kernels.

### 3.1    PATTERN COUNTING KERNEL

A graph kernel $K : \mathbb{G} \times \mathbb{G} \to \mathbb{R}$ aims to evaluate the similarity between two graphs. Let $G_i$ and $G_j$ be two graphs in the graph dataset $\mathcal{G}$ and let $\mathcal{H}$ be a high-dimensional vector space. The key to a graph kernel is defining a mapping from the graph space to the high-dimensional vector space as $\phi : \mathbb{G} \to \mathcal{H}$, where $\boldsymbol{h}_i = \phi(G_i)$ and $\boldsymbol{h}_j = \phi(G_j)$. Then, the graph kernel can be defined as the inner product of $\boldsymbol{h}_i$ and $\boldsymbol{h}_j$, i.e., $K(G_i, G_j) := \boldsymbol{h}_i^\top \boldsymbol{h}_j$. The most widely used mapping $\phi$ is the one counting the occurrences of each example in the pattern $\mathcal{P}$ within graph $G$. The corresponding pattern counting vector is defined as follows.

**Definition 3.1** (Pattern Counting Vector). Given a graph $G$ and a pattern $\mathcal{P} = \{P_1, P_2, \ldots, P_i, \ldots\}$, a pattern counting mapping $\phi : \mathbb{G} \to \mathcal{H}$ is defined as

$$\boldsymbol{h} = \phi(G; \mathcal{P}), \quad \text{with} \ \boldsymbol{h} = [h^{(1)}, h^{(2)}, \ldots, h^{(i)}, \ldots], \tag{1}$$

where $h^{(i)}$ is the number of occurrences of pattern example $P_i$ as a substructure within graph $G$. We call $\boldsymbol{h}$ a pattern counting vector of $G$ related to pattern $\mathcal{P}$.

Then the pattern counting kernel $K_{\mathcal{P}} : \mathbb{G} \times \mathbb{G} \to \mathbb{R}$ based on pattern $\mathcal{P}$ can be defined.

**Definition 3.2** (Pattern Counting Kernel). Given the a pattern counting mapping $\phi(G; \mathcal{P})$, a pattern counting kernel is defined as

$$K_{\mathcal{P}}(G_i, G_j) := \langle \phi(G_i; \mathcal{P}), \phi(G_j; \mathcal{P}) \rangle = \boldsymbol{h}_i^\top \boldsymbol{h}_j \tag{2}$$

The pattern counting kernel $K_{\mathcal{P}}$ is uniquely determined by the pattern $\mathcal{P}$. For example, if $\mathcal{P}$ is selected as the path pattern $\mathcal{P}_{\text{path}}$, we obtain a random walk kernel (Borgwardt et al., 2005; Gärtner et al., 2003). If $\mathcal{P}$ is the tree pattern $\mathcal{P}_T$, we get a sub-tree kernel (Da San Martino et al., 2012; Smola & Vishwanathan, 2002). Similarly, if $\mathcal{P}$ is the graphlet pattern $\mathcal{P}_{\text{gl}}$, we derive a graphlet kernel (Pržulj, 2007).

## 3.2 PATTERN ANALYSIS USING GRAPH KERNELS

Let $\{\mathcal{P}_1, \mathcal{P}_2, \ldots, \mathcal{P}_M\}$ be a set of $M$ different graph patterns. For instance, $\mathcal{P}_1$ represents the path pattern and $\mathcal{P}_2$ represents the tree pattern. Then, we can define a set of $M$ different graph kernels as $\{K_{\mathcal{P}_1}, K_{\mathcal{P}_2}, \ldots, K_{\mathcal{P}_M}\}$. Since the pattern counting kernel $K_{\mathcal{P}_m}$ is uniquely determined by the pattern $\mathcal{P}_m$, we can analyze the importance of pattern $\mathcal{P}_m$ by evaluating the importance of its pattern counting kernel $K_{\mathcal{P}_m}$. To achieve this, we define a learnable ensemble kernel as follows:

**Definition 3.3** (Learnable Ensemble Kernel). Let $\boldsymbol{\lambda} = [\lambda_1, \lambda_2, ..., \lambda_m, ..., \lambda_M]^\top$ be a positive weight parameter vector. The ensemble kernel matrix $\boldsymbol{K}(\boldsymbol{\lambda}) \in \mathbb{R}^{|\mathcal{G}| \times |\mathcal{G}|}$ is defined as the weighted sum of $M$ different kernels $\{K_{\mathcal{P}_1}, K_{\mathcal{P}_2}, \ldots, K_{\mathcal{P}_M}\}$. Given two graphs $G_i$ and $G_j$ in $\mathcal{G}$, the element at the $i$-th row and $j$-th column of $\boldsymbol{K}(\boldsymbol{\lambda})$ is given by

$$K_{ij}(\boldsymbol{\lambda}) := \sum_{m=1}^M \lambda_m \, K_{\mathcal{P}_m}(G_i, G_j), \;\; \text{s.t} \;\; \sum_{m=1}^M \lambda_m = 1, \;\; \text{and} \;\; \lambda_m \geq 0, \;\; \forall m \in [M]. \tag{3}$$

Here, the weight parameter $\lambda_m$ indicates the importance of the kernel $K_{\mathcal{P}_m}$ as well as the corresponding graph pattern $\mathcal{P}_m$ within the dataset $\mathcal{G}$. Instead of the constrained optimization (3), we may consider replacing $\lambda_m$ with $\exp(w_m)/\sum_{m=1}^M \exp(w_m)$ such that the constraints are satisfied inherently, which leads to an unconstrained optimization in terms of $\boldsymbol{w} = [w_1, \ldots, w_M]^\top$. In the following context, for convenience, we just focus on (3), though all results are applicable to the unconstrained optimization. To obtain the weight parameter $\boldsymbol{\lambda}$, we provide the supervised and unsupervised loss functions as follows.

**Supervised Contrastive Loss** Following (Oord et al., 2018), given a kernel matrix $\boldsymbol{K}(\boldsymbol{\lambda}) \in \mathbb{R}^{N \times N}$, we define the supervised InfoNEC as follows

$$\mathcal{L}_{\text{SCL}}(\boldsymbol{\lambda}) = - \sum_{i \neq j} \mathbb{I}_{[\boldsymbol{y}_i = \boldsymbol{y}_j]} \left( \log K_{ij}(\boldsymbol{\lambda}) - \log \left[ \sum_k \mathbb{I}_{[\boldsymbol{y}_i = \boldsymbol{y}_k, i \neq k]} K_{ik}(\boldsymbol{\lambda}) + \mu \sum_k \mathbb{I}_{[\boldsymbol{y}_i \neq \boldsymbol{y}_k]} K_{ik}(\boldsymbol{\lambda}) \right] \right), \tag{4}$$

where $\mathbb{I}_{[\cdot]}$ is an indicator function and $\mu > 0$ is a hyperparameter.

**Unsupervised KL Divergence** Inspired by (Xie et al., 2016), given a kernel matrix $\boldsymbol{K} \in \mathbb{R}^{N \times N}$, we define the unsupervised KL divergence loss as follows

$$\mathcal{L}_{\text{KL}}(\boldsymbol{\lambda}) = \mathbb{KL}(\boldsymbol{K}(\boldsymbol{\lambda}), \boldsymbol{K}'(\boldsymbol{\lambda})), \;\; \text{with} \;\; \boldsymbol{K}'_{ij}(\boldsymbol{\lambda}) = \frac{K_{ij}^2(\boldsymbol{\lambda})/r_j}{\sum_{j'} K_{ij'}^2(\boldsymbol{\lambda})/r_{j'}} \;\; \text{and} \;\; r_j = \sum_j K_{ij}(\boldsymbol{\lambda}), \tag{5}$$

where $r_j$ are soft cluster frequencies. By minimizing the KL divergence, the model adjusts the parameters $\boldsymbol{\lambda}$ to more accurately represent the natural clustering property of the dataset.

We use the $\mathcal{L}_{\text{SCL}}$ or $\mathcal{L}_{\text{KL}}$ as our loss function, i.e., $\mathcal{L}_{\text{ker}}(\boldsymbol{\lambda}) = \mathcal{L}_{\text{SCL}}(\boldsymbol{K}(\boldsymbol{\lambda}))$ or $\mathcal{L}_{\text{KL}}(\boldsymbol{K}(\boldsymbol{\lambda}))$, when the graphs are labeled or unlabeled. Then the weight parameter $\boldsymbol{\lambda}$ can be obtain by solving

$$\boldsymbol{\lambda}^* = \operatorname{argmin}_{\mathbf{1}_M^\top \boldsymbol{\lambda} = 1, \, \boldsymbol{\lambda} \geq 0} \; \mathcal{L}_{\text{ker}}(\boldsymbol{\lambda}), \tag{6}$$

where $\boldsymbol{\lambda}^* = [\lambda_1^*, ..., \lambda_m^*, ...\lambda_M^*]^\top$ and $\lambda_m^*$ indicates the importance of kernel $K_{\mathcal{P}_m}$ as well as pattern $\mathcal{P}_m$. In Figure 2, we can see that the ensemble Kernel performs better than each single kernel and

the pattern analysis identifies the importance of each kernel as well as the related graph pattern. We call this method pattern-based XGL with ensemble graph kernel, abbreviated as **PXGL-EGK**. This method not only yields explainable similarity learning but also provides an approach to selecting graph kernels and their hyperparameters automatically if we consider different kernel types with different hyperparameters.

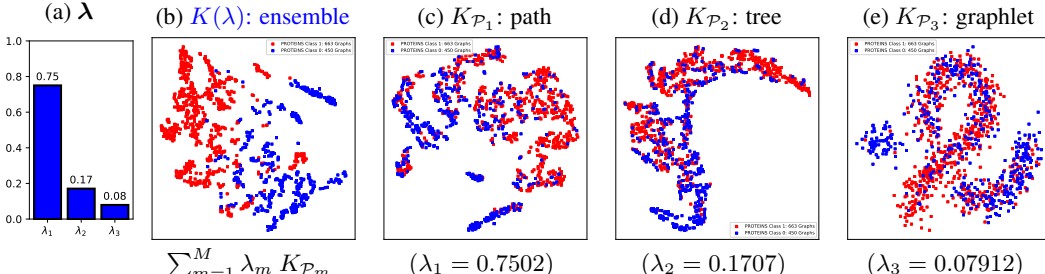

$$\sum_{m=1}^{M} \lambda_m \, K_{\mathcal{P}_m} \qquad (\lambda_1 = 0.7502) \qquad (\lambda_2 = 0.1707) \qquad (\lambda_3 = 0.07912)$$

Figure 2: t-SNE visualizations of different kernel embeddings for the dataset PROTEINS.

### 3.3 LIMITATIONS OF PATTERN COUNTING VECTOR

The pattern counting vector $\boldsymbol{h}$ from Definition 3.1 is easy to understand and its importance can be evaluated using the weight parameter $\boldsymbol{\lambda}^*$ from (6). However, it cannot directly explain the representation of graph $G$ due to the following limitations, which are also the limitations of the proposed PXGL-EGK.

- **Ignoring Node Features:** $\boldsymbol{h}$ captures the topology of $G$ but ignores node features $\boldsymbol{X}$. As shown by previous GNN works, node features are crucial for learning graph representations.

- **High Dimensionality:** The pattern set $\mathcal{P} = \{P_1, P_2, \ldots, P_i, \ldots\}$ can be vast, making $\boldsymbol{h}$ high-dimensional and impractical for many tasks.

- **High Computational Complexity:** Counting patterns $P_i$ in $G$ is time-consuming due to the large number of patterns in $\mathcal{P}$. The function $\phi(G; \mathcal{P})$ needs to be run for each new graph. In addition, in PXGL-EGK, the computation of the $M$ kernel matrices of size $|\mathcal{G}| \times |\mathcal{G}|$ is very expensive especially when $|\mathcal{G}|$ is large.

- **Lacking Implicit Information and Strong Expressiveness:** $\boldsymbol{h}$ is fixed and not learnable. GNN (Kipf & Welling, 2016) shows that message passing can learn implicit information and provide better representations, which should be considered if possible.

## 4 LEARNING EXPLAINABLE GRAPH REPRESENTATIONS VIA GNNS

In this section, we address the limitations pointed out in Section 3.3 by proposing a GNN framework to learn and explain graph representations via pattern analysis. We first present the definitions of the pattern sample set, pattern representation, and ensemble representation and then show the objective functions of unsupervised and supervised learning.

**Definition 4.1** (Pattern Sample Set). A $\mathcal{P}$-pattern sample set $\mathcal{S}$ of a given graph $G$ is defined as

$$\mathcal{S} := \{S_1, S_2, \ldots, S_q, \ldots, S_Q\}, \tag{7}$$

where $S_q$, $q \in [Q]$, is a subgraph of pattern $\mathcal{P}$ (see the examples in Figure 1) randomly sampled from $G$ using some sampling function $\Phi$[1].

**Definition 4.2** (Pattern Representation). Let $\mathcal{S}$ be a $\mathcal{P}$-pattern sample set of a graph $G$. For each subgraph $S \in \mathcal{S}$, denote its node set, adjacency matrix, and node feature matrix as $V_S$, $\boldsymbol{A}_S$, and $\boldsymbol{X}_S$ respectively. Let $F : \{0,1\}^{|V_S| \times |V_S|} \times \mathbb{R}^{|V_S| \times d} \to \mathbb{R}^{d'}$ be a pattern representation learning function parameterized by $\mathcal{W}$, then the $\mathcal{P}$-pattern representation $\boldsymbol{z} \in \mathbb{R}^{d'}$ of $G$ is defined as

$$\boldsymbol{z} = \frac{1}{|\mathcal{S}|} \sum_{S \in \mathcal{S}} F(\boldsymbol{A}_S, \boldsymbol{X}_S; \mathcal{W}). \tag{8}$$

---

[1] The specific $\Phi$ follows `https://ysig.github.io/GraKeL/0.1a8/`

The pattern representation learning function $F$ could be any graph neural network such as GCN (Kipf & Welling, 2016), GIN (Xu et al., 2018), and graph transformer (Rampášek et al., 2022). In this paper, we use GCN only for convenience. Because of the presence of node features, the chance that overlaps occur between patterns is tiny. Nevertheless, we can use the WL-test (Huang & Villar, 2021) in each sampling phase to ensure that new samples are unique from existing ones, which is efficient as the subgraphs are small.

Finally, the ensemble representation $g$ is a weighted sum of the $M$ pattern representations as follows.

**Definition 4.3** (Ensemble Representation). Given a graph $G$ and consider a set of $M$ different patterns $\{\mathcal{P}_1, \mathcal{P}_2, \ldots, \mathcal{P}_m, \ldots, \mathcal{P}_M\}$, we denote $z^{(m)}$ the $\mathcal{P}_m$-pattern representation obtained from the $\mathcal{P}_m$-pattern set $\mathcal{S}^{(m)}$ using a pattern representation learning function $F_m$. Let $\lambda = [\lambda_1, \lambda_2, \ldots, \lambda_m, \ldots, \lambda_M]^\top$ be a parameter vector, where $\mathbf{1}_M^\top \lambda = 1$ and $\lambda_m \geq 0 \ \forall \ m \in [M]$. Then the ensemble representation $g \in \mathbb{R}^{d'}$ of $G$ is defined as

$$g = \sum_{m=1}^{M} \lambda_m z^{(m)}, \ \text{ with } \ z^{(m)} = \frac{1}{|\mathcal{S}^{(m)}|} \sum_{S \in \mathcal{S}^{(m)}} F_m(A_S, X_S; \mathcal{W}^{(m)}), \ \forall \, m \in [M]. \quad (9)$$

Note that instead of explicitly considering the constraints for $\lambda$, we can use the same softmax trick in computing the ensemble kernel (3) to simplify the problem.

Let $\mathbb{W} := \{\mathcal{W}^{(1)}, \mathcal{W}^{(2)}, \ldots, \mathcal{W}^{(m)}, \ldots, \mathcal{W}^{(M)}\}$ be the parameters of the $M$ GNNs. In unsupervised representation learning, we define the similarity between two graphs' ensemble representations as $K_{ij}(\lambda, \mathbb{W}) = \exp\left(-\gamma\|g_i - g_j\|^2\right)$, where $\gamma > 0$ is a hyperparameter. Then similar to (5), we minimize the following objective function to optimize $\mathbb{W}$

$$\mathcal{L}_{\text{KL}}(\lambda, \mathbb{W}) = \mathbb{KL}(K(\lambda, \mathbb{W}), K'(\lambda, \mathbb{W})) \quad (10)$$

where the computation of $K'$ is the same as that in (5).

In supervised learning, given a graph $G \in \mathcal{G}$ with ensemble representation $g$, denote $y \in \{0,1\}^C$ the ground truth label. Let $\hat{y} \in [0,1]^C$ be the predicted label given by a softmax classifier $f_c : \mathbb{R}^d \to \mathbb{R}^C$ parameterized by $W_C$, i.e., $\hat{y} = f_c(g)$. Let $\ell_{\text{CE}}$ be the multi-class cross-entropy loss, i.e., $\ell_{\text{CE}}(y, \hat{y}) = \sum_{c=1}^{C} y_c \log \hat{y}_c$. Then we minimize the following objective to optimize the parameters $\bar{\mathbb{W}} = \{\mathbb{W}, W_C\}$:

$$\mathcal{L}_{\text{CE}}(\lambda, \bar{\mathbb{W}}) = \frac{1}{N} \sum_{i=1}^{N} \ell_{\text{CE}}\left(y_i, f_c(g_i)\right) \quad (11)$$

Let $\lambda^* = [\lambda_1^*, \ldots, \lambda_m^*, \ldots, \lambda_M^*]^\top$ be the optimal $\lambda$ obtained from minimizing (10) or (11). $\lambda_m^*$ indicates the contribution of the pattern representation $z^{(m)}$ to the ensemble graph representation $g$. In Figure 4, we visualize the $g$ and each $z^{(m)}$ and show that the ensemble representation $g$ performs the best and the $\lambda_m^*$ explains the contribution of each pattern representation $z^{(m)}$ to learning $g$. For convenience, we call this method pattern-based XGL with GNNs, abbreviated as **PXGL-GNN**.

## 5 THEORETICAL ANALYSIS

In this section, we analyze the robustness property, generalization ability, and computational complexity of our methods theoretically, which not only is important to understand the proposed methods but also provides theoretical support for the effectiveness of the proposed methods. We defer the detailed proof to Appendices D and E.

### 5.1 ROBUSTNESS ANALYSIS

Following (O'Bray et al., 2021), a learning method should be robust to small perturbations. Let $\Delta_A$ and $\Delta_X$ be perturbations on the adjacency matrix and node attributes of a graph $G$ whose representation is denoted as $g$. Then the perturbed graph is $\tilde{G} = (A + \Delta_A, X + \Delta_X)$, of which the representation is denoted as $\tilde{g}$. We seek the upper bound of $\|\tilde{g} - g\|$ and want to know how it is related to $\Delta_A$ and $\Delta_X$ as well as the representation learning function $F$. Without loss of

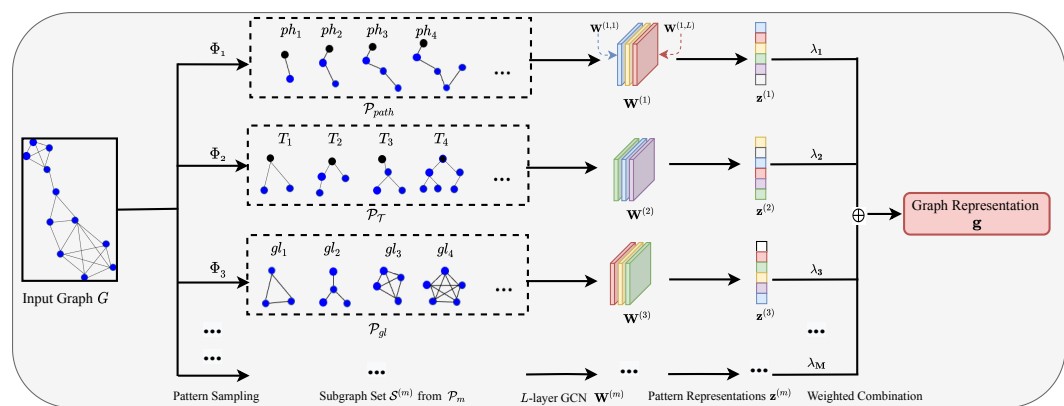

Figure 3: Proposed GNN framework for computing the ensemble graph representation

generality, we assume that $G$ has $n$ nodes, $F$ is an $L$-layer GCN (Kipf & Welling, 2016), and all the activation functions are $\sigma(\cdot)$. For each pattern $\mathcal{P}_m$, the parameter set of $F(\boldsymbol{A}, \boldsymbol{X}; \mathcal{W}^{(m)})$ are $\mathcal{W}^{(m)} = \{\boldsymbol{W}^{(m,1)}, \ldots, \boldsymbol{W}^{(m,L)}\}$, where $\boldsymbol{W}^{(m,l)}$ denotes the parameter matrix in the $l$-th layer. We further assume that for each pattern $\mathcal{P}_m$, the output vector representation is obtained by the average pooling. Then we have the following theorem.

**Theorem 5.1.** *Let* $\tilde{\boldsymbol{A}} = \boldsymbol{A} + \Delta_A$ *and* $\tilde{\boldsymbol{X}} = \boldsymbol{X} + \Delta_X$. *Suppose* $\|\boldsymbol{A}\|_2 \leq \beta_A$, $\|\boldsymbol{X}\|_F \leq \beta_X$, $\|\boldsymbol{W}^{(m,l)}\|_2 \leq \beta_W$ *for all* $m \in [M]$ *and* $l \in [L]$, *and* $\sigma(\cdot)$ *is* $\rho$-*Lipschitz continuous. Let* $\alpha$ *be the minimum node degree of* $G$, *and* $\Delta_D := \boldsymbol{I} - diag(\boldsymbol{1}^\top(\boldsymbol{I} + \boldsymbol{A} + \Delta_A))^{\frac{1}{2}} diag(\boldsymbol{1}^\top \boldsymbol{A})^{-\frac{1}{2}}$. *Let* $\bar{\beta}_A = 1 + \beta_A$. *Then the representation robustness of PXGL-GNN to perturbations* $\Delta_A$ *and* $\Delta_X$ *is shown as*

$$\|\tilde{\boldsymbol{g}} - \boldsymbol{g}\| \leq \frac{1}{\sqrt{n}} \rho^L \beta_W^L (\bar{\beta}_A + \|\Delta_A\|_2)^{L-1} (1 + \alpha)^{-L} \left[ (\bar{\beta}_A + 2\|\Delta_A\|_2) \|\Delta_X\|_F + 2L\beta_X \bar{\beta}_A \|\Delta_D\|_2 \right]$$

The bound reveals that PXGL-GNN is sensitive to the graph structure perturbation $\Delta_A$ when $L$ is large and is relatively not sensitive to the feature matrix perturbation on $\Delta_X$. On the other hand, when $\alpha$, the minimum node degree, is larger, the method is more robust.

## 5.2 GENERALIZATION ANALYSIS

Following (Bousquet & Elisseeff, 2002; Feldman & Vondrak, 2019), we use uniform stability to derive the generalization bound for PXGL-GNN. Let $\boldsymbol{\lambda}$ and $\mathbb{W}$ be known parameters. The supervised loss $\ell_{\text{CE}}$ in (11) is guaranteed with a uniform stability parameter $\eta$. For convenience, we let $\ell(\boldsymbol{\lambda}, \bar{\mathbb{W}}; G) := \ell_{\text{CE}}(\boldsymbol{y}, \hat{\boldsymbol{y}})$. Considering the empirical risk $\mathcal{E}[\ell(\boldsymbol{\lambda}, \bar{\mathbb{W}}; \mathcal{G})] := \frac{1}{N} \sum_{i=1}^N \ell(\boldsymbol{\lambda}, \bar{\mathbb{W}}; G_i)$ and true risk $\mathbb{E}[\ell(\boldsymbol{\lambda}, \bar{\mathbb{W}}; G)]$, we have the following high-probability generalization bound: for constant $c$ and $\delta \in (0, 1)$,

$$\mathbf{Pr}\left[ |\mathbb{E}[\ell_{\text{CE}}(\boldsymbol{\lambda}, \bar{\mathbb{W}}; G) - \mathcal{E}[\ell_{\text{CE}}(\boldsymbol{\lambda}, \bar{\mathbb{W}}; \mathcal{G})]| \geq c \left( \eta \log(N) \log\left(\frac{N}{\delta}\right) + \sqrt{\frac{\log(1/\delta)}{N}} \right) \right] \leq \delta. \quad (12)$$

Let $\mathcal{D} := \{G_1, \ldots, G_N\}$ be the training data. By removing the $i$-th graph $G_i$, we get $\mathcal{D}^{\setminus i} = \{G_1, \ldots, G_{i-1}, G_{i+1}, \ldots, G_N\}$. Let $\boldsymbol{\lambda}_\mathcal{D}$ and $\bar{\mathbb{W}}_\mathcal{D} := \{\boldsymbol{W}_C, \boldsymbol{W}_\mathcal{D}^{(m,l)}, \forall m \in [M], l \in [L]\}$ be the parameters trained on $\mathcal{D}$. Let $\boldsymbol{\lambda}_{\mathcal{D}^{\setminus i}}$ and $\bar{\mathbb{W}}_{\mathcal{D}^{\setminus i}} := \{\boldsymbol{W}_{C^{\setminus i}}, \boldsymbol{W}_{\mathcal{D}^{\setminus i}}^{(m,l)}, \forall m \in [M], l \in [L]\}$ be the parameters trained on $\mathcal{D}^{\setminus i}$. We aim to find an $\eta$ such that

$$|\ell_{\text{CE}}(\boldsymbol{\lambda}_\mathcal{D}, \bar{\mathbb{W}}_\mathcal{D}; G) - \ell_{\text{CE}}(\boldsymbol{\lambda}_{\mathcal{D}^{\setminus i}}, \bar{\mathbb{W}}_{\mathcal{D}^{\setminus i}}; G)| \leq \eta \quad (13)$$

We have the following result for $\eta$.

**Theorem 5.2.** *Suppose* $\max\{\max_{m \in [M], l \in [L]} \|\boldsymbol{W}_\mathcal{D}^{(m,l)}\|_2, \max_{m \in [M], l \in [L]} \|\boldsymbol{W}_{\mathcal{D}^{\setminus i}}^{(m,l)}\|_2\} \leq \hat{\beta}_W$ *and* $\max_{m \in [M], l \in [L]} \|\boldsymbol{W}_\mathcal{D}^{(m,l)} - \boldsymbol{W}_{\mathcal{D}^{\setminus i}}^{(m,l)}\|_2 \leq \hat{\beta}_{\Delta W}$, $\|\boldsymbol{W}_C - \boldsymbol{W}_{C^{\setminus i}}\|_2 \leq \gamma_{\Delta C}$, $\|\boldsymbol{W}_{C^{\setminus i}}\|_2 \leq \gamma_C$.

*Suppose the $f_c$ in $\ell_{CE}$ (11) is a linear classifier, which is $\tau$-Lipschitz continuous. Suppose Thus the $\eta$ for estimation error (12) and uniform stability (13) is:*

$$\eta = \frac{\tau}{\sqrt{n}} \rho^L \hat{\beta}_W^{L-1} \beta_X (1 + \beta_A)^L (1 + \alpha)^{-L} \left[ \hat{\beta}_W \gamma_{\Delta C} + \gamma_C \left( 2\hat{\beta}_W + L\hat{\beta}_{\Delta W} \right) \right] \quad (14)$$

Invoking (14) into (12), we obtain the generalization error bound of our model. We see that when $\alpha$ is larger and $\beta_A, \beta_X$ are smaller, the generalization ability is stronger. It is worth noting that in the proof (see (35)) of the theorem, we used an aggressive relaxation such that $\boldsymbol{\lambda}$ was not present in $\eta$. By keeping $\boldsymbol{\lambda}$, we can obtain

$$\eta = \frac{\tau}{\sqrt{n}} \rho^L \hat{\beta}_W^{L-1} \beta_X (1 + \beta_A)^L (1 + \alpha)^{-L} \left[ \hat{\beta}_W \gamma_{\Delta C} + \gamma_C \left( \hat{\beta}_W \| \boldsymbol{\lambda}_{\mathcal{D}} - \boldsymbol{\lambda}_{\mathcal{D} \setminus i} \| + L\hat{\beta}_{\Delta W} \| \boldsymbol{\lambda}_{\mathcal{D} \setminus i} \| \right) \right]$$

(15)

Since $\| \boldsymbol{\lambda}_{\mathcal{D}} \|_1 = \| \boldsymbol{\lambda}_{\mathcal{D} \setminus i} \|_1 = 1$, when $M$ is larger, $\| \boldsymbol{\lambda}_{\mathcal{D}} - \boldsymbol{\lambda}_{\mathcal{D} \setminus i} \|$ and $\| \boldsymbol{\lambda}_{\mathcal{D} \setminus i} \|$ are potentially smaller. This means that when we include more graph patterns, the generalization bound of our PXGL-GNN becomes tighter, which potentially leads to higher classification accuracy.

### 5.3 Time and Space Complexity

Given a dataset with $N$ graphs (each has $n$ nodes and $e$ edges), we select $M$ different patterns and sample $Q$ subgraphs of each pattern. First, our PXGL-EGK requires computing $M$ kernel matrices, of which the space complexity is $\mathcal{O}(MN^2)$ and the time complexity is $\mathcal{O}(N^2 \sum_{m=1}^{M} \psi_i)$, where $\psi_i$ denotes the time complexity of the $m$-th graph kernel. For instance, the time complexities of the graphlet kernel, shortest path kernel, and Weisfeiler-Lehman Subtree kernel are $\mathcal{O}(n^k)$, $\mathcal{O}(n^4)$, and $\mathcal{O}(hn + he)$ respectively, where $k$ and $h$ are some kernel-specific hyperparameters. When $N$ is large, the method has high time and space complexities. Regarding PXGL-GNN, suppose each representation learning function $F_m$ is an $L$-layer GCN, of which the width is linear with $d$. For both supervised and unsupervised learning, suppose the batch size and the number of iterations in the optimization are $B$ and $T$ respectively. Then, in supervised learning, the space complexity and time complexity are $\mathcal{O}(BMQ(e + nd) + MLd^2 + Cd)$ and $\mathcal{O}(TBMQL(ed + nd^2) + NQ \sum_{m=1}^{M} \vartheta_m)$ respectively, where $\vartheta_m$ denotes the time complexity of generating a sample of the $m$-th pattern. For instance, when the $m$-th pattern is graphlets with size $k \in \{3, 4, 5\}$, we have $\vartheta_m \leq nu^{k-1}$ (Shervashidze et al., 2009), where $u$ denotes the maximum node degree of the graph. In unsupervised learning, the space complexity and time complexity are $\mathcal{O}(BMQ(e + nd) + MLd^2 + Cd + B^2)$ and $\mathcal{O}(TBMQL(ed + nd^2) + TB^2 + NQ \sum_{m=1}^{M} \vartheta_m)$ respectively. PXGL-GNN is scalable to large graph datasets because the complexities are linear with $BMQ$ and $B^2$ and $\vartheta_m$ are controllable.

## 6 Related Works

Due to space limitation, we introduce previous works on explainable graph learning (XGL), graph representation learning (GCL), and graph kernels in Appendix C.

## 7 Experiments

We test our method on the TUdataset (Morris et al., 2020) for both supervised and unsupervised learning tasks, as shown in Table 1. Our goal is to learn explainable graph representations. We provide the weight parameter $\boldsymbol{\lambda}$ and visualize the ensemble representation $\boldsymbol{g}$ and the pattern representation $\boldsymbol{z}^{(m)}$. We use seven graph patterns: paths, trees, graphlets, cycles, cliques, wheels, and stars, sampling $Q = 50$ subgraphs for each. We use a 5-layer GCN for the representation learning function $F$ and a 3-layer DNN with softmax for classification function $f_c$. We repeat the experiments ten times

Table 1: Statistics of Datasets

| Name | # of graphs | # of classes | # of nodes | node labels | node attributes |
|------|-------------|--------------|------------|-------------|-----------------|
| MUTAG | 188 | 2 | 17.9 | yes | no |
| PROTEINS | 1113 | 2 | 39.1 | yes | yes |
| DD | 1178 | 2 | 284.32 | yes | no |
| NCI1 | 4110 | 2 | 29.9 | yes | no |
| COLLAB | 5000 | 3 | 74.49 | no | no |
| IMDB-B | 1000 | 2 | 19.8 | no | no |
| REDDIT-B | 2000 | 2 | 429.63 | no | no |
| REDDIT-M5K | 4999 | 5 | 508.52 | no | no |

and report the average value with standard de-
viation. Due to the space limitation, the results of PXGL-EGK and other figures are shown in Appendix F.

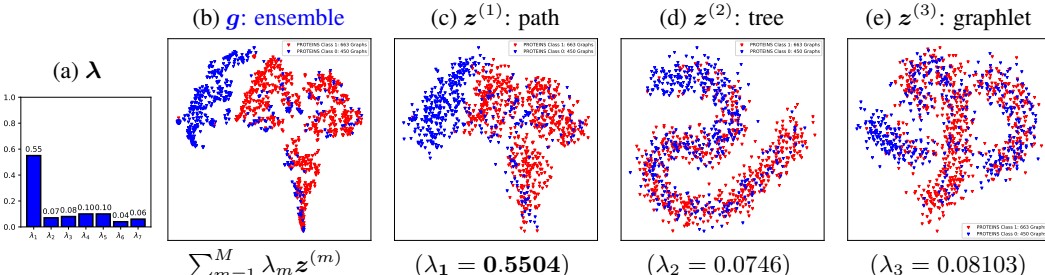

Figure 4: t-SNE visualizations of GNNs' pattern representations (supervised) for PROTEINS.

## 7.1 SUPERVISED LEARNING

We conduct supervised XGL via pattern analysis by solving optimization with the classification loss (11). The dataset is split into 80% training, 10% validation, and 10% testing data. The weight parameter $\lambda$, indicating each pattern's contribution to graph representation learning, is reported in Table 2. We also visualize the graph representation $g$ and three pattern representations $z^{(m)}$ of PROTEINS in Figure 4. Results show the paths pattern is most important for learning $g$, and the ensemble representation $g$ outperforms single pattern representations $z^{(m)}$.

Table 2: $\lambda$ of supervised PXGL-GNN. The largest value is **bold** and the second largest value is blue.

| Pattern | MUTAG | PROTEINS | DD | NCI1 | COLLAB | IMDB-B | REDDIT-B | REDDIT-M5K |
|---|---|---|---|---|---|---|---|---|
| paths | $0.095 \pm 0.014$ | $\mathbf{0.550 \pm 0.070}$ | $0.093 \pm 0.012$ | $0.022 \pm 0.002$ | $\mathbf{0.587 \pm 0.065}$ | $0.145 \pm 0.018$ | $0.131 \pm 0.027$ | $0.027 \pm 0.003$ |
| trees | $0.046 \pm 0.005$ | $0.074 \pm 0.009$ | $0.054 \pm 0.006$ | $0.063 \pm 0.008$ | $0.105 \pm 0.013$ | $0.022 \pm 0.003$ | $0.055 \pm 0.007$ | $0.025 \pm 0.003$ |
| graphlets | $0.062 \pm 0.008$ | $0.081 \pm 0.011$ | $0.125 \pm 0.015$ | $0.101 \pm 0.013$ | $0.063 \pm 0.008$ | $0.084 \pm 0.011$ | $0.026 \pm 0.003$ | $0.054 \pm 0.007$ |
| cycles | $\mathbf{0.654 \pm 0.085}$ | $0.099 \pm 0.013$ | $0.094 \pm 0.012$ | $0.176 \pm 0.022$ | $0.022 \pm 0.003$ | $0.123 \pm 0.016$ | $0.039 \pm 0.005$ | $0.037 \pm 0.005$ |
| cliques | $0.082 \pm 0.011$ | $0.098 \pm 0.012$ | $0.572 \pm 0.073$ | $0.574 \pm 0.075$ | $0.134 \pm 0.017$ | $0.453 \pm 0.054$ | $0.279 \pm 0.069$ | $0.256 \pm 0.067$ |
| wheels | $0.026 \pm 0.003$ | $0.039 \pm 0.005$ | $0.051 \pm 0.007$ | $0.012 \pm 0.002$ | $0.068 \pm 0.009$ | $0.037 \pm 0.004$ | $0.036 \pm 0.005$ | $0.023 \pm 0.003$ |
| stars | $0.035 \pm 0.005$ | $0.056 \pm 0.007$ | $0.011 \pm 0.002$ | $0.052 \pm 0.007$ | $0.021 \pm 0.003$ | $0.136 \pm 0.017$ | $\mathbf{0.447 \pm 0.006}$ | $\mathbf{0.578 \pm 0.033}$ |

We compare our method with classical GNNs including GIN (Xu et al., 2018), DiffPool (Ying et al., 2018), DGCNN (Zhang et al., 2018), GRAPHSAGE (Hamilton et al., 2017), subgraph-based GNNs including SubGNN (Kriege & Mutzel, 2012), SAN (Zhao et al., 2018), SAGNN (Zeng et al., 2023a), and recent methods including S2GAE (Tan et al., 2023) and ICL (Zhao et al., 2024). The accuracies in Table 3 show that our method performs the best.

Table 3: Graph Classification Accuracy (%). The best accuracy is **bold** and the second best is blue.

| Method | MUTAG | PROTEINS | DD | NCI1 | COLLAB | IMDB-B | REDDIT-B | REDDIT-M5K |
|---|---|---|---|---|---|---|---|---|
| GIN | $84.53 \pm 2.38$ | $73.38 \pm 2.16$ | $76.38 \pm 1.58$ | $73.36 \pm 1.78$ | $75.83 \pm 1.29$ | $72.52 \pm 1.62$ | $83.27 \pm 1.30$ | $52.48 \pm 1.57$ |
| DiffPool | $86.72 \pm 1.95$ | $76.07 \pm 1.62$ | $77.42 \pm 2.14$ | $75.42 \pm 2.16$ | $78.77 \pm 1.36$ | $73.55 \pm 2.14$ | $84.16 \pm 1.28$ | $51.39 \pm 1.48$ |
| DGCNN | $84.29 \pm 1.16$ | $75.53 \pm 2.14$ | $76.57 \pm 1.09$ | $74.81 \pm 1.53$ | $77.59 \pm 2.24$ | $72.19 \pm 1.97$ | $86.33 \pm 2.29$ | $53.18 \pm 2.41$ |
| GRAPHSAGE | $86.35 \pm 1.31$ | $74.21 \pm 1.85$ | $79.24 \pm 2.25$ | $77.93 \pm 2.04$ | $76.37 \pm 2.11$ | $73.86 \pm 2.17$ | $85.59 \pm 1.92$ | $51.65 \pm 2.55$ |
| SubGNN | $87.52 \pm 2.37$ | $76.38 \pm 1.57$ | $82.51 \pm 1.67$ | $82.58 \pm 1.79$ | $81.26 \pm 1.53$ | $71.58 \pm 1.20$ | $88.47 \pm 1.83$ | $53.27 \pm 1.93$ |
| SAN | $92.65 \pm 1.53$ | $75.62 \pm 2.39$ | $81.36 \pm 2.10$ | $83.07 \pm 1.54$ | $82.73 \pm 1.92$ | $75.27 \pm 1.43$ | $90.38 \pm 1.54$ | $55.49 \pm 1.75$ |
| SAGNN | $93.24 \pm 2.51$ | $75.61 \pm 2.28$ | $84.12 \pm 1.73$ | $81.29 \pm 1.22$ | $79.94 \pm 1.83$ | $74.53 \pm 2.57$ | $89.57 \pm 2.13$ | $54.11 \pm 1.22$ |
| ICL | $91.34 \pm 2.19$ | $75.44 \pm 1.26$ | $82.77 \pm 1.42$ | $83.45 \pm 1.78$ | $81.45 \pm 1.21$ | $73.29 \pm 1.46$ | $90.13 \pm 1.40$ | $56.21 \pm 1.35$ |
| S2GAE | $89.27 \pm 1.53$ | $76.47 \pm 1.12$ | $84.30 \pm 1.77$ | $82.37 \pm 2.24$ | $82.35 \pm 2.34$ | $75.77 \pm 1.72$ | $90.21 \pm 1.52$ | $54.53 \pm 2.17$ |
| PXGL-GNN | $\mathbf{94.87 \pm 2.26}$ | $\mathbf{78.23 \pm 2.46}$ | $\mathbf{86.54 \pm 1.95}$ | $\mathbf{85.78 \pm 2.07}$ | $\mathbf{83.96 \pm 1.59}$ | $\mathbf{77.35 \pm 2.32}$ | $\mathbf{91.84 \pm 1.69}$ | $\mathbf{57.36 \pm 2.14}$ |

## 7.2 UNSUPERVISED LEARNING

We conduct unsupervised XGL via pattern analysis by solving optimization (with the KL divergence loss (10). The weight parameter $\lambda$ for XGL is reported in Table 4. The visualization of unsupervised XGL results are in Appendix F.4. Results show that the ensemble representation $g$ outperforms single pattern representations $z^{(m)}$.

For clustering performance, we use clustering accuracy (ACC) and Normalized Mutual Information (NMI). Baselines include four kernels: Random walk kernel (RW) (Borgwardt et al., 2005),

Table 4: $\lambda$ of unsupervised PXGL-GNN. The largest value is **bold** and the second largest value is blue.

| Pattern | MUTAG | PROTEINS | DD | NCI1 | COLLAB | IMDB-B | REDDIT-B | REDDIT-M5K |
|---|---|---|---|---|---|---|---|---|
| paths | $0.085 \pm 0.021$ | $\mathbf{0.463 \pm 0.057}$ | $0.083 \pm 0.010$ | $0.023 \pm 0.001$ | $\mathbf{0.478 \pm 0.046}$ | $0.153 \pm 0.018$ | $0.101 \pm 0.007$ | $0.084 \pm 0.006$ |
| trees | $0.027 \pm 0.005$ | $0.082 \pm 0.008$ | $0.069 \pm 0.007$ | $0.042 \pm 0.002$ | $0.127 \pm 0.017$ | $0.082 \pm 0.009$ | $0.060 \pm 0.003$ | $0.036 \pm 0.002$ |
| graphlets | $0.074 \pm 0.009$ | $0.085 \pm 0.010$ | $0.172 \pm 0.020$ | $0.105 \pm 0.012$ | $0.055 \pm 0.006$ | $0.098 \pm 0.011$ | $0.025 \pm 0.002$ | $0.055 \pm 0.005$ |
| cycles | $\mathbf{0.546 \pm 0.065}$ | $0.095 \pm 0.011$ | $0.108 \pm 0.013$ | $0.276 \pm 0.033$ | $0.022 \pm 0.002$ | $0.124 \pm 0.014$ | $0.043 \pm 0.005$ | $0.028 \pm 0.003$ |
| cliques | $0.197 \pm 0.023$ | $0.207 \pm 0.025$ | $\mathbf{0.527 \pm 0.063}$ | $\mathbf{0.482 \pm 0.058}$ | $0.243 \pm 0.029$ | $\mathbf{0.423 \pm 0.051}$ | $0.212 \pm 0.061$ | $0.157 \pm 0.067$ |
| wheels | $0.032 \pm 0.003$ | $0.036 \pm 0.004$ | $0.018 \pm 0.002$ | $0.013 \pm 0.001$ | $0.044 \pm 0.005$ | $0.035 \pm 0.004$ | $0.036 \pm 0.003$ | $0.025 \pm 0.013$ |
| stars | $0.039 \pm 0.004$ | $0.032 \pm 0.002$ | $0.023 \pm 0.003$ | $0.059 \pm 0.007$ | $0.031 \pm 0.001$ | $0.085 \pm 0.010$ | $\mathbf{0.455 \pm 0.019}$ | $\mathbf{0.585 \pm 0.022}$ |

Sub-tree kernels (Da San Martino et al., 2012; Smola & Vishwanathan, 2002), Graphlet kernels (Pržulj, 2007), Weisfeiler-Lehman (WL) kernels (Kriege & Mutzel, 2012); and three unsupervised graph representation learning methods with Gaussian kernel in (10): InfoGraph (Sun et al., 2019), GCL (You et al., 2020), GraphACL (Luo et al., 2023). The results are in Table 5. Our method outperformed all competitors in almost all cases.

Table 5: ACC and NMI of Graph Clustering. The best ACC is **bold** and the the second best ACC is blue. The best NMI is green and the second best NMI is with $^*$.

| Method | Metric | MUTAG | PROTEINS | DD | NCI1 | COLLAB | IMDB-B | REDDIT-B | REDDIT-M5K |
|---|---|---|---|---|---|---|---|---|---|
| RW | ACC | $0.724 \pm 0.023$ | $0.718 \pm 0.019$ | $0.529 \pm 0.017$ | $0.519 \pm 0.025$ | $0.596 \pm 0.019$ | $0.669 \pm 0.028$ | $\geq 1$ day | $\geq 1$ day |
| | NMI | $0.283 \pm 0.008$ | $0.226 \pm 0.008$ | $0.207 \pm 0.003$ | $0.218 \pm 0.003$ | $0.218 \pm 0.005$ | $0.295 \pm 0.006$ | $\geq 1$ day | $\geq 1$ day |
| sub-tree | ACC | $0.716 \pm 0.017$ | $0.683 \pm 0.023$ | $0.563 \pm 0.026$ | $0.532 \pm 0.016$ | $0.533 \pm 0.021$ | $0.627 \pm 0.022$ | $\geq 1$ day | $\geq 1$ day |
| | NMI | $0.217 \pm 0.005$ | $0.167 \pm 0.004$ | $0.225 \pm 0.005$ | $0.295 \pm 0.004$ | $0.198 \pm 0.005$ | $0.254 \pm 0.007$ | $\geq 1$ day | $\geq 1$ day |
| Graphlet | ACC | $0.727 \pm 0.020$ | $0.654 \pm 0.017$ | $\mathbf{0.581 \pm 0.014}$ | $0.526 \pm 0.032$ | $0.525 \pm 0.026$ | $0.617 \pm 0.019$ | $\geq 1$ day | $\geq 1$ day |
| | NMI | $0.225 \pm 0.003$ | $0.131 \pm 0.009$ | $0.320 \pm 0.009$ | $0.273 \pm 0.005$ | $0.217 \pm 0.003$ | $0.210 \pm 0.004$ | $\geq 1$ day | $\geq 1$ day |
| WL | ACC | $0.695 \pm 0.031$ | $0.647 \pm 0.032$ | $0.517 \pm 0.020$ | $0.517 \pm 0.028$ | $0.569 \pm 0.017$ | $0.635 \pm 0.017$ | $\geq 1$ day | $\geq 1$ day |
| | NMI | $0.185 \pm 0.007$ | $0.135 \pm 0.001$ | $0.192 \pm 0.008$ | $0.234 \pm 0.007$ | $0.253 \pm 0.007$ | $0.261 \pm 0.003$ | $\geq 1$ day | $\geq 1$ day |
| InfoGraph | ACC | $0.729 \pm 0.021$ | $0.716 \pm 0.019$ | $0.549 \pm 0.035$ | $0.535 \pm 0.012$ | $0.597 \pm 0.020$ | $0.624 \pm 0.016$ | $0.582 \pm 0.023$ | $0.597 \pm 0.019$ |
| | NMI | $0.236 \pm 0.005$ | $0.231 \pm 0.003$ | $0.266 \pm 0.004$ | $0.263 \pm 0.005$ | $0.311 \pm 0.008$ | $0.198 \pm 0.005$ | $0.206 \pm 0.006$ | $0.286^* \pm 0.006$ |
| GCL | ACC | $0.761 \pm 0.014$ | $0.723 \pm 0.025$ | $0.563 \pm 0.016$ | $0.558 \pm 0.010$ | $0.582 \pm 0.015$ | $0.653 \pm 0.024$ | $0.573 \pm 0.015$ | $0.582 \pm 0.017$ |
| | NMI | $0.337 \pm 0.003$ | $0.258 \pm 0.002$ | $0.289 \pm 0.009$ | $0.341 \pm 0.002$ | $0.293 \pm 0.009$ | $0.253 \pm 0.008$ | $0.195 \pm 0.005$ | $0.266 \pm 0.005$ |
| GraphACL | ACC | $0.742 \pm 0.023$ | $0.731 \pm 0.027$ | $0.572 \pm 0.027$ | $0.522 \pm 0.013$ | $0.554 \pm 0.013$ | $0.679 \pm 0.013$ | $0.594 \pm 0.014$ | $0.567 \pm 0.023$ |
| | NMI | $0.347^* \pm 0.007$ | $0.274^* \pm 0.008$ | $0.312 \pm 0.003$ | $0.260 \pm 0.007$ | $0.236 \pm 0.006$ | $0.315^* \pm 0.007$ | $0.215^* \pm 0.006$ | $0.238 \pm 0.009$ |
| PXGL-GNN | ACC | $\mathbf{0.778 \pm 0.029}$ | $\mathbf{0.746 \pm 0.019}$ | $0.576 \pm 0.035$ | $\mathbf{0.564 \pm 0.013}$ | $\mathbf{0.612 \pm 0.014}$ | $\mathbf{0.686 \pm 0.027}$ | $\mathbf{0.616 \pm 0.017}$ | $\mathbf{0.608 \pm 0.023}$ |
| | NMI | $0.352 \pm 0.006$ | $0.292 \pm 0.010$ | $0.317^* \pm 0.003$ | $0.327^* \pm 0.008$ | $0.372 \pm 0.007$ | $0.324 \pm 0.011$ | $0.224 \pm 0.009$ | $0.295 \pm 0.012$ |

## 8 CONCLUSION

This paper studied the explainability of graph representations. We proposed two strategies to learn and explain effective graph representations. The first one is based on graph ensemble kernel and the second one is based GNNs that learns from different graph patterns such as path, tree, etc. We also provide some theoretical analysis for the proposed method, including robustness analysis and generalization bound. The experiments showed that our method not only provides higher accuracy of classification and clustering than its competitors but also yields explainable results.

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

## A  APPENDIX

You may include other additional sections here.

## B  MATH DEFINITIONS OF PATTERNS

In our work, graph patterns refer to as subgraphs with practical meanings. Let $G = (V, E)$ be a graph. A subgraph $S = (V_S, E_S)$ of $G$ is defined such that $V_S \subseteq V$ and $E_S \subseteq E \cap (V_S \times V_S)$. The math definitions of graph patterns are as follows:

- **Paths:** $S$ is a *path* if there exists a sequence of distinct vertices $v_1, \dots, v_k \in V_S$ such that $E_S = ((v_i, v_{i+1}) : i = 1, \dots, k - 1)$.

- **Trees:** $S$ is a *tree* if it is connected and contains no cycles, i.e., it is acyclic and $|E_S| = |V_S| - 1$.

- **Graphlets:** $S$ is a *graphlet* if it is a small connected induced subgraph of $G$, typically consisting of 2 to 5 vertices.

- **Cycles:** $S$ is a *cycle* if there exists a sequence of distinct vertices $v_1, \dots, v_k \in V_S$ such that $E_S = ((v_i, v_{i+1}) : i = 1, \dots, k - 1) \cup ((v_k, v_1))$.

- **Cliques:** $S$ is a *clique* if every two distinct vertices in $V_S$ are adjacent, thus $E_S = ((v_i, v_j) : v_i, v_j \in V_S, i \neq j)$.

- **Wheels:** $S$ is a *wheel* if it consists of a cycle with vertices $v_1, \dots, v_{k-1}$ and an additional central vertex $v_k$ such that $v_k$ is connected to all vertices of the cycle.

- **Stars:** $S$ is a *star* if it consists of one central vertex $v_c$ and several leaf vertices $v_1, \dots, v_{k-1}$, where each leaf vertex is only connected to $v_c$. Thus, $E_S = ((v_c, v_i) : i = 1, \dots, k - 1)$.

## C  RELATED WORKS

In this section, we introduce previous works on explainable graph learning (XGL), graph representation learning (GRL), and graph kernels.

## C.1 Explainable Graph Learning (XGL)

Explainable artificial intelligence (XAI) is a rapidly growing area in the AI community (Došilović et al., 2018; Adadi & Berrada, 2018; Angelov et al., 2021; Hassija et al., 2024). Explainable graph learning (XGL) (Kosan et al., 2023) can be roughly classified into two categories: model-level methods and instance-level methods.

**Model-level**  Model-level or global explanations aim to understand the overall behavior of a model by identifying patterns in its predictions. For example, XGNN(Yuan et al., 2020) trains a graph generator to create graph patterns that maximize a certain prediction, providing high-level insights into GNN behavior. GLG-Explainer(Azzolin et al., 2022) combines local explanations into a logical formula over graphical concepts, offering human-interpretable global explanations aligned with ground-truth or domain knowledge. GCFExplainer(Huang et al., 2023) uses global counterfactual reasoning to find representative counterfactual graphs, providing a summary of global explanations through vertex-reinforced random walks on an edit map of graphs.

**Instance-level**  Instance-level methods offer explanations tailored to specific predictions, focusing on why particular instances are classified in a certain manner. For instance, GNNExplainer (Ying et al., 2019) identifies a compact subgraph structure and a small subset of node features crucial for a GNN's prediction. PGExplainer (Luo et al., 2020) trains a graph generator to incorporate global information and uses a deep neural network (DNN) to parameterize the explanation generation process. SubgraphX (Yuan et al., 2021) efficiently explores different subgraphs using Monte Carlo tree search to explain predictions. RG-Explainer (Shan et al., 2021) constructs a connected explanatory subgraph by sequentially adding nodes, consistent with the message passing scheme. MixupExplainer (Zhang et al., 2023a) introduces a general form of Graph Information Bottleneck (GIB) to address distribution shifting issues in post-hoc graph explanation. AutoGR (Wang et al., 2021) introduces an explainable AutoML approach for graph representation learning. UNR-Explainer (Kang et al., 2024) identifies the top-k most important nodes in a graph to determine the most significant subgraph. It is a classic instance-level explainable graph learning method focused on node representation. However, this task is entirely different from our approach, as it addresses node-level representation rather than representation-level explainability. For this reason, we did not include a comparison.

## C.2 Graph Representation Learning

Graph representation learning is crucial for transforming complex graphs into vectors, particularly for tasks like classification. The methods for graph representation learning are mainly classified into two categories: supervised and unsupervised learning.

**Supervised Representation Learning**  Most GNNs can be used in supervised graph representation learning tasks by aggregating all the node embeddings into a graph representation using a readout function (Hamilton, 2020; Chami et al., 2022). Besides traditional GNNs like GCN (Kipf & Welling, 2016), GIN (Xu et al., 2018), and GAT (Veličković et al., 2017), recent works include: Template-based Fused Gromov-Wasserstein (FGW) (Vincent-Cuaz et al., 2022) computes a vector of FGW distances to learnable graph templates, acting as an alternative to global pooling layers. Path Isomorphism Network (PIN) (Truong & Chin, 2024) introduces a graph isomorphism test and a topological message-passing scheme operating on path complexes. Graph U-Net (Amouzad et al., 2024) proposes GIUNet for graph classification, combining node features and graph structure information using a pqPooling layer. Unified Graph Transformer Networks (UGT) (Lee et al., 2024) integrate local and global structural information into fixed-length vector representations using self-attention. CIN++ (Giusti et al., 2023) enhances topological message passing to account for higher-order and long-range interactions, achieving state-of-the-art results. Graph Joint-Embedding Predictive Architectures (Graph-JEPA) (Skenderi et al., 2023) use masked modeling to learn embeddings for subgraphs and predict their coordinates on the unit hyperbola in the 2D plane.

**Unsupervised Representation Learning**  Unsupervised methods aim to learn graph representations without labeled data. Notable methodologies include: InfoGraph (Sun et al., 2019) emphasizes mutual information between graph-level and node-level representations. Graph Contrastive Learn-

ing techniques (You et al., 2020; Suresh et al., 2021; You et al., 2021) enhance graph representations through diverse augmentation strategies. AutoGCL (Yin et al., 2022) introduces learnable graph view generators. GraphACL (Luo et al., 2023) adopts a novel self-supervised approach. InfoGCL (Xu et al., 2021) and SFA (Zhang et al., 2023b) focus on information transfer and feature augmentation in contrastive learning. Techniques like GCS (Wei et al., 2023), NCLA (Shen et al., 2023), $S^3$-CL (Ding et al., 2023), and ImGCL (Zeng et al., 2023b) refine graph augmentation and learning methods. GRADATE (Duan et al., 2023) integrates subgraph contrast into multi-scale learning networks.

**GNNs using Subgraphs and Substructures**   Our pattern analysis method samples subgraphs from different graph patterns to conduct explainable graph representation learning. The key novelty and contribution of our paper is that graph pattern analysis provides explainability for representations. We discuss other GNN methods based on subgraphs and substructures here: Subgraph Neural Networks (SubGNN) (Kriege & Mutzel, 2012) learn disentangled subgraph representations using a novel subgraph routing mechanism, but they sample subgraphs randomly, lacking explainability. Substructure Aware Graph Neural Networks (SAGNN) (Zeng et al., 2023a) use cut subgraphs and return probability to capture structural information but focus on expressiveness rather than explainability. Mutual Information (MI) Induced Substructure-aware GRL (Wang et al., 2020) maximizes MI between original and learned representations at both node and graph levels but does not provide explainable representation learning. Substructure Assembling Network (SAN) (Zhao et al., 2018) hierarchically assembles graph components using an RNN variant but lacks explainability in representation learning.

Several works focus on analyzing the expressiveness of methods by their ability to count substructures, but they do not provide explainable representation learning. For example: (Chen et al., 2020) analyze the expressiveness of MPNNs (Gilmer et al., 2017) and 2nd-order Invariant Graph Networks (2-IGNs) (Maron et al., 2019) based on their ability to count specific subgraphs, highlighting tasks that are challenging for classical GNN architectures but not focusing on explainability. (Frasca et al., 2022) compare the expressiveness of SubGNN (Kriege & Mutzel, 2012) and 2-IGNs (Maron et al., 2019) using symmetry analysis, establishing a link between Subgraph GNNs and Invariant Graph Networks.

## C.3   GRAPH KERNELS

Graph kernels evaluate the similarity between two graphs. Over the past decades, numerous graph kernels have been proposed (Siglidis et al., 2020). We classify them into two categories: pattern counting kernels and non-pattern counting kernels.

**Pattern Counting Kernels**   Pattern counting kernels compare specific substructures within graphs to evaluate similarity (Kriege et al., 2020). For examples, Random walk kernels (Borgwardt et al., 2005; Gärtner et al., 2003) measure graph similarity by counting common random walks between graphs. Shortest-path kernels(Borgwardt & Kriegel, 2005) compare graphs using the shortest distance matrix generated by the Floyd-Warshall algorithm, based on edge values and node labels. Sub-tree kernels (Da San Martino et al., 2012; Smola & Vishwanathan, 2002) decompose graphs into ordered Directed Acyclic Graphs (DAGs) and use tree kernels extended to DAGs. Graphlet kernels (Pržulj, 2007) count small connected non-isomorphic subgraphs (graphlets) within graphs and compare their distributions. Weisfeiler-Lehman subtree kernels (Kriege & Mutzel, 2012) use small subgraphs, like graphlets, to compare graphs, allowing flexibility to compare vertex and edge attributes with arbitrary kernel functions.

**Non-pattern Counting Kernels**   Non-pattern counting kernels evaluate graph similarity without relying on specific substructure counts. For examples, Neighborhood hash kernel (Hido & Kashima, 2009) use binary arrays to represent node labels and logical operations on connected node labels. This kernel has linear time complexity. GraphHopper kernel (Feragen et al., 2013) compare shortest paths between node pairs using kernels on nodes encountered while hopping along shortest paths. Graph hash kernel (Shi et al., 2009) use hashing for efficient kernel computation, suitable for data streams and sparse feature spaces, with deviation bounds from the exact kernel matrix. Multiscale Laplacian Graph (MLG) kernel (Kondor & Pan, 2016) account for structure at different scales using

Feature Space Laplacian Graph (FLG) kernels, applied recursively to subgraphs. They introduce a randomized projection procedure similar to the Nystrom method for RKHS operators.

# D   PROOF FOR ROBUSTNESS ANALYSIS

Let $\Delta_A$ and $\Delta_X$ be some perturbations on adjacency matrix and node attributes, then the perturbed graph is denoted as $\tilde{G} = (A + \Delta_A, X + \Delta_X)$. Let $g$ be the graph representation of $G$ and $\tilde{g}$ be the graph representation of $\tilde{G}$. The robustness analysis is to find the upper bound of $\|\tilde{g} - g\|$.

**Assumptions and Notations:**  Let $\tilde{A} = A + \Delta_A$ and $\tilde{X} = X + \Delta_X$. We suppose that $\|A\|_2 \leq \beta_A$, $\|X\|_F \leq \beta_B$ and $\|W^{(m,l)}\|_2 \leq \beta_W$, $(\forall\ m \in [M],\ l \in [L])$, the activation $\sigma(\cdot)$ of GCN is $\rho$-Lipschitz continuous. We denote the minimum node degree of $G$ as $\alpha$, the effects of structural perturbation as $\kappa = \min(\mathbf{1}^\top \Delta_A)$, and $\Delta_D := I - \text{diag}(\mathbf{1}^\top(I + A + \Delta_A))^{\frac{1}{2}} \text{diag}(\mathbf{1}^\top A)^{-\frac{1}{2}}$.

**Theorem:**  Our conclusion for robustness analysis is as follows:

$$\|\tilde{g} - g\| \leq \frac{1}{\sqrt{n}} \rho^L \beta_W^L \beta_X (1+\alpha)^{-L} (1 + \beta_A + \|\Delta_A\|_2)^L \Big( 1 + 2L\|\Delta_D\|_2 + L(1 + \beta_A + \|\Delta_A\|_2)^{-1}\|\Delta_A\|_2 \Big) \tag{16}$$

To provide a clearer analysis, we first use the whole graph $G$ and $\tilde{G}$ as the input of the pattern representation learning function $F$ without sampling the subgraphs. Then we consider using the subgraph sampling to analyze $g$ and $\tilde{g}$ and finally finish the proof of robustness analysis.

## D.1   LEARNING PATTERN REPRESENTATIONS USING THE WHOLE GRAPH WITHOUT SAMPLING

In this section, we first consider using the whole graph $G$ and $\tilde{G}$ as the input of the pattern representation learning function $F$ without sampling the subgraphs, i.e., we analyze $F(A, X; \mathcal{W}^{(m)})$ and $F(\tilde{A}, \tilde{X}; \mathcal{W}^{(m)})$.

**Representation Learning Function $F$**   In theoretical analysis, we suppose the pattern representation learning function $F$ is a $L$-layer GCN (Kipf & Welling, 2016) with an average pooling avg-pool : $\mathbb{R}^{n \times d} \to \mathbb{R}^d$ as the output layer. The pattern learning function for the pattern $\mathcal{P}_m$ is denoted as $F(A, X; \mathcal{W}^{(m)})$, where $\mathcal{W}^{(m)} = \{W^{(m,1)}, ..., W^{(m,l)}, ..., W^{(m,L)}\}$ and $W^{(m,l)}$ is the trainable parameter of the $l$-th layer. We use the adjacency matrix $A$ and node feature matrix $X$ of $G$ as the input. Then the self-connected adjacency matrix is $\hat{A} = I + A$, the diagonal matrix is $\hat{D} = \text{diag}(\mathbf{1}^\top \hat{A})$, then the normalized self-connected adjacency matrix is $U = \hat{D}^{-\frac{1}{2}} \hat{A} \hat{D}^{-\frac{1}{2}}$. Let $\sigma(\cdot)$ be an activation function, then the hidden embedding $X^{(m,L)}$ of the $l$-th layer is defined as follows

$$X^{(m,l)} = \sigma(\underbrace{U...\sigma(U}_{l\ \text{times}} X \underbrace{W^{(m,1)})...W^{(m,l)}}_{l\ \text{times}}), \ \forall\, l \in [L], \tag{17}$$

The pattern representation $z^{(m)}$ of pattern $\mathcal{P}_m$ is obtained by

$$z^{(m)} = F(A, X; \mathcal{W}^{(m)}) = \text{avg-pool}(X^{(m,L)}) = \frac{1}{n} \mathbf{1}^\top X^{(m,L)} \tag{18}$$

For a perturbed graph $\tilde{G}$, we use $\tilde{A}$ and $\tilde{X}$ to denote the adjacency matrix and feature matrix respectively. The corresponding self-connected adjacency matrix is $\hat{A}' = I + \tilde{A}$ and the degree matrix as $\hat{D}' = \text{diag}(\mathbf{1}^\top \hat{A}')$. Then the normalized self-connected adjacency matrix is $\tilde{U} = \hat{D}'^{-\frac{1}{2}} \hat{A}' \hat{D}'^{-\frac{1}{2}}$. The $l$-th layer hidden embedding of $\tilde{G}$ is defined as follows

$$\tilde{X}^{(m,l)} = \sigma(\underbrace{\tilde{U}...\sigma(\tilde{U}}_{l\ \text{times}} \tilde{X} \underbrace{W^{(m,1)})...W^{(m,l)}}_{l\ \text{times}}), \ \forall\, l \in [L], \tag{19}$$

The perturbed pattern representation $\tilde{z}^{(m)}$ of pattern $\mathcal{P}_m$ is obtained by

$$\tilde{z}^{(m)} = F(\tilde{A}, \tilde{X}; \mathcal{W}^{(m)}) = \text{avg-pool}(\tilde{X}^{(m,L)}) = \frac{1}{n} \mathbf{1}^\top \tilde{X}^{(m,L)} \tag{20}$$

**Lemma D.1.** *Let $X$ and $Y$ be two square matrices, $\|\cdot\|_2$ be the spectral norm and $\|\cdot\|_F$ be the Frobenius norm , then $\|X\|_2 \leq \|X\|_F$, $\|XY\|_2 \leq \|X\|_2\|Y\|_2$ and $\|XY\|_F \leq \|X\|_2\|Y\|_F$.*

**Lemma D.2** (Inequalities). *Some inequalities that will be used in our proof:*

$$\|U\|_2 \leq (1+\alpha)^{-1}(1+\beta_A)$$
$$\|\tilde{U}\|_2 \leq (1+\alpha+\kappa)^{-1}(1+\beta_A+\|\Delta_A\|_2)$$
$$\|\Delta_U\|_2 \leq 2(1+\beta_A)(1+\alpha)^{-1}\|\Delta_D\|_2 + (1+\alpha+\kappa)^{-1}\|\Delta_A\|_2$$
$$\|\Delta_{X^{(m,l)}}\|_F \leq \rho^l\beta_W^l\beta_X(1+\alpha)^{-l}(1+\beta_A+\|\Delta_A\|_2)^l\left(1+2l\|\Delta_D\|_2+l(1+\beta_A+\|\Delta_A\|_2)^{-1}\|\Delta_A\|_2\right)$$

*Proof.* Since the minimum node degree of $G$ is $\alpha$, then we have $\|\hat{D}^{-\frac{1}{2}}\|_2 \leq (1+\alpha)^{-\frac{1}{2}}$. Since $\|A\|_2 \leq \beta_A$, then $\|\hat{A}\|_2 \leq 1+\beta_A$. We have

$$\|U\|_2 \leq \|\hat{D}^{-\frac{1}{2}}\|_2\|\hat{A}\|_2\|\hat{D}^{-\frac{1}{2}}\|_2 \leq (1+\alpha)^{-1}(1+\beta_A). \tag{21}$$

Similarly, since the effects of structural perturbation is $\kappa = \min(\mathbf{1}^\top\Delta_A)$, we have $\|\hat{D}'^{-\frac{1}{2}}\|_2 \leq (1+\alpha+\kappa)^{-\frac{1}{2}}$. Since $\|\tilde{A}'\|_2 \leq \|\hat{A}\|_2 + \|\Delta_A\|_2 \leq 1+\beta_A+\|\Delta_A\|_2$, we obtain

$$\|\tilde{U}\|_2 \leq \|\hat{D}'^{-\frac{1}{2}}\|_2\|\hat{A}'\|_2\|\hat{D}'^{-\frac{1}{2}}\|_2 \leq (1+\alpha+\kappa)^{-1}(1+\beta_A+\|\Delta_A\|_2). \tag{22}$$

Letting $\Delta_U = \tilde{U} - U$, we have

$$\begin{aligned}
\|\Delta_U\|_2 &= \|\tilde{U} - U\|_2 = \|\hat{D}'^{-\frac{1}{2}}(\hat{A}+\Delta_A)\hat{D}'^{-\frac{1}{2}} - \hat{D}^{-\frac{1}{2}}\hat{A}\hat{D}^{-\frac{1}{2}}\|_2 \\
&= \|\hat{D}'^{-\frac{1}{2}}\hat{A}\hat{D}'^{-\frac{1}{2}} - \hat{D}'^{-\frac{1}{2}}\hat{A}\hat{D}^{-\frac{1}{2}} + \hat{D}'^{-\frac{1}{2}}\hat{A}\hat{D}^{-\frac{1}{2}} - \hat{D}^{-\frac{1}{2}}\hat{A}\hat{D}^{-\frac{1}{2}} + \hat{D}'^{-\frac{1}{2}}\Delta_A\hat{D}'^{-\frac{1}{2}}\|_2 \\
&\leq \|\hat{D}'^{-\frac{1}{2}}\hat{A}(\hat{D}'^{-\frac{1}{2}} - \hat{D}^{-\frac{1}{2}})\|_2 + \|(\hat{D}'^{-\frac{1}{2}} - \hat{D}^{-\frac{1}{2}})\hat{A}\hat{D}^{-\frac{1}{2}}\|_2 + \|\hat{D}'^{-\frac{1}{2}}\Delta_A\hat{D}'^{-\frac{1}{2}}\|_2 \\
&\leq (\|\hat{D}^{-\frac{1}{2}}\|_2 + \|\hat{D}'^{-\frac{1}{2}}\|_2)\|\hat{A}\|_2\|\hat{D}'^{-\frac{1}{2}} - \hat{D}^{-\frac{1}{2}}\|_2 + \|\hat{D}'^{-\frac{1}{2}}\|_2\|\Delta_A\|_2\|\hat{D}'^{-\frac{1}{2}}\|_2 \\
&\leq ((1+\alpha)^{-\frac{1}{2}} + (1+\alpha+\kappa)^{-\frac{1}{2}})(1+\beta_A)\|\hat{D}'^{-\frac{1}{2}} - \hat{D}^{-\frac{1}{2}}\|_2 + (1+\alpha+\kappa)^{-1}\|\Delta_A\|_2 \\
&\leq 2(1+\beta_A)(1+\alpha)^{-\frac{1}{2}}\|\hat{D}'^{-\frac{1}{2}} - \hat{D}^{-\frac{1}{2}}\|_2 + (1+\alpha+\kappa)^{-1}\|\Delta_A\|_2 \\
&\leq 2(1+\beta_A)(1+\alpha)^{-\frac{1}{2}}(1+\alpha+\kappa)^{-\frac{1}{2}}\|I - \hat{D}'^{\frac{1}{2}}\hat{D}^{-\frac{1}{2}}\|_2 + (1+\alpha+\kappa)^{-1}\|\Delta_A\|_2 \\
&= 2(1+\beta_A)(1+\alpha)^{-\frac{1}{2}}(1+\alpha+\kappa)^{-\frac{1}{2}}\|\Delta_D\|_2 + (1+\alpha+\kappa)^{-1}\|\Delta_A\|_2 \\
&\leq 2(1+\beta_A)(1+\alpha)^{-1}\|\Delta_D\|_2 + (1+\alpha+\kappa)^{-1}\|\Delta_A\|_2
\end{aligned} \tag{23}$$

where $\Delta_D = I - \hat{D}'^{\frac{1}{2}}\hat{D}^{-\frac{1}{2}} = I - \text{diag}(\mathbf{1}^\top(I+A+\Delta_A))^{\frac{1}{2}}\text{diag}(\mathbf{1}^\top A)^{-\frac{1}{2}}$.

The $X^{(m,l)}$ is the hidden embedding of the $l$-layer GCN of $F(A, (X); \mathcal{W}^{(m,l)})$, which is the representation learning function related to $\mathcal{P}_m$. Then we have

$$\begin{aligned}
\|X^{(m,l)}\|_F &= \|\sigma(UX^{(m,l-1)}W^{(m,l)})\|_F \\
&\leq \rho\|UX^{(m,l-1)}W^{(m,l)}\|_F \\
&\leq \rho\|U\|_2\|X^{(m,l-1)}\|_F\|W^{(m,l)}\|_2 \\
&\leq \rho\beta_W(1+\alpha)^{-1}(1+\beta_A)\|X^{(m,l-1)}\|_F \\
&\leq \rho^l\beta_W^l(1+\beta_A)^l(1+\alpha)^{-l}\|X\|_F \\
&\leq \rho^l\beta_W^l\beta_X(1+\beta_A)^l(1+\alpha)^{-l}
\end{aligned} \tag{24}$$

For $\Delta_{X^{(m,l)}} = \tilde{X}^{(m,l)} - X^{(m,l)}$, we have

$$
\begin{aligned}
\|\Delta_{X^{(m,l)}}\|_F &= \|\tilde{X}^{(m,l)} - X^{(m,l)}\|_F \\
&= \|\sigma(\tilde{U}\tilde{X}^{(m,l-1)}W^{(l)}) - \sigma(UX^{(l-1)}W^{(l)})\|_F \\
&\leq \rho\|\tilde{U}\tilde{X}^{(m,l-1)} - UX^{(m,l-1)}\|_F\|W^{(m,l)}\|_2 \\
&\leq \rho\beta_W\left(\|\tilde{U}\|_2\|\Delta_{X^{(m,l-1)}}\|_F + \|\Delta_U\|_2\|X^{(m,l-1)}\|_F\right) \\
&\leq \rho^2\beta_W^2\|\tilde{U}\|_2^2\|\Delta_{X^{(m,l-2)}}\|_F + \rho^2\beta_W^2\|\tilde{U}\|_2\|\Delta_U\|_2\|X^{(m,l-2)}\|_F + \rho\beta_W\|\Delta_U\|_2\|X^{(m,l-1)}\|_F \\
&\leq \rho^l\beta_W^l\|\tilde{U}\|_2^l\|\Delta_X\|_F + \sum_{k=1}^{l}\rho^k\beta_W^k\|\tilde{U}\|_2^{k-1}\|\Delta_U\|_2\|X^{(m,l-k)}\|_F \\
&\leq \rho^l\beta_W^l(1 + \beta_A + \|\Delta_A\|_2)^{l-1}(1 + \alpha)^{-l}\left[(1 + \beta_A + 2\|\Delta_A\|_2)\|\Delta_X\|_F + 2l\beta_X(1 + \beta_A)\|\Delta_D\|_2\right]
\end{aligned}
$$
(25)

$\square$

## D.2 Learning Graph Representations via Sampling Subgraphs

In this section, we consider learning the graph representation $g$ and $\tilde{g}$ respectively by sampling subgraphs of graph patterns. That is, we analyse $F(A_S, X_S; \mathcal{W}^{(m)})$ and $F(\tilde{A}_{\tilde{S}}, \tilde{X}_{\tilde{S}}; \mathcal{W}^{(m)})$. And then we provide the upper bound of $\|\tilde{g} - g\|$.

Let $S$ be a subgraph of graph $G$ and $\tilde{S}$ be a subgraph of graph $\tilde{G}$. Let $\Delta_{A_S}$ and $\Delta_{X_S}$ be some perturbations on adjacency matrix and node attributes, then the perturbed graph is denoted as $\tilde{S} = (A_S + \Delta_{A_S}, X_S + \Delta_{X_S})$.

**Assumptions and Notations:** Let $\tilde{A} = A + \Delta_A$ and $\tilde{X} = X + \Delta_X$. We suppose that $\|A\|_2 \leq \beta_A$, $\|X\|_F \leq \beta_B$ and $\|W^{(m,l)}\|_2 \leq \beta_W$, ($\forall m \in [M]$, $l \in [L]$), the activation $\sigma(\cdot)$ of GCN is $\rho$-Lipschitz continuous. We denote the minimum node degree of $G$ as $\alpha$, the effects of structural perturbation as $\kappa = \min(\mathbf{1}^\top\Delta_A)$, and $\Delta_D := I - \operatorname{diag}(\mathbf{1}^\top(I + A + \Delta_A))^{\frac{1}{2}}\operatorname{diag}(\mathbf{1}^\top A)^{-\frac{1}{2}}$. We present the following useful lemmas.

**Lemma D.3** (Eigenvalue Interlacing Theorem (Hwang, 2004)). *Suppose $A \in \mathbb{R}^{n \times n}$ is symmetric. Let $B \in \mathbb{R}^{m \times m}$ with $m < n$ be a principal submatrix (obtained by deleting both the $i$-th row and $i$-th column for some value of $i$). Suppose $A$ has eigenvalues $\lambda_1 \leq \cdots \leq \lambda_n$ and $B$ has eigenvalues $\beta_1 \leq \cdots \leq \beta_m$. Then*

$$
\lambda_k \leq \beta_k \leq \lambda_{k+n-m} \quad \text{for } k = 1, \cdots, m.
$$

**Lemma D.4.** *Since $X_S$ and $\Delta_{X_S}$ are submatrices of $X$ and $\Delta_X$ respectively, then we have*

$$
\|X_S\|_F \leq \|X\|_F, \quad \text{and} \quad \|\Delta_{X_S}\|_F \leq \|\Delta_X\|_F.
$$

*Let $\Delta_{D_S} := I - \operatorname{diag}(\mathbf{1}^\top(I + A_S + \Delta_{A_S}))^{\frac{1}{2}}\operatorname{diag}(\mathbf{1}^\top A_S)^{-\frac{1}{2}}$. Base on the Eigenvalue Interlacing Theorem, for any subgraph $S$ of graph $G$, since $A_S$, $\Delta_{A_S}$, $\Delta_{D_S}$ are principal submatrices of $A$, $\Delta_A$, $\Delta_D$ respectively, then we have*

$$
\|A_S\|_2 \leq \|A\|_2 \leq \beta_A, \quad \|\Delta_{A_S}\|_2 \leq \|\Delta_A\|_2, \quad \|\Delta_{D_S}\|_2 \leq \|\Delta_D\|_2.
$$

**Notations:** For a subgraph $S$ of graph $G$, the self-connected adjacency matrix is $\hat{A}_S = I + A_S$, the degree matrix is $\hat{D}_S = \operatorname{diag}(\mathbf{1}^\top\hat{A}_S)$, and the normalized self-connected adjacency matrix is $U_S = \hat{D}_S^{-\frac{1}{2}}\hat{A}_S\hat{D}_S^{-\frac{1}{2}}$.

For a subgraph $\tilde{S}$ of graph $\tilde{G}$, we define some notations here. We denote the self-connected adjacency matrix as $\hat{A}'_{\tilde{S}} = I + \tilde{A}_{\tilde{S}}$, the diagonal matrix as $\hat{D}'_{\tilde{S}} = \operatorname{diag}(\mathbf{1}^\top\hat{A}'_{\tilde{S}})$, and the normalized self-connected adjacency matrix as $\tilde{U}_{\tilde{S}} = \hat{D}'^{-\frac{1}{2}}_{\tilde{S}}\hat{A}'_{\tilde{S}}\hat{D}'^{-\frac{1}{2}}_{\tilde{S}}$. We also denote $\Delta_{U_S} = \tilde{U}_{\tilde{S}} - U_S$ and $\Delta_{X_S^{(m,l)}} = \tilde{X}_{\tilde{S}}^{(m,l)} - X_S^{(m,l)}$.

**Lemma D.5** (Inequalities). *Base on Lemma D.4, for any subgraph $S$ of graph $G$, the inequalities in the Lemma D.2 still holds for $S$, shown as follows:*

$$\|\boldsymbol{U}_S\|_2 \le (1+\alpha)^{-1}(1+\beta_A)$$

$$\|\tilde{\boldsymbol{U}}_S\|_2 \le (1+\alpha+\kappa)^{-1}(1+\beta_A+\|\Delta_A\|_2)$$

$$\|\Delta_{U_S}\|_2 \le 2(1+\beta_A)(1+\alpha)^{-1}\|\Delta_D\|_2 + (1+\alpha+\kappa)^{-1}\|\Delta_A\|_2$$

$$\|\boldsymbol{X}_S^{(m,l)}\|_F \le \rho^l \beta_W^l \beta_X (1+\beta_A)^l (1+\alpha)^{-l}$$

$$\|\Delta_{X_S^{(m,l)}}\|_F \le \rho^l \beta_W^l (1+\beta_A+\|\Delta_A\|_2)^{l-1}(1+\alpha)^{-l}\left[(1+\beta_A+2\|\Delta_A\|_2)\|\Delta_X\|_F + 2l\beta_X(1+\beta_A)\|\Delta_D\|_2\right]$$

$$\tag{26}$$

*Proof.* The proof is mainly based on Lemma D.4.

Similar to (21), we have

$$\|\boldsymbol{U}_S\|_2 \le \|\hat{\boldsymbol{D}}_S^{-\frac{1}{2}}\|_2\|\hat{\boldsymbol{A}}_S\|_2\|\hat{\boldsymbol{D}}_S^{-\frac{1}{2}}\|_2 \le \|\hat{\boldsymbol{D}}^{-\frac{1}{2}}\|_2\|\hat{\boldsymbol{A}}\|_2\|\hat{\boldsymbol{D}}^{-\frac{1}{2}}\|_2 \le (1+\alpha)^{-1}(1+\beta_A). \tag{27}$$

Similar to (22), we have

$$\|\tilde{\boldsymbol{U}}_S\|_2 \le \|\hat{\boldsymbol{D}}_{\tilde{S}}'^{-\frac{1}{2}}\|_2\|\hat{\boldsymbol{A}}'\|_2\|\hat{\boldsymbol{D}}_{\tilde{S}}'^{-\frac{1}{2}}\|_2 \le \|\hat{\boldsymbol{D}}'^{-\frac{1}{2}}\|_2\|\hat{\boldsymbol{A}}'\|_2\|\hat{\boldsymbol{D}}'^{-\frac{1}{2}}\|_2$$
$$\le (1+\alpha+\kappa)^{-1}(1+\beta_A+\|\Delta_A\|_2). \tag{28}$$

Similar to (23), we have

$$\|\Delta_U\|_2 \le (\|\hat{\boldsymbol{D}}_S^{-\frac{1}{2}}\|_2 + \|\hat{\boldsymbol{D}}_{\tilde{S}}'^{-\frac{1}{2}}\|_2)\|\hat{\boldsymbol{A}}\|_2\|\hat{\boldsymbol{D}}_{\tilde{S}}'^{-\frac{1}{2}} - \hat{\boldsymbol{D}}_S^{-\frac{1}{2}}\|_2 + \|\hat{\boldsymbol{D}}_{\tilde{S}}'^{-\frac{1}{2}}\|\Delta_A\|_2\|\hat{\boldsymbol{D}}_{\tilde{S}}'^{-\frac{1}{2}}\|_2$$
$$\le (\|\hat{\boldsymbol{D}}^{-\frac{1}{2}}\|_2 + \|\hat{\boldsymbol{D}}'^{-\frac{1}{2}}\|_2)\|\hat{\boldsymbol{A}}\|_2\|\hat{\boldsymbol{D}}'^{-\frac{1}{2}} - \hat{\boldsymbol{D}}^{-\frac{1}{2}}\|_2 + \|\hat{\boldsymbol{D}}'^{-\frac{1}{2}}\|_2\|\Delta_A\|_2\|\hat{\boldsymbol{D}}'^{-\frac{1}{2}}\|_2 \tag{29}$$
$$\le 2(1+\beta_A)(1+\alpha)^{-1}\|\Delta_D\|_2 + (1+\alpha+\kappa)^{-1}\|\Delta_A\|_2$$

Similar to (24), we have

$$\|\boldsymbol{X}_S^{(m,l)}\|_F \le \rho\|\boldsymbol{U}_S\|_2\|\boldsymbol{X}_S^{(m,l-1)}\|_F\|\boldsymbol{W}^{(m,l)}\|_2$$
$$\le \rho\|\boldsymbol{U}\|_2\|\boldsymbol{X}^{(m,l-1)}\|_F\|\boldsymbol{W}^{(m,l)}\|_2 \tag{30}$$
$$\le \rho^l \beta_W^l \beta_X (1+\beta_A)^l (1+\alpha)^{-l}$$

Similar to (25), we have

$$\|\Delta_{X_S^{(m,l)}}\|_F \le \rho^2 \beta_W^2 \|\tilde{\boldsymbol{U}}_S\|_2^2\|\Delta_{X_S^{(l-2)}}\|_F + \rho^2\beta_W^2\|\tilde{\boldsymbol{U}}_S\|_2\|\Delta_{U_S}\|_2\|\boldsymbol{X}_S^{(l-2)}\|_F + \rho\beta_W\|\Delta_{U_S}\|_2\|\boldsymbol{X}_S^{(l-1)}\|_F$$
$$\le \rho^2\beta_W^2\|\tilde{\boldsymbol{U}}\|_2^2\|\Delta_{X^{(l-2)}}\|_F + \rho^2\beta_W^2\|\tilde{\boldsymbol{U}}\|_2\|\Delta_U\|_2\|\boldsymbol{X}^{(l-2)}\|_F + \rho\beta_W\|\Delta_U\|_2\|\boldsymbol{X}^{(l-1)}\|_F$$
$$\le \rho^l\beta_W^l(1+\beta_A+\|\Delta_A\|_2)^{l-1}(1+\alpha)^{-l}\left[(1+\beta_A+2\|\Delta_A\|_2)\|\Delta_X\|_F + 2l\beta_X(1+\beta_A)\|\Delta_D\|_2\right]$$
$$\tag{31}$$
$$\square$$

Finally, we can prove our theorem of robustness analysis in the main paper using Lemma D.5 as follows.

*Proof.* Given a pattern sampling set $\mathcal{S}^{(m)}$, we assume the $S^*$ satisfies

$$S^* = \underset{S \in \mathcal{S}^{(m)}}{\arg\max}\|\Delta_{X_S^{(m,L)}}\|_F.$$

Since the Lemma D.5 holds for any subgraph $S$, we have

$$\|\Delta_{X_{S^*}^{(m,l)}}\|_F \le \rho^l\beta_W^l(1+\beta_A+\|\Delta_A\|_2)^{l-1}(1+\alpha)^{-l}\left[(1+\beta_A+2\|\Delta_A\|_2)\|\Delta_X\|_F + 2l\beta_X(1+\beta_A)\|\Delta_D\|_2\right]$$

Then the upper bound of $\|\tilde{\boldsymbol{g}} - \boldsymbol{g}\|$ is given by

$$
\|\tilde{\boldsymbol{g}} - \boldsymbol{g}\| = \left\| \sum_{m=1}^{M} \lambda_m \left( \tilde{\boldsymbol{z}}^{(m)} - \boldsymbol{z}^{(m)} \right) \right\| \leq \sum_{m=1}^{M} \lambda_m \left\| \tilde{\boldsymbol{z}}^{(m)} - \boldsymbol{z}^{(m)} \right\|
$$

$$
= \frac{1}{Q} \sum_{m=1}^{M} \lambda_m \left\| \sum_{S \in \mathcal{S}^{(m)}} F(\tilde{\boldsymbol{A}}_S, \tilde{\boldsymbol{X}}_S; \mathcal{W}^{(m)}) - \sum_{S \in \mathcal{S}^{(m)}} F(\boldsymbol{A}_S, \boldsymbol{X}_S; \mathcal{W}^{(m)}) \right\|
$$

$$
\leq \frac{1}{Q} \sum_{m=1}^{M} \lambda_m \sum_{S \in \mathcal{S}^{(m)}} \left\| F(\tilde{\boldsymbol{A}}_S, \tilde{\boldsymbol{X}}_S; \mathcal{W}^{(m)}) - F(\boldsymbol{A}_S, \boldsymbol{X}_S; \mathcal{W}^{(m)}) \right\|
$$

$$
= \frac{1}{Q} \sum_{m=1}^{M} \lambda_m \sum_{S \in \mathcal{S}^{(m)}} \frac{1}{n} \left\| \boldsymbol{1}^{\top} (\tilde{\boldsymbol{X}}_S^{(m,L)} - \boldsymbol{X}_S^{(m,L)}) \right\|_F
$$

$$
\leq \frac{1}{Q} \sum_{m=1}^{M} \lambda_m \frac{1}{n} \sum_{S \in \mathcal{S}^{(m)}} \|\boldsymbol{1}\| \left\| \tilde{\boldsymbol{X}}_S^{(m,L)} - \boldsymbol{X}_S^{(m,L)} \right\|_F
$$

$$
= \frac{1}{Q\sqrt{n}} \sum_{m=1}^{M} \lambda_m \sum_{S \in \mathcal{S}^{(m)}} \left\| \Delta_{X_S^{(m,L)}} \right\|_F
$$

$$
\leq \frac{1}{Q\sqrt{n}} \sum_{m=1}^{M} \lambda_m \, Q \left\| \Delta_{X_{S^*}^{(m,L)}} \right\|_F
$$

$$
\leq \frac{1}{\sqrt{n}} \rho^l \beta_W^l (1 + \beta_A + \|\Delta_A\|_2)^{l-1} (1 + \alpha)^{-l} \left[ (1 + \beta_A + 2\|\Delta_A\|_2)\|\Delta_X\|_F + 2L\beta_X(1 + \beta_A)\|\Delta_D\|_2 \right]
$$

$$
\tag{32}
$$

$\square$

## E  PROOF FOR GENERALIZATION ANALYSIS OF SUPERVISED LOSS

Before providing our theorem, we need to provide the classification loss function $f_c$.

**Classification loss function $f_c$:**  We use a linear classifier with parameter $\boldsymbol{W}_C \in \mathbb{R}^{d \times C}$ and use softmax as the activation function as the classification function $f_c$, i.e., $\hat{\boldsymbol{y}} = \text{softmax}(\boldsymbol{g}\boldsymbol{W}_C)$. We suppose that $\|\boldsymbol{W}_C\|_2 \leq \beta_C$.

Then the classification loss is as follows

$$
\ell_{\text{CE}}(\boldsymbol{\lambda}, \mathbb{W}) = \text{cross-entropy}(\boldsymbol{y}, \hat{\boldsymbol{y}}) = \text{cross-entropy}(\boldsymbol{y}, \text{softmax}(\boldsymbol{g}\boldsymbol{W}_C)). \tag{33}
$$

To simplify the proof, we rewrite supervised loss $\ell_{\text{CE}}(\boldsymbol{\lambda}, \mathbb{W})$ function as

$$
\varphi(\boldsymbol{g}\boldsymbol{W}_C) := \text{cross-entropy}(\boldsymbol{y}, \hat{\boldsymbol{y}}) = \text{cross-entropy}(\boldsymbol{y}, \text{softmax}(\boldsymbol{g}\boldsymbol{W}_C)).
$$

**Lemma E.1.** *Let $\boldsymbol{v}$ be a vector, there exits a positive constant $\tau$ such that $\varphi(\boldsymbol{v})$ is a $\tau$-Lipschitz continuous function.*

**Generalization Error**  Let $\mathcal{D} := \{G_1, ..., G_{|\mathcal{D}|}\}$ be the training data. By removing the $i$-th graph of $\mathcal{D}$, we have $\mathcal{D}^{\backslash i} = \{G_1, ..., G_{i-1}, G_{i+1}, ..., G_{|\mathcal{D}|-1}\}$. Let $\boldsymbol{\lambda}_{\mathcal{D}}$ and $\bar{\mathbb{W}}_{\mathcal{D}} := \{\boldsymbol{W}_C, \boldsymbol{W}_{\mathcal{D}}^{(m,l)}, \forall m \in [M], l \in [L]\}$ be the parameters trained on $\mathcal{D}$. Let $\boldsymbol{\lambda}_{\mathcal{D}\backslash i}$ and $\bar{\mathbb{W}}_{\mathcal{D}\backslash i} := \{\boldsymbol{W}_{C\backslash i}, \boldsymbol{W}_{\mathcal{D}\backslash i}^{(m,l)}, \forall m \in [M], l \in [L]\}$ be the parameters trained on $\mathcal{D}^{\backslash i}$. Then our goal is to find a $\eta$ such that

$$
|\ell_{\text{CE}}(\boldsymbol{\lambda}_{\mathcal{D}}, \bar{\mathbb{W}}_{\mathcal{D}}; G) - \ell_{\text{CE}}(\boldsymbol{\lambda}_{\mathcal{D}\backslash i}, \bar{\mathbb{W}}_{\mathcal{D}\backslash i}; G)| \leq \eta \tag{34}
$$

**Theorem E.2.** *Given a graph $G$, let $\boldsymbol{g}$ be the graph representations learned with parameter $\boldsymbol{\lambda}_{\mathcal{D}}$ and $\bar{\mathbb{W}}_{\mathcal{D}}$ and $\boldsymbol{g}^{\backslash i}$ be the graph representations learned with parameter $\boldsymbol{\lambda}_{\mathcal{D}\backslash i}$ and $\bar{\mathbb{W}}_{\mathcal{D}\backslash i}$.*

*To simplify the proof, we denote that $\hat{\beta}_W = \max(\hat{\beta}_{W\mathcal{D}}, \hat{\beta}_{W\mathcal{D}\backslash i})$, where*

$$
\hat{\beta}_{W\mathcal{D}} = \max_{m \in [M], l \in [L]} \|W_{\mathcal{D}}^{(m,l)}\|_2, \text{ and } \hat{\beta}_{W\mathcal{D}\backslash i} = \max_{m \in [M], l \in [L]} \|W_{\mathcal{D}\backslash i}^{(m,l)}\|_2.
$$

*We also denote that*

$$\hat{\beta}_{\Delta W} = \max_{m \in [M], l \in [L]} \|\mathcal{W}_{\mathcal{D}}^{(m,l)} - \mathcal{W}_{\mathcal{D}^{\backslash i}}^{(m,l)}\|_2.$$

*Then we have*

$$\eta = \frac{\tau}{\sqrt{n}} \rho^L \hat{\beta}_W^{L-1} \beta_X (1+\beta_A)^L (1+\alpha)^{-L} \left[ \hat{\beta}_W \|\boldsymbol{W}_C - \boldsymbol{W}_{C^{\backslash i}}\|_2 + \|\boldsymbol{W}_{C^{\backslash i}}\|_2 \left( \hat{\beta}_W \|\boldsymbol{\lambda}_{\mathcal{D}} - \boldsymbol{\lambda}_{\mathcal{D}^{\backslash i}}\| + L\hat{\beta}_{\Delta W} \|\boldsymbol{\lambda}_{\mathcal{D}^{\backslash i}}\| \right) \right]$$

*Proof.* We provide two lemmas used in our proof

**Lemma E.3.** $\|\boldsymbol{g}\| \leq \frac{1}{\sqrt{n}} \rho^L \hat{\beta}_W^L \beta_X (1+\beta_A)^L (1+\alpha)^{-L}$

**Lemma E.4.**

$$\|\boldsymbol{g} - \boldsymbol{g}^{\backslash i}\| \leq \frac{1}{\sqrt{n}} \rho^L \hat{\beta}_W^{L-1} \beta_X (1+\beta_A)^L (1+\alpha)^{-L} \left( \hat{\beta}_W \|\boldsymbol{\lambda}_{\mathcal{D}} - \boldsymbol{\lambda}_{\mathcal{D}^{\backslash i}}\| + L\hat{\beta}_{\Delta W} \|\boldsymbol{\lambda}_{\mathcal{D}^{\backslash i}}\| \right)$$

**The main proof of our Theorem**

$$|\ell_{\text{CE}}(\boldsymbol{\lambda}_{\mathcal{D}}, \bar{\mathbb{W}}_{\mathcal{D}}; G) - \ell_{\text{CE}}(\boldsymbol{\lambda}_{\mathcal{D}^{\backslash i}}, \bar{\mathbb{W}}_{\mathcal{D}^{\backslash i}}; G)| = \|\varphi(\boldsymbol{g}^{\backslash i} \boldsymbol{W}_{C^{\backslash i}}) - \varphi(\boldsymbol{g} \boldsymbol{W}_C)\|$$

$$\leq \tau \|\boldsymbol{g} \boldsymbol{W}_C - \boldsymbol{g}^{\backslash i} \boldsymbol{W}_{C^{\backslash i}}\|$$

$$= \tau \|\boldsymbol{g} \boldsymbol{W}_C - \boldsymbol{g} \boldsymbol{W}_{C^{\backslash i}} + \boldsymbol{g} \boldsymbol{W}_{C^{\backslash i}} - \boldsymbol{g}^{\backslash i} \boldsymbol{W}_{C^{\backslash i}}\|$$

$$\leq \tau \|\boldsymbol{g}\| \|\boldsymbol{W}_C - \boldsymbol{W}_{C^{\backslash i}}\|_2 + \tau \|\boldsymbol{g} - \boldsymbol{g}^{\backslash i}\| \|\boldsymbol{W}_{C^{\backslash i}}\|_2$$

$$\leq \tau \|\boldsymbol{W}_C - \boldsymbol{W}_{C^{\backslash i}}\|_2 \frac{1}{\sqrt{n}} \rho^L \hat{\beta}_W^L \beta_X (1+\beta_A)^L (1+\alpha)^{-L}$$

$$+ \tau \|\boldsymbol{W}_{C^{\backslash i}}\|_2 \frac{1}{\sqrt{n}} \rho^L \hat{\beta}_W^{L-1} \beta_X (1+\beta_A)^L (1+\alpha)^{-L} \left( \hat{\beta}_W \|\boldsymbol{\lambda}_{\mathcal{D}} - \boldsymbol{\lambda}_{\mathcal{D}^{\backslash i}}\| + L\hat{\beta}_{\Delta W} \|\boldsymbol{\lambda}_{\mathcal{D}^{\backslash i}}\| \right)$$

$$= \frac{\tau}{\sqrt{n}} \rho^L \hat{\beta}_W^{L-1} \beta_X (1+\beta_A)^L (1+\alpha)^{-L} \left[ \hat{\beta}_W \|\boldsymbol{W}_C - \boldsymbol{W}_{C^{\backslash i}}\|_2 + \|\boldsymbol{W}_{C^{\backslash i}}\|_2 \left( \hat{\beta}_W \|\boldsymbol{\lambda}_{\mathcal{D}} - \boldsymbol{\lambda}_{\mathcal{D}^{\backslash i}}\| + L\hat{\beta}_{\Delta W} \|\boldsymbol{\lambda}_{\mathcal{D}^{\backslash i}}\| \right) \right]$$

$$\tag{35}$$

Since $\sum_{i=1}^{M} \lambda_i \leq 1$ and $\lambda_i \geq 0$, we have $\|\boldsymbol{\lambda}\| \leq 1$ and $\|\boldsymbol{\lambda} - \boldsymbol{\lambda}_{\mathcal{D}^{\backslash i}}\| \leq 2$. This finished the proof. $\square$

### E.1 Proof for Lemmas

**Lemma E.5.** *Let $\boldsymbol{v}$ be a vector, there exits a positive constant $\tau$ such that $\varphi(\boldsymbol{v})$ is a $\tau$-Lipschitz continuous function.*

*Proof.* **Step 1: Softmax is Lipschitz** The softmax function is known to be Lipschitz continuous. Specifically, there exists a constant $K$ such that:

$$\|\text{softmax}(v) - \text{softmax}(w)\|_1 \leq L_1 \|v - w\|_2,$$

where $\|\cdot\|_1$ is the $\ell_1$-norm and $\|\cdot\|_2$ is the $\ell_2$-norm. For the $\ell_1$-norm, $L_1$ can be bounded by 1, but generally, for different norms, the exact Lipschitz constant might vary.

**Step 2: Cross-Entropy is Lipschitz on the Simplex** Given $\mathbf{q} = \text{softmax}(v)$ and $\mathbf{r} = \text{softmax}(w)$, we need to check the Lipschitz continuity of the cross-entropy loss function with respect to these distributions:

$$|\text{cross-entropy}(\mathbf{p}, \mathbf{q}) - \text{cross-entropy}(\mathbf{p}, \mathbf{r})| \leq L_2 \|\mathbf{q} - \mathbf{r}\|.$$

The cross-entropy loss is a convex function and it is smooth with respect to the probability distributions $\mathbf{q}$ and $\mathbf{r}$. Given the boundedness of the probability values (since $\mathbf{q}$ and $\mathbf{r}$ lie in the probability simplex), the gradient of the cross-entropy loss is also bounded.

**Combining Steps** Since both the softmax function and the cross-entropy loss function are Lipschitz continuous, their composition will also be Lipschitz continuous. Therefore, there exists a constant $\tau = L_1 L_2$ such that:

$$|\varphi(v) - \varphi(w)| \leq \tau \|v - w\|.$$

Hence, $\varphi(v) = \text{cross-entropy}(\text{softmax}(v))$ is $\tau$-Lipschitz continuous. $\square$

**Lemma E.6.** $\|\boldsymbol{g}\| \leq \frac{1}{\sqrt{n}} \rho^L \hat{\beta}_W^L \beta_X (1 + \beta_A)^L (1 + \alpha)^{-L}$

*Proof.* Given a pattern sampling set $\mathcal{S}^{(m)}$, we assume the $S^*$ satisfies

$$S^* = \arg\max_{S \in \mathcal{S}^{(m)}} \|\boldsymbol{X}_S^{(L)}\|_F.$$

Since the Lemma D.5 holds for any subgraph $S$, then we have

$$\|\boldsymbol{X}_{S^*}^{(m,l)}\|_F \leq \rho^l \hat{\beta}_W^l \beta_X (1 + \beta_A)^l (1 + \alpha)^{-l}.$$

Then, we have

$$
\begin{aligned}
\|\boldsymbol{g}\| = \|\sum_{m=1}^M \lambda_m \, \boldsymbol{z}^{(m)}\| &\leq \sum_{m=1}^M \lambda_m \|\boldsymbol{z}^{(m)}\| \\
&= \frac{1}{Q} \sum_{m=1}^M \lambda_m \|\sum_{S \in \mathcal{S}^{(m)}} F(\boldsymbol{A}_S, \boldsymbol{X}_S; \mathcal{W}^{(m)})\| \\
&\leq \frac{1}{Q} \sum_{m=1}^M \lambda_m \sum_{S \in \mathcal{S}^{(m)}} \|F(\boldsymbol{A}_S, \boldsymbol{X}_S; \mathcal{W}^{(m)})\| \\
&= \frac{1}{Q} \sum_{m=1}^M \lambda_m \sum_{S \in \mathcal{S}^{(m)}} \frac{1}{n} \|\boldsymbol{1}^\top (\boldsymbol{X}_S^{(m,L)})\|_F \\
&\leq \frac{1}{Q} \sum_{m=1}^M \lambda_m \frac{1}{n} \sum_{S \in \mathcal{S}^{(m)}} \|\boldsymbol{1}\|_2 \|\boldsymbol{X}_S^{(m,L)}\|_F \\
&= \frac{1}{Q\sqrt{n}} \sum_{m=1}^M \lambda_m \sum_{S \in \mathcal{S}^{(m)}} \|\boldsymbol{X}_S^{(m,L)}\|_F \\
&\leq \frac{1}{\sqrt{n}} \sum_{m=1}^M \lambda_m \|\boldsymbol{X}_{S^*}^{(m,L)}\|_F \\
&\leq \frac{1}{\sqrt{n}} \rho^L \hat{\beta}_W^L \beta_X (1 + \beta_A)^L (1 + \alpha)^{-L}
\end{aligned}
\tag{36}
$$

$\square$

**Lemma E.7.**

$$\|\boldsymbol{g} - \boldsymbol{g}^{\backslash i}\| \leq \frac{1}{\sqrt{n}} \rho^L \hat{\beta}_W^{L-1} \beta_X (1 + \beta_A)^L (1 + \alpha)^{-L} \left( \hat{\beta}_W \|\boldsymbol{\lambda}_{\mathcal{D}} - \boldsymbol{\lambda}_{\mathcal{D}\backslash i}\| + L\hat{\beta}_{\Delta W} \|\boldsymbol{\lambda}_{\mathcal{D}\backslash i}\| \right)$$

*Proof.* To simplify the proof, we denote

$$
\begin{aligned}
\hat{\beta}_W &= \max\{\max_{m \in [M], l \in [L]} \|\boldsymbol{W}_{\mathcal{D}}^{(m,l)}\|_2, \max_{m \in [M], l \in [L]} \|\boldsymbol{W}_{\mathcal{D}\backslash i}^{(m,l)}\|_2\} \\
\hat{\beta}_{\Delta W} &= \max_{m \in [M], l \in [L]} \|\mathcal{W}_{\mathcal{D}}^{(m,l)} - \mathcal{W}_{\mathcal{D}\backslash i}^{(m,l)}\|_2.
\end{aligned}
\tag{37}
$$

Let $\boldsymbol{X}_{S\mathcal{D}}^{(m,l)}$ be the embedding features of the $l$-th layer GCN with the parameter $\mathcal{W}_{\mathcal{D}}^{(m)}$ learned from dataset $\mathcal{D}$. Let $\boldsymbol{X}_{S\mathcal{D}\backslash i}^{(m,l)}$ be the embedding features of the $l$-th layer GCN with the parameter $\mathcal{W}_{\mathcal{D}\backslash i}^{(m)}$ learned from dataset $\mathcal{D}^{\backslash i}$.

We denote $\boldsymbol{Z}_{\mathcal{D}} = [\boldsymbol{z}_{\mathcal{D}}^{(1)}, ..., \boldsymbol{z}_{\mathcal{D}}^{(m)}]^\top$ and $\boldsymbol{Z}_{\mathcal{D}\backslash i} = [\boldsymbol{z}_{\mathcal{D}\backslash i}^{(1)}, ..., \boldsymbol{z}_{\mathcal{D}\backslash i}^{(m)}]^\top$. Let

$$q_1 = \arg\max_{m \in [M]} \|\boldsymbol{z}_{\mathcal{D}}^{(m)}\|, \quad q_2 = \arg\max_{m \in [M]} \|\boldsymbol{z}_{\mathcal{D}}^{(l_2)} - \boldsymbol{z}_{\mathcal{D}\backslash i}^{(l_2)}\|.$$

Then we have

$$\|\boldsymbol{Z}_{\mathcal{D}}\|_2 \leq \|\boldsymbol{z}_{\mathcal{D}}^{(q_1)}\|, \quad \|\boldsymbol{Z}_{\mathcal{D}} - \boldsymbol{Z}_{\mathcal{D}\backslash i}\|_2 \leq \|\boldsymbol{z}_{\mathcal{D}}^{(q_2)} - \boldsymbol{z}_{\mathcal{D}\backslash i}^{(q_2)}\|.$$

Similar to (36), we have

$$\|\boldsymbol{z}_{\mathcal{D}}^{(q_1)}\| \le \frac{1}{\sqrt{n}} \rho^L \hat{\beta}_W^L \beta_X (1 + \beta_A)^L (1 + \alpha)^{-L} \tag{38}$$

Denote $\Delta_{X_{S\mathcal{D}}^{(q_2,l)}} := \boldsymbol{X}_{S\mathcal{D}}^{(q_2,l)} - \boldsymbol{X}_{S\mathcal{D}\backslash i}^{(q_2,l)}$, then, similar to inequality (25) we have

$$
\begin{aligned}
\|\Delta_{X_{S\mathcal{D}}^{(q_2,l)}}\|_F &= \|\sigma(\boldsymbol{U}_S \boldsymbol{X}_{S\mathcal{D}}^{(q_2,l-1)} \mathcal{W}_{\mathcal{D}}^{(q_2)}) - \sigma(\boldsymbol{U}_S \boldsymbol{X}_{S\mathcal{D}\backslash i}^{(q_2,l)} \mathcal{W}_{\mathcal{D}\backslash i}^{(q_2)})\|_F \\
&\le \rho \|\boldsymbol{U}_S\|_2 \|\boldsymbol{X}_{S\mathcal{D}}^{(q_2,l-1)} W_{\mathcal{D}}^{(q_2,l-1)} - \boldsymbol{X}_{S\mathcal{D}\backslash i}^{(q_2,l-1)} W_{\mathcal{D}\backslash i}^{(q_2,l-1)}\|_F \\
&\le \rho \|\boldsymbol{U}_S\|_2 \|\boldsymbol{X}_{S\mathcal{D}}^{(q_2,l-1)} W_{\mathcal{D}}^{(q_2,l-1)} - \boldsymbol{X}_{S\mathcal{D}}^{(q_2,l-1)} W_{\mathcal{D}\backslash i}^{(q_2,l-1)} + \boldsymbol{X}_{S\mathcal{D}}^{(q_2,l-1)} W_{\mathcal{D}\backslash i}^{(q_2,l-1)} - \boldsymbol{X}_{S\mathcal{D}\backslash i}^{(q_2,l-1)} W_{\mathcal{D}\backslash i}^{(q_2,l-1)}\|_F \\
&\le \rho \|\boldsymbol{U}_S\|_2 \|\boldsymbol{X}_{S\mathcal{D}}^{(q_2,l-1)} (W_{\mathcal{D}}^{(q_2,l-1)} - W_{\mathcal{D}\backslash i}^{(q_2,l-1)}) + (\boldsymbol{X}_{S\mathcal{D}}^{(q_2,l-1)} - \boldsymbol{X}_{S\mathcal{D}\backslash i}^{(q_2,l-1)}) W_{\mathcal{D}\backslash i}^{(q_2,l-1)}\|_F \\
&\le \rho \|\boldsymbol{U}_S\|_2 (\|\boldsymbol{X}_{S\mathcal{D}}^{(q_2,l-1)}\|_F \|W_{\mathcal{D}}^{(q_2,l-1)} - W_{\mathcal{D}\backslash i}^{(q_2,l-1)}\|_2 + \|\boldsymbol{X}_{S\mathcal{D}}^{(q_2,l-1)} - \boldsymbol{X}_{S\mathcal{D}\backslash i}^{(q_2,l-1)}\|_F \|W_{\mathcal{D}\backslash i}^{(q_2,l-1)}\|_2) \\
&= \rho \|\boldsymbol{U}_S\|_2 \hat{\beta}_W \|\Delta_{X_{S\mathcal{D}}^{(q_2,l-1)}}\|_F + \rho \|\boldsymbol{U}_S\|_2 \hat{\beta}_{\Delta W} \|\boldsymbol{X}_{S\mathcal{D}}^{(q_2,l-1)}\|_F \\
&\le \rho^l \|\boldsymbol{U}_S\|_2^l \hat{\beta}_W^l \|\Delta_{X_{S\mathcal{D}}^{(q_2,0)}}\|_F + \sum_{k=1}^{l} \rho^k \|\boldsymbol{U}_S\|_2^k \hat{\beta}_W^{k-1} \hat{\beta}_{\Delta W} \|\boldsymbol{X}_{S\mathcal{D}}^{(q_2,l-k)}\|_F
\end{aligned}
\tag{39}
$$

where $\|\Delta_{X_{S\mathcal{D}}^{(q_2,0)}}\|_F = \|\boldsymbol{X}_S - \boldsymbol{X}_S\|_F = 0$. We can directly use the inequality (24), such that

$$\|\boldsymbol{X}_{S\mathcal{D}}^{(m,l)}\|_F \le \rho^l \hat{\beta}_W^l \beta_X (1 + \beta_A)^l (1 + \alpha)^{-l} \tag{40}$$

Thus, we continue the proof

$$
\begin{aligned}
\|\Delta_{X_{S\mathcal{D}}^{(q_2,l)}}\|_F &\le \rho^l \|\boldsymbol{U}_S\|_2^l \hat{\beta}_W^l \|\Delta_{X_{S\mathcal{D}}^{(q_2,0)}}\|_F + \sum_{k=1}^{l} \rho^k \|\boldsymbol{U}_S\|_2^k \hat{\beta}_W^{k-1} \hat{\beta}_{\Delta W} \|\boldsymbol{X}_{S\mathcal{D}}^{(q_2,l-k)}\|_F \\
&\le l \rho^l (1 + \alpha)^{-l} (1 + \beta_A)^l \hat{\beta}_W^{l-1} \hat{\beta}_{\Delta W} \beta_X
\end{aligned}
\tag{41}
$$

Also similar to (D.5), we have

$$
\begin{aligned}
\|\boldsymbol{z}_{\mathcal{D}}^{(q_2)} - \boldsymbol{z}_{\mathcal{D}\backslash i}^{(q_2)}\| &= \|F(\boldsymbol{A}_S, \boldsymbol{X}_S; \mathcal{W}_{\mathcal{D}}^{(q_2)}) - F(\boldsymbol{A}_S, \boldsymbol{X}_S; \mathcal{W}_{\mathcal{D}\backslash i}^{(q_2)})\| \\
&= \frac{1}{n} \|\mathbf{1}^\top (\boldsymbol{X}_{S\mathcal{D}}^{(q_2,L)}) - \mathbf{1}^\top (\boldsymbol{X}_{S\mathcal{D}\backslash i}^{(q_2,L)})\| \\
&= \frac{1}{\sqrt{n}} \|\boldsymbol{X}_{S\mathcal{D}}^{(q_2,L)} - \boldsymbol{X}_{S\mathcal{D}\backslash i}^{(q_2,L)}\|_F = \frac{1}{\sqrt{n}} \|\Delta_{X_{S\mathcal{D}}^{(q_2,L)}}\|_F \\
&\le \frac{L}{\sqrt{n}} \rho^L (1 + \alpha)^{-L} (1 + \beta_A)^L \hat{\beta}_W^{L-1} \hat{\beta}_{\Delta W} \beta_X
\end{aligned}
\tag{42}
$$

Finally, we have

$$
\begin{aligned}
\|\boldsymbol{g} - \boldsymbol{g}^{\backslash i}\| &= \|\boldsymbol{\lambda}_{\mathcal{D}}^\top \boldsymbol{Z}_{\mathcal{D}} - \boldsymbol{\lambda}_{\mathcal{D}\backslash i}^\top \boldsymbol{Z}_{\mathcal{D}\backslash i}\| \\
&= \|\boldsymbol{\lambda}_{\mathcal{D}}^\top \boldsymbol{Z}_{\mathcal{D}} - \boldsymbol{\lambda}_{\mathcal{D}\backslash i}^\top \boldsymbol{Z}_{\mathcal{D}} + \boldsymbol{\lambda}_{\mathcal{D}\backslash i}^\top \boldsymbol{Z}_{\mathcal{D}} - \boldsymbol{\lambda}_{\mathcal{D}\backslash i}^\top \boldsymbol{Z}_{\mathcal{D}\backslash i}\| \\
&= \|(\boldsymbol{\lambda}_{\mathcal{D}} - \boldsymbol{\lambda}_{\mathcal{D}\backslash i})^\top \boldsymbol{Z}_{\mathcal{D}} + \boldsymbol{\lambda}_{\mathcal{D}\backslash i}^\top (\boldsymbol{Z}_{\mathcal{D}} - \boldsymbol{Z}_{\mathcal{D}\backslash i})\| \\
&\le \|\boldsymbol{\lambda}_{\mathcal{D}} - \boldsymbol{\lambda}_{\mathcal{D}\backslash i}\| \|\boldsymbol{Z}_{\mathcal{D}}\|_2 + \|\boldsymbol{\lambda}_{\mathcal{D}\backslash i}\| \|\boldsymbol{Z}_{\mathcal{D}} - \boldsymbol{Z}_{\mathcal{D}\backslash i}\|_2 \\
&\le \|\boldsymbol{\lambda}_{\mathcal{D}} - \boldsymbol{\lambda}_{\mathcal{D}\backslash i}\| \|\boldsymbol{z}_{\mathcal{D}}^{(q_1)}\| + \|\boldsymbol{\lambda}_{\mathcal{D}\backslash i}\| \|\boldsymbol{z}_{\mathcal{D}}^{(q_2)} - \boldsymbol{z}_{\mathcal{D}\backslash i}^{(q_2)}\| \\
&\le \|\boldsymbol{\lambda}_{\mathcal{D}} - \boldsymbol{\lambda}_{\mathcal{D}\backslash i}\| \frac{1}{\sqrt{n}} \rho^L \hat{\beta}_W^L \beta_X (1 + \beta_A)^L (1 + \alpha)^{-L} \\
&\quad + \|\boldsymbol{\lambda}_{\mathcal{D}\backslash i}\| \frac{L}{\sqrt{n}} \rho^L (1 + \alpha)^{-L} (1 + \beta_A)^L \hat{\beta}_W^{L-1} \hat{\beta}_{\Delta W} \beta_X \\
&= \frac{1}{\sqrt{n}} \rho^L \hat{\beta}_W^{L-1} \beta_X (1 + \beta_A)^L (1 + \alpha)^{-L} \left(\hat{\beta}_W \|\boldsymbol{\lambda}_{\mathcal{D}} - \boldsymbol{\lambda}_{\mathcal{D}\backslash i}\| + L \hat{\beta}_{\Delta W} \|\boldsymbol{\lambda}_{\mathcal{D}\backslash i}\|\right)
\end{aligned}
\tag{43}
$$

$\square$

# F  MORE EXPERIMENTAL RESULTS

In this section, we present additional experiments and supplementary figures.

## F.1  EVALUATING THE ENSEMBLE KERNEL (PXGL-EGK)

Here, we compare our ensemble kernel (PXGL-EGK) as defined in Definition 3.3 with individual kernels $K_{\mathcal{P}}$. We report the results as follows. Specifically, we use three pattern counting kernels in the ensemble method: Random Walk (RW) kernels (Borgwardt et al., 2005; Gärtner et al., 2003), Sub-tree kernels (Da San Martino et al., 2012; Smola & Vishwanathan, 2002), and Graphlet kernels (Pržulj, 2007). Since graph kernels are unsupervised learning methods, we compare the clustering accuracy and Normalized Mutual Information (NMI) of each kernel, as shown in Table 6. The result shows that PXGL-EGK outperform each individual kernels it used.

Table 6: ACC and NMI of Graph Clustering. The best ACC is **bold** and the best NMI is green.

| Method | Metric | MUTAG | PROTEINS | DD | IMDB-B |
|---|---|---|---|---|---|
| RW | ACC | $0.743 \pm 0.052$ | $0.712 \pm 0.021$ | $0.516 \pm 0.015$ | $0.658 \pm 0.014$ |
| | NMI | $0.238 \pm 0.016$ | $0.268 \pm 0.016$ | $0.187 \pm 0.002$ | $0.266 \pm 0.019$ |
| Sub-tree | ACC | $0.729 \pm 0.013$ | $0.692 \pm 0.027$ | $0.542 \pm 0.016$ | $0.612 \pm 0.018$ |
| | NMI | $0.195 \pm 0.047$ | $0.151 \pm 0.028$ | $0.229 \pm 0.015$ | $0.242 \pm 0.013$ |
| Graphlet | ACC | $0.735 \pm 0.026$ | $0.636 \pm 0.017$ | $0.568 \pm 0.013$ | $0.614 \pm 0.012$ |
| | NMI | $0.214 \pm 0.019$ | $0.154 \pm 0.026$ | $0.285 \pm 0.011$ | $0.214 \pm 0.025$ |
| PXGL-EGK | ACC | $\mathbf{0.761 \pm 0.025}$ | $\mathbf{0.721 \pm 0.028}$ | $\mathbf{0.572 \pm 0.025}$ | $\mathbf{0.672 \pm 0.023}$ |
| | NMI | $0.328 \pm 0.046$ | $0.321 \pm 0.019$ | $0.296 \pm 0.013$ | $0.310 \pm 0.021$ |

## F.2  SENSITIVITY ANALYSIS

**Sensitivity of PXGL-GNN to $Q$**   Here we use the MUTAG dataset to show the sensitivity of accuracy and time cost to the number of samples $Q$ for each pattern. We see that the time cost is roughly linear with $Q$ and the accuracy is not sensitive to $Q$ when it is larger than 5.

Table 7: Impact of sampling number Q on MUTAG dataset (20 epochs, 7 patterns)

| Q | 3 | 5 | 7 | 10 | 15 |
|---|---|---|---|---|---|
| Accuracy (%) | $87.63 \pm 1.42$ | $94.87 \pm 2.26$ | $94.26 \pm 2.17$ | $95.35 \pm 1.89$ | $95.33 \pm 2.48$ |
| Training Time (s) | 636s | 877s | 1035s | 1563s | 2351s |

**Sensitivity of PXGL-GNN to $L$**   In the following table, we use three datasets to show the accuracy of the graph classification of our PXGL-EGK model with different number of layers $L$. The results

Table 8: Impact of the number of layers of GNN

| **Model** | L = 1 | L = 3 | L = 5 | L = 7 | L = 9 |
|---|---|---|---|---|---|
| MUTAG | $81.44 \pm 1.29$ | $86.73 \pm 2.78$ | $94.87 \pm 2.26$ | $91.25 \pm 1.14$ | $89.66 \pm 1.15$ |
| PROTEINS | $62.17 \pm 1.53$ | $67.22 \pm 1.16$ | $78.23 \pm 2.46$ | $73.21 \pm 1.98$ | $71.07 \pm 1.63$ |
| DD | $75.36 \pm 1.21$ | $79.35 \pm 1.20$ | $86.54 \pm 1.95$ | $82.36 \pm 1.24$ | $82.17 \pm 1.54$ |

reveal that the model performs best at $L = 5$. With fewer layers, the model lacks sufficient capacity for representation; with more layers, the model is too complex and has overfitting performances. This is consistent with our theoretical analysis, since when the model is complex the gap between training error and the testing error becomes large.

**Sensitivity of PXGL-GNN to pattern combination**   The following table shows the classification accuracy given by PXGL-GNN with different combinations of patterns on the MUTAG dataset. We see that by including more patterns, the classification accuracy tends to be higher.

Table 9: Classification accuracy of PXGL-GNN with different pattern combinations on MUTAG dataset. The best performance is shown in **bold**.

| Pattern Combinations | Accuracy (%) | $\lambda$ weights |
|---|---|---|
| Paths only | 80.47 ± 1.24 | 1.0 |
| Trees only | 86.39 ± 2.73 | 1.0 |
| Cycles only | 89.24 ± 1.76 | 1.0 |
| Paths + Trees | 87.11 ± 2.93 | 0.274 / 0.716 |
| Paths + Cycles | 91.62 ± 1.14 | 0.207 / 0.793 |
| Trees + Cycles | 92.31 ± 2.65 | 0.325 / 0.675 |
| **All Patterns** | 94.87 ± 2.26 | 0.095/0.046/0.654 |

## F.3 SUPERVISED LEARNING

In this section, we provide the figures to visualize weight vector $\boldsymbol{\lambda}$, graph representation $\boldsymbol{g}$ and pattern representations $\boldsymbol{z}^{(m)}$ learned by solving the supervised loss (11).

## F.4 UNSUPERVISED LEARNING

In this section, we provide the figures to visualize weight vector $\boldsymbol{\lambda}$, graph representation $\boldsymbol{g}$ and pattern representations $\boldsymbol{z}^{(m)}$ learned by solving the unsupervised loss (10).

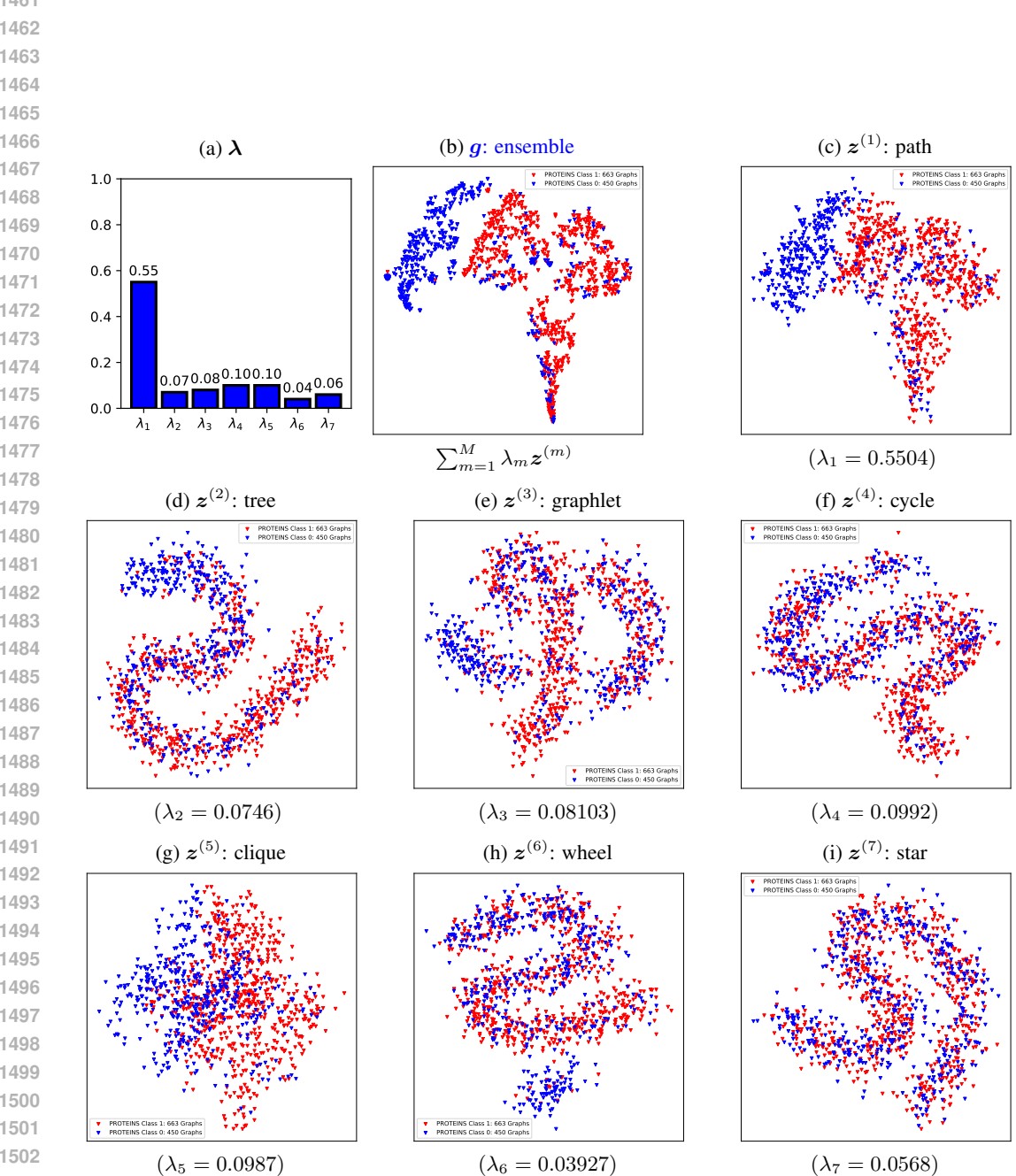

Figure 5: t-SNE visualizations of GNNs' pattern representations (supervised) for the dataset PROTEINS.

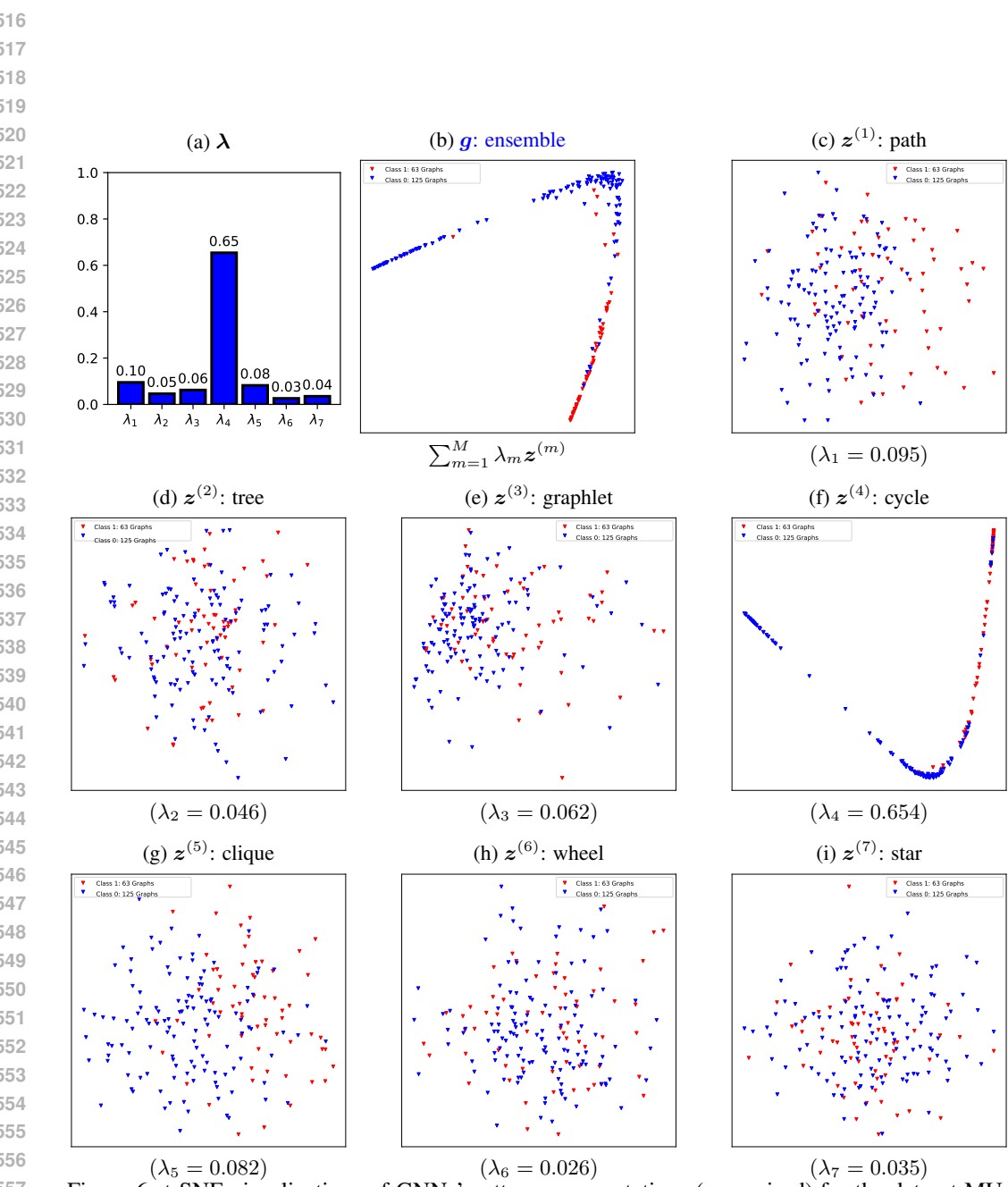

Figure 6: t-SNE visualizations of GNNs' pattern representations (supervised) for the dataset MU-TAG.

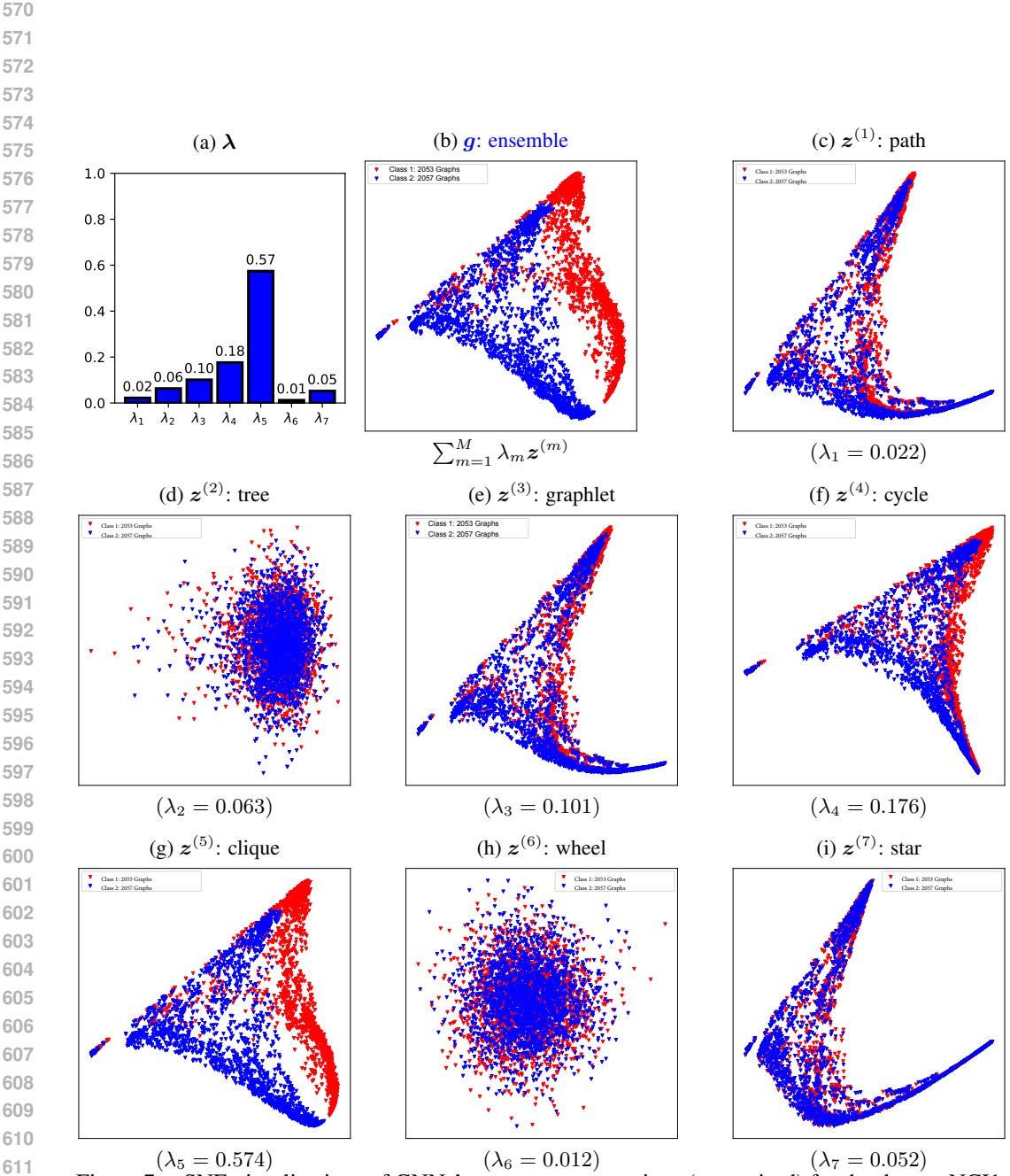

Figure 7: t-SNE visualizations of GNNs' pattern representations (supervised) for the dataset NCI1.

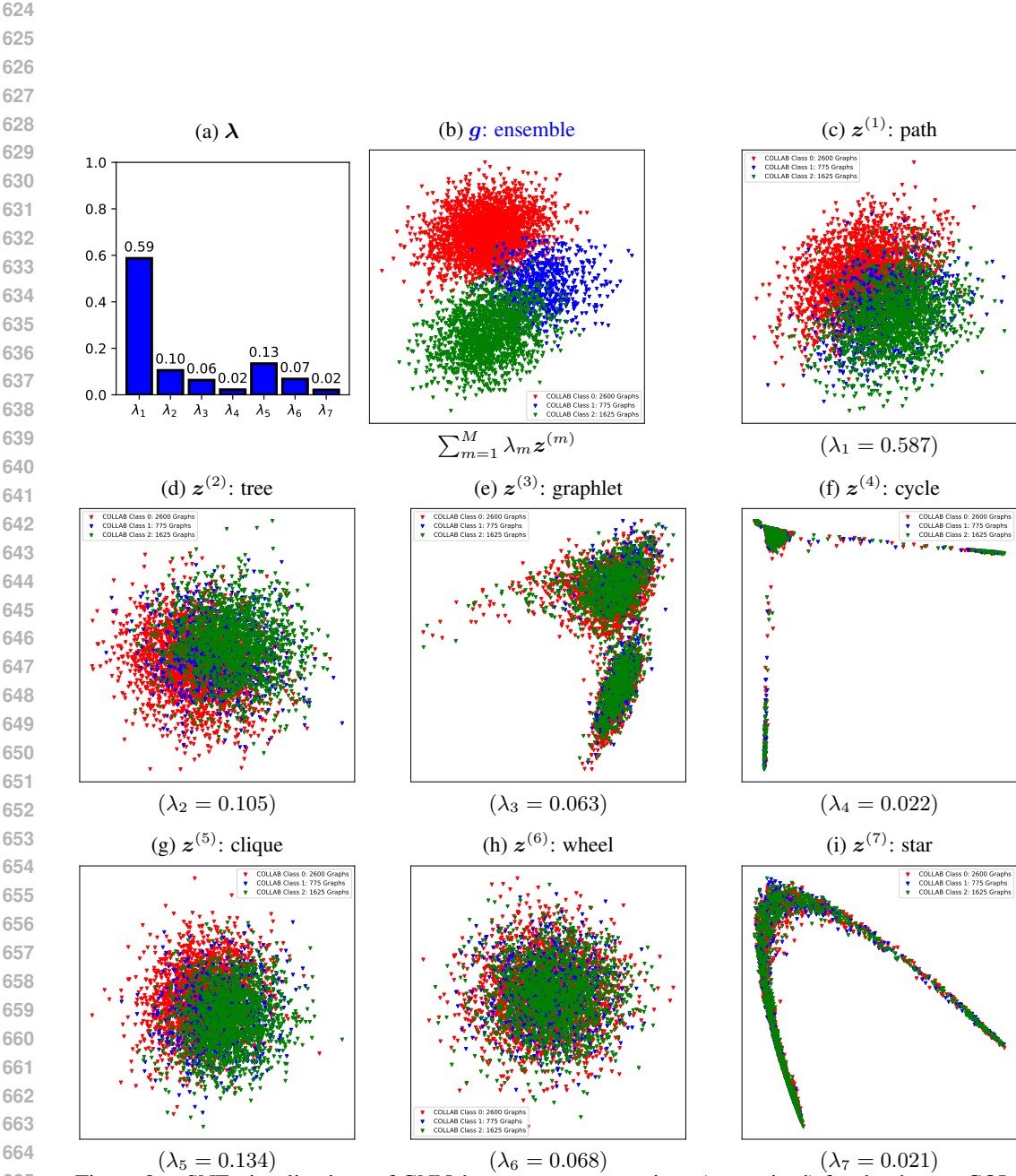

Figure 8: t-SNE visualizations of GNNs' pattern representations (supervised) for the dataset COL-LAB.

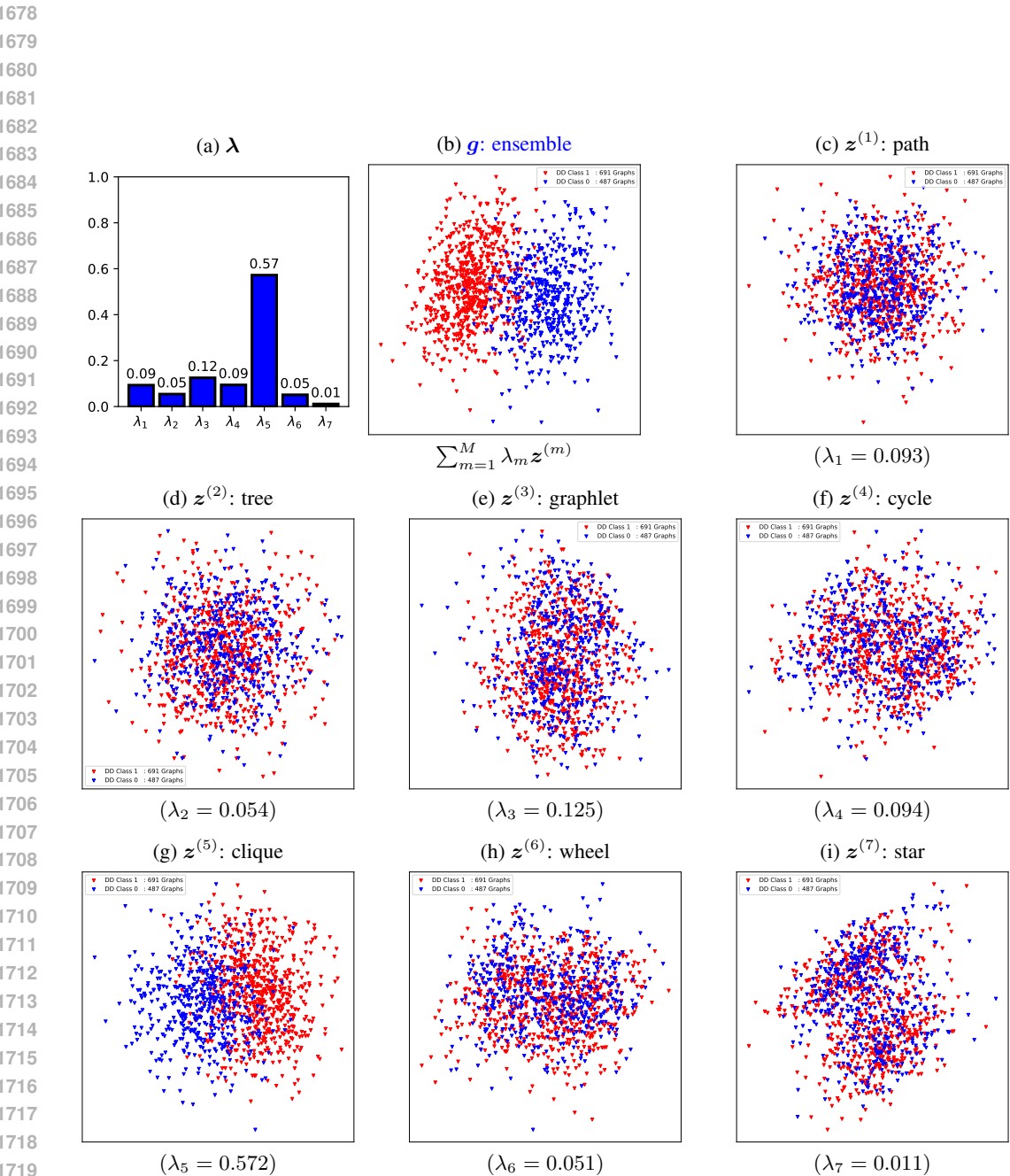

Figure 9: t-SNE visualizations of GNNs' pattern representations (supervised) for the dataset DD.

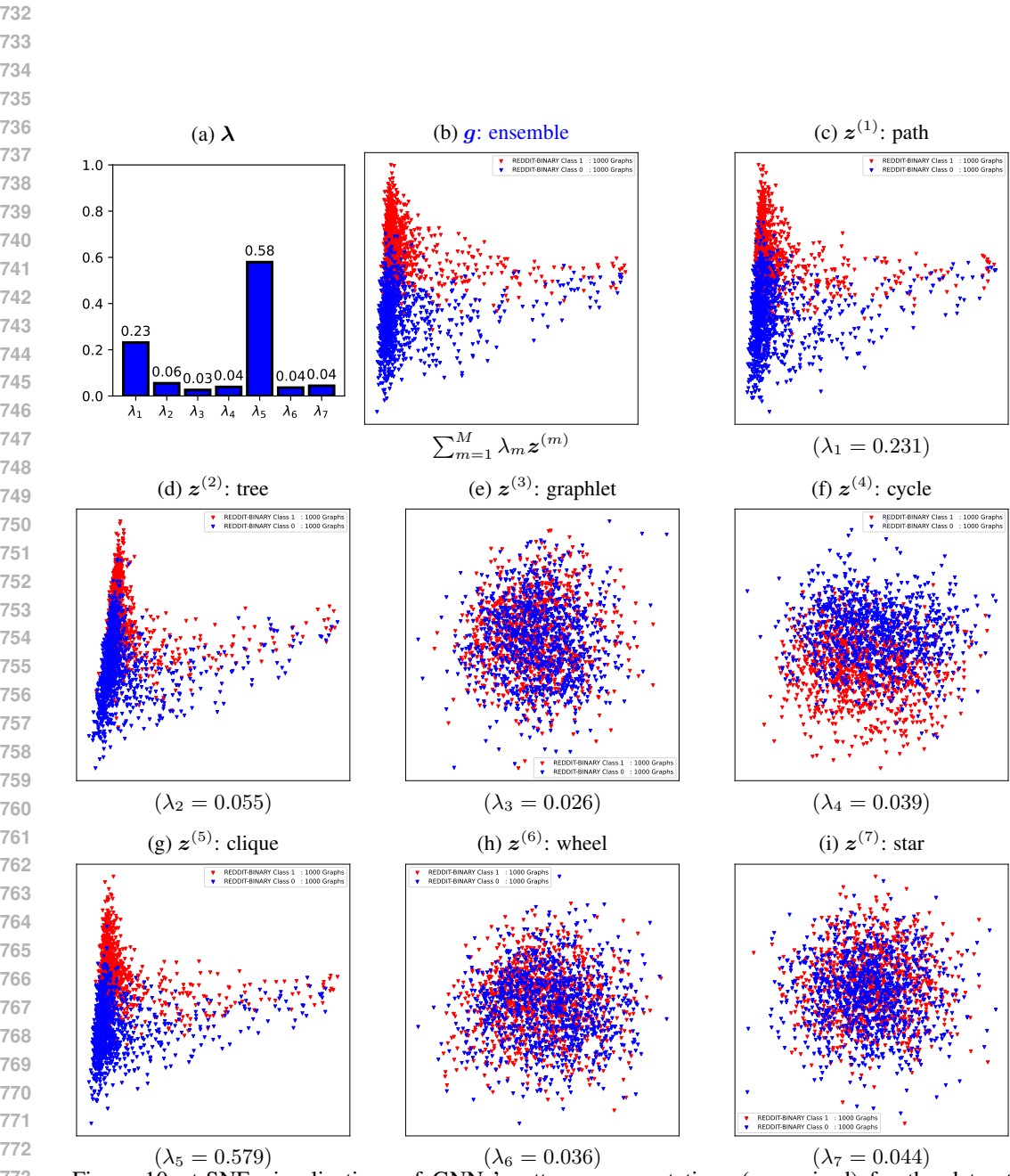

Figure 10: t-SNE visualizations of GNNs' pattern representations (supervised) for the dataset REDDIT-BINARY.

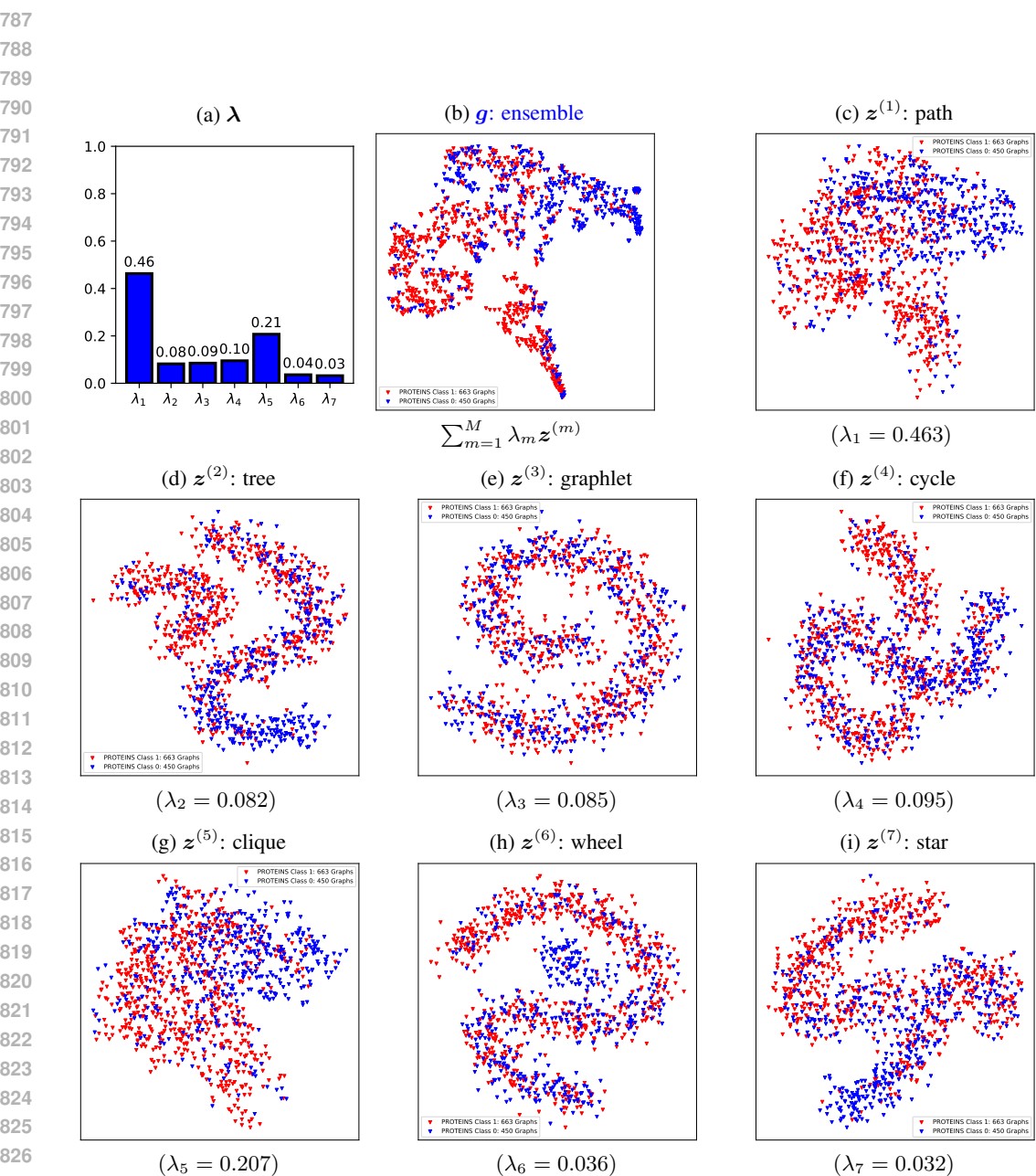

Figure 11: t-SNE visualizations of GNNs' pattern representations (unsupervised) for the dataset PROTEINS.

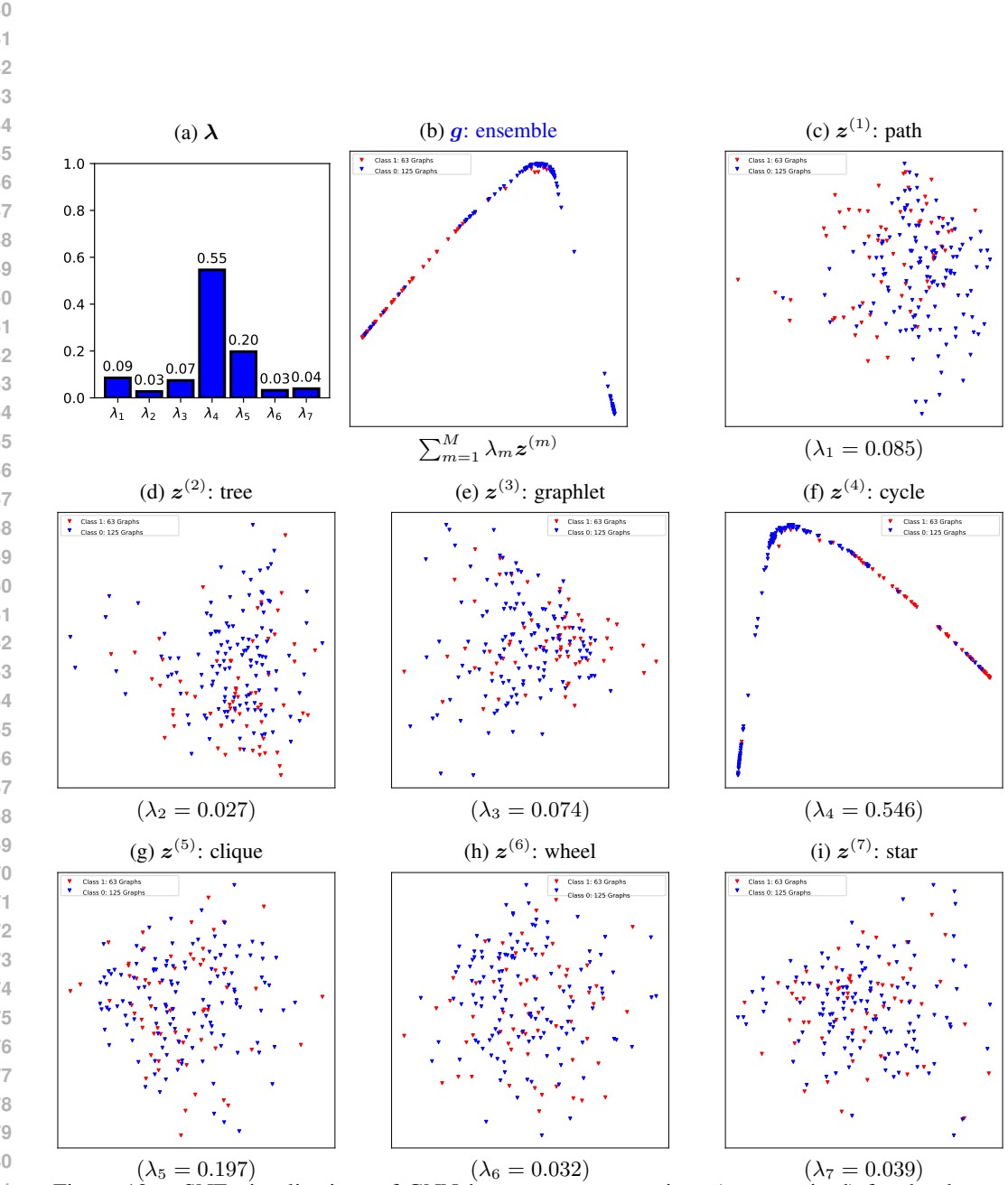

Figure 12: t-SNE visualizations of GNNs' pattern representations (unsupervised) for the dataset MUTAG.

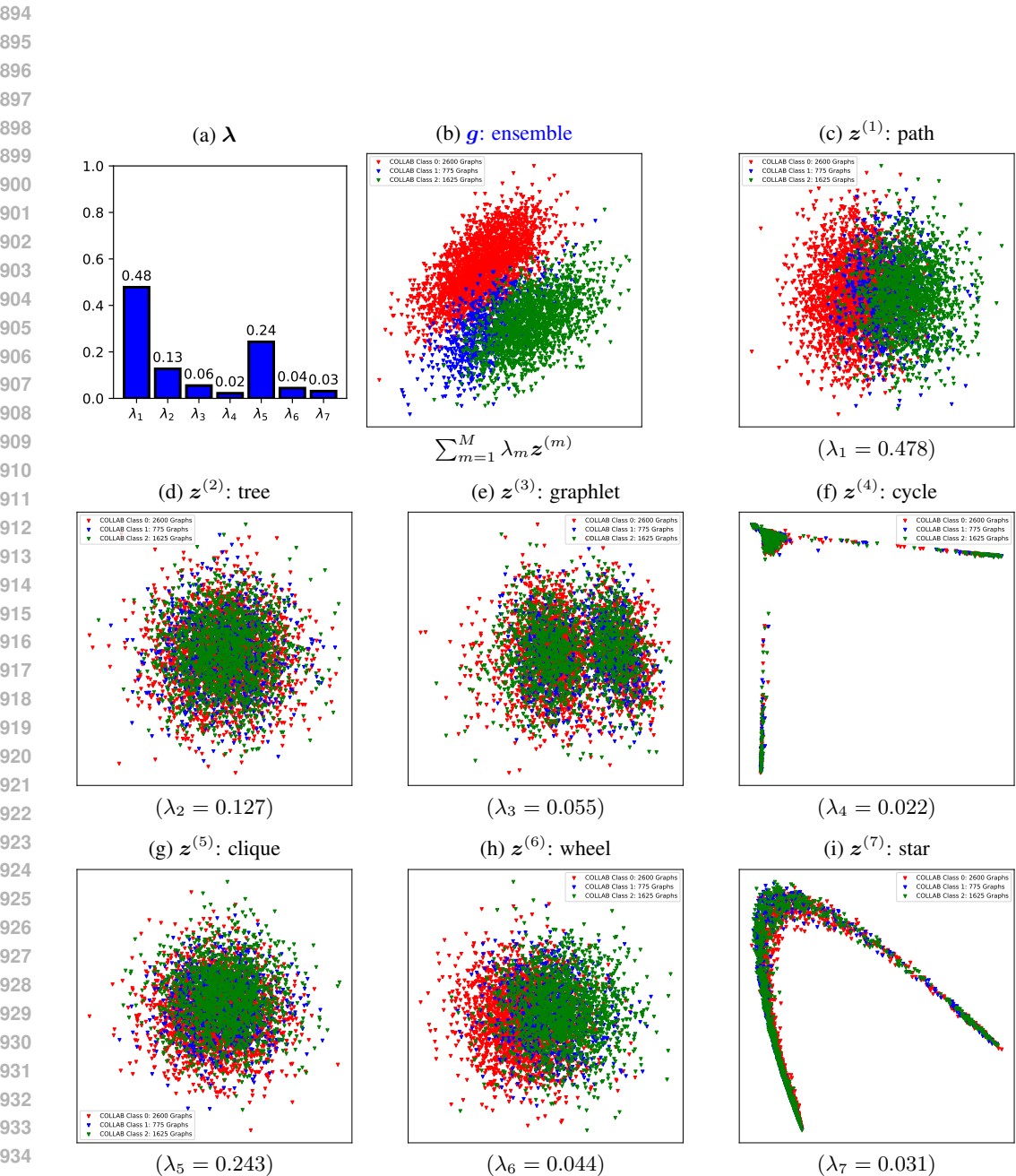

Figure 13: t-SNE visualizations of GNNs' pattern representations (unsupervised) for the dataset COLLAB.

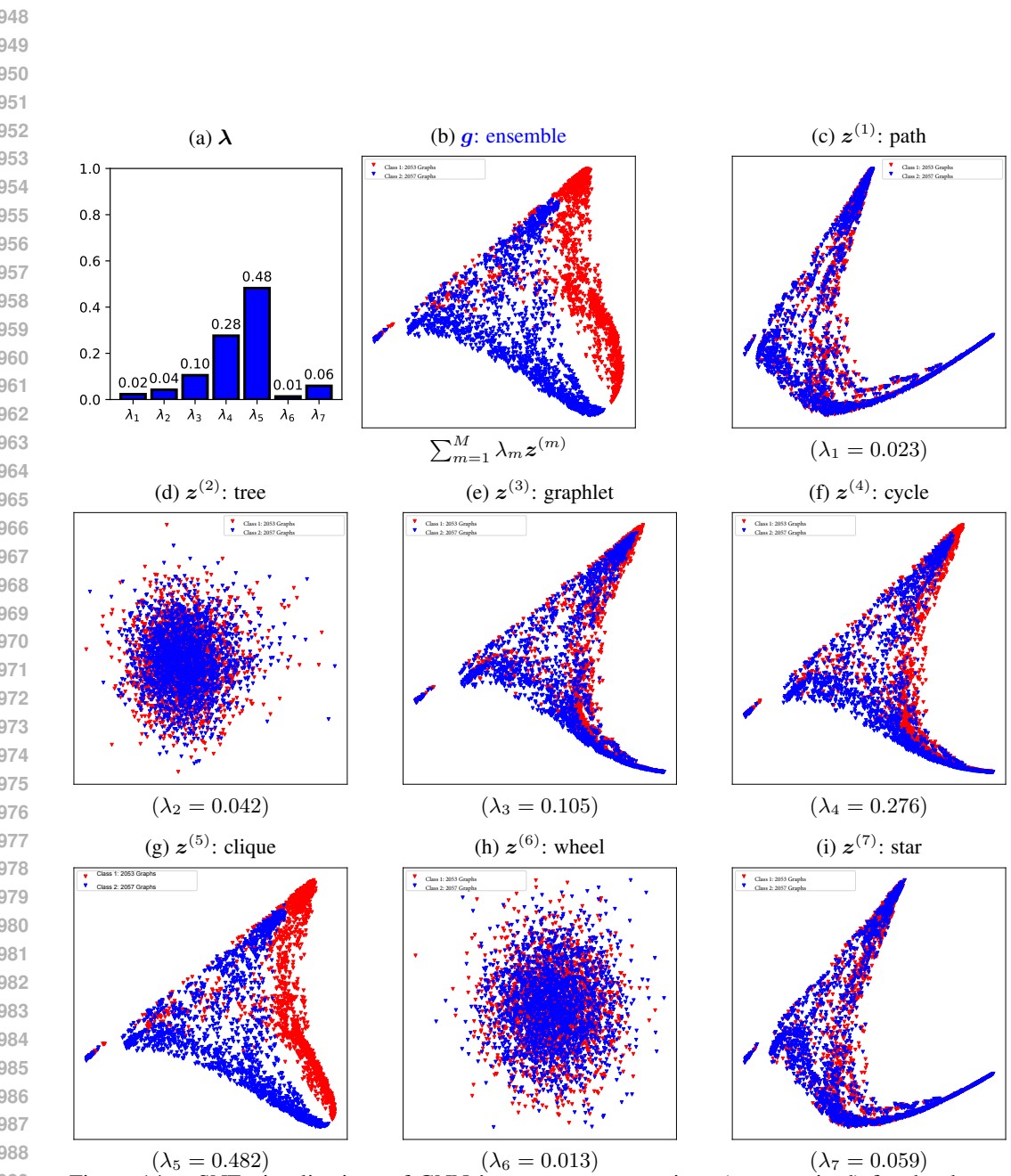

Figure 14: t-SNE visualizations of GNNs' pattern representations (unsupervised) for the dataset NCI1.

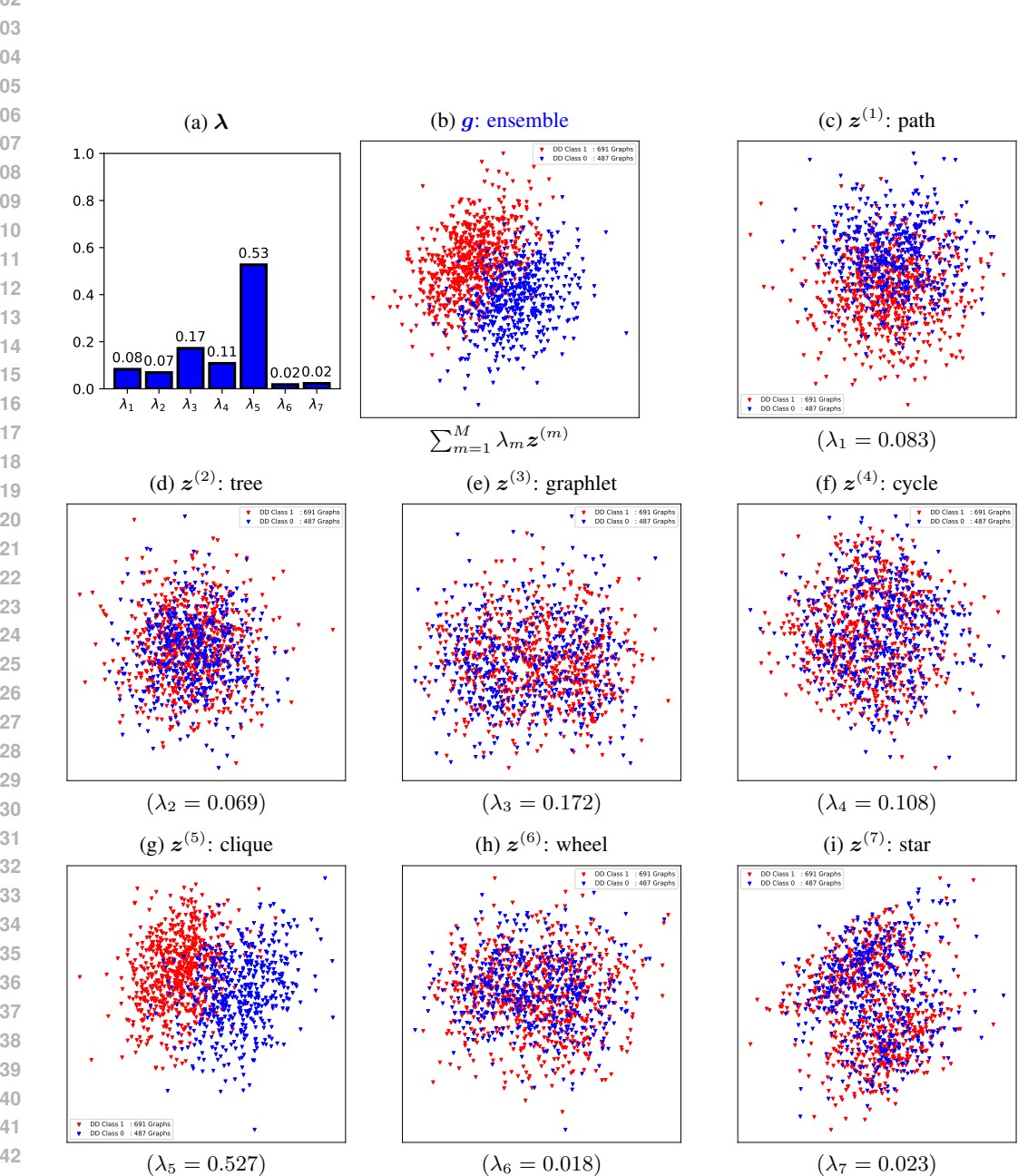

Figure 15: t-SNE visualizations of GNNs' pattern representations (unsupervised) for the dataset DD.

