# OpenReview forum: "Explainable Graph Representation Learning via Graph Pattern Analysis"
_ICLR.cc/2025/Conference — Submitted to ICLR 2025_

### Official Review · Reviewer_Un2e · 2024-10-31

**Soundness:** 3
**Presentation:** 2
**Contribution:** 3
**Rating:** 6
**Confidence:** 4

**Summary:**

This paper presents a novel method for explainable graph representation learning based on graph pattern analysis. It aims to answer what information about a graph is captured in its representations by analyzing substructures within different graph patterns. The method includes two approaches: a graph ensemble kernel (PXGL-EGK) for explainable similarity learning and a graph neural network (PXGL-GNN) that combines learned representations from various patterns. Theoretical analyses cover aspects like robustness and generalization, and experiments demonstrate the proposed method’s effectiveness in classification and clustering tasks.

**Strengths:**

Novelty in Explainable Representation: By focusing on pattern-based explainability, the paper addresses a unique gap in graph representation learning, offering insights into the contributions of specific patterns.

Theoretical Rigor: Theoretical analyses on robustness and generalization bounds add depth to the paper, strengthening the credibility of the proposed method.

Experimental Validation: The method outperforms several baselines in both supervised and unsupervised learning tasks, showing its practical effectiveness and explainability in graph-based tasks.

**Weaknesses:**

1. Language, Formatting, and Text Issues:

There are numerous language and formatting issues throughout the manuscript. For instance, the citation formats at the beginning of Chapters 1 and 4 are incorrect. Some terms that should be consistent, such as "generality" and "generalization," are not used uniformly. Additionally, some equations end with punctuation while others do not, and line 297 redundantly uses "Eq. equation." Furthermore, the equation on line 332 exceeds the page length significantly and would benefit from a line break. While I do not typically have strict formatting requirements, the frequency of these issues greatly impacts readability. I recommend a thorough review of the entire manuscript for consistency and clarity.

2. Structure of Chapters 4 and 5:

I suggest a complete rewrite and restructuring of Chapters 4 and 5. I found Chapters 1 to 3 to be relatively clear and easy to follow, but the latter chapters are quite challenging to understand. I read through them multiple times to grasp the progression between theoretical sections, yet some parts remain obscure. I recommend the following:
1.	Emphasize the relationship between Definition 4.2 and Definition 4.3 and the subsequent sections of the manuscript, explaining why these definitions are highlighted at the beginning.
2.	After each section of theoretical derivation, include a statement summarizing the conclusions that can be drawn from that section.

3. Limited Diversity in Pattern Selection:

The experiments focus on common graph patterns but lack exploration of more complex or specialized patterns, which may affect the method's generality and adaptability to various application domains.  More diverse and application-specific patterns could be explored to enhance the method’s flexibility and relevance cross various domains. Furthermore, future work could expand on computational efficiency, providing empirical results on time and memory usage for large datasets.


4. Scalability Concerns:

Although the method discusses time and space complexity, it lacks empirical results demonstrating efficiency on large-scale graph datasets, which raises questions about its scalability.

**Questions:**

Please see Weaknesses.

---

> ### Author Response · Authors · 2024-11-28
>
> Dear Reviewer Un2e,
>
> We highly appreciate your detailed comments that have improved the quality of our work significantly Our responses to the weaknesses you mentioned are as follows.
>
> **Response  to Weakness 1:**
>
> Thank you so much for pointing out these issues. We have revised the manuscript thoroughly to correct all these language and formatting errors and enhance readability. Your detailed suggestions improved the quality of our paper a lot.
>
> **Response  to Weakness 2:**
>
> Thank you for your nice suggestion. We followed them and have rewritten Chapters 4 and 5 thoroughly. The text with significant changes are highlighted in red.
>
> **Response  to Weakness 3:**
>
>  Currently, our pattern selection focuses on commonly used graph patterns in graph theory and we utilized **seven** patterns in the experiments. More importantly, as can be seen from Tables 3 and 5, our method already outperformed strong baselines of graph classification and clustering.
>
> However, there are domain-specific patterns (especially when the size of each graph is large), like certain functional groups in biochemistry or particular methods of community detection in social networks, that we have not yet explored. We plan to include these specialized patterns in our revised paper to broaden the method's applicability and enhance its relevance to various fields.
>
> **Response  to Weakness 4:**
>
> We have used the REDDIT-B dataset, which consists of 2000 graphs with an average of 429.63 nodes per graph, and the REDDIT-M5K dataset, which includes 4999 graphs with an average of 508.52 nodes per graph. These datasets are considered large-scale to some extent, especially when the computation source is limited. Currently, our equipment can only support experiments at this scale. (REDDIT-B: 7063s around 2h for one epoch running with 10 subgraphs sampled for each pattern; REDDIT-M5K, 25260s around 7h for one epoch running with 10 subgraphs sampled for each pattern).
>
> **It would be highly appreciated if you could check our paper again. We are continuously improving the writing and organization after the revision deadline. We are looking forward to your feedback.**

---

> > ### Comment · Reviewer_Un2e · 2024-12-02
> >
> > Thanks for the author's response. I have raised my score.

---

> > > ### Author Response · Authors · 2024-12-02
> > >
> > > It is our great pleasure to get your positive assessment. Thank you.

---

### Official Review · Reviewer_T9g2 · 2024-11-01

**Soundness:** 3
**Presentation:** 2
**Contribution:** 3
**Rating:** 5
**Confidence:** 4

**Summary:**

PXGL-GNN is an explainable graph representation learning method based on the analysis of graph patterns such as paths, trees, graphlets, cycles, cliques, wheels, and stars. It explicitly defines the important graph patterns of interest for analysis and learns the importance of each pattern with regard to supervised or unsupervised learning tasks.

**Strengths:**

The proposed method is novel in employing graph pattern analysis to introduce an explainable graph representation learning model. It reveals the importance of pre-selected graph patterns and their impact on the learned representation vector. Furthermore, this method is applicable in both supervised and unsupervised settings. While maintaining explainability, the model's accuracy in predictive tasks is comparably superior to other black-box models. Additionally, the authors provide a theoretical analysis of the proposed method in terms of robustness, generality, and complexity.

**Weaknesses:**

W1. The Introduction's storyline is confusing. The authors primarily discuss post-hoc explainers, including GNN-Explainer and XGNN, in the second paragraph and set the research goal as "What specific information about a graph is captured in graph representation learning?" On first reading, this easily misleads readers into thinking the proposed method is similar to post-hoc explanations. However, this is not the case of PXGL-GNN which is a graph representation learning that incorporates explainability.

W2. The authors claim the explainability of their GNN, but no evaluation of the explanations is conducted.

W3. The explanation method for graph representation learning is more valuable to discuss in the context of PXGL-GNN. However, the discussion of post-hoc graph explanations for supervised learning models is sufficient (in my opinion, it's excessive). Additionally, the authors omit the recent UNR-Explainer method, which explains node representation learning, in the related works section.

W4. Although PXGL-GNN is an explainable graph learning model, it lacks explicit explanations. While it provides the importance of pre-selected graph patterns, mapping these to specific subgraphs in the input graph is challenging. Users can understand the overall pattern for the output but struggle to map the insight into the real dataset. This becomes problematic when a certain graph pattern (e.g., a cycle) is important, but slight differences in its components lead to significant changes in prediction. Thus, providing only the graph pattern is insufficient to explain the prediction.

Minor: In Tables 3 and 5, the second-best results for the COLLAB dataset and REDDIT-B are not highlighted.

**Questions:**

Please refer to the weaknesses W1, W2, W3, and W4, as well as the minor issue mentioned.

Q1. What are the advantages of analyzing graph patterns for graph representation learning?

Q2. How does understanding important patterns in graph representation learning tasks benefit real-world scenarios?

Q3. Is it feasible to extend the proposed method to node representation learning?

Q4. Have you considered extracting explicit explanations, given that the model already possesses knowledge of important graph patterns in the input graph?


----------

 I appreciate the authors' dedicated efforts in addressing the reviewers’ concerns. As the review process has concluded, I will now present my feedback more directly for clarity.

Explaining representation $g$ is not exclusive to approaches of the model-level and instance-level explanations, for example, TAGE explains the representation vectors in view of the instance-level. Though authors point out the unique novelty compared to post-hoc explainers, in fact,  PXGL-GNN is an interpretable representation learning model or a self-explainable model that learns and explains the inherent important pattern in an integrated way. Since I appreciate the author's effort and novel work, I recommend improving the clarity of the introduction. I hope authors have a changed to consider my concerns.

---

> ### Author Response · Authors · 2024-11-28
> **Part I of Rebuttal**
>
> Dear Reviewer T9g2,
>
> We thank you for your comments that have improved the quality of our work a lot. Our responses to your comments are as follows.
>
> **Response  to Weakness 1:**
>
> We appreciate this suggestion regarding the organization of our introduction. While our paper introduces a novel concept of representation-level explainability in graph learning, we agree that the transition from post-hoc explainers could be better presented. In the revision, we reorganized the introduction to:
>
> i) First emphasize that unlike previous post-hoc and instance-level methods, our work focuses on a fundamental question: "What specific information about a graph is captured in graph representations?"
> ii) Then explain how pattern analysis provides a natural solution - comparing substructures within specific graph patterns allows us to understand what information is embedded in representations.
> iii) Finally present our technical approach to learning and explaining representations via pattern analysis.
>
> **Response  to Weakness 2:**
>
> Thank you for raising this point. Previous studies of explaining GNNs are usually supervised learning and use classification accuracy, AUROC, Fidelity, Probability of Sufficiency, etc, as scores for node features, edges, and motifs to evaluate their contribution, in comparison to the entire input [1]. Our work is different from these previous work in two aspects: 1) our methods work are mainly unsupervised, though we have supervised variants; 2) we aim at representation-level explanation.  However, we have provided evaluations through the following aspects:
> * In many figures such as Figure 2 and Figure 4, we use t-SNE to visualize the learned features of different graph patterns, which intuitively show the discrimination ability of different graph patterns.
> * For each dataset, as shown in Table 2 in our paper, we highlighted the two patterns with the largest contribution. The results are consistent with the properties of the graphs in each dataset. For instance, the chemical rings are important to determine the mutagenicity, while our method identified that circles make the largest contribution.
> * Direct evaluation in our method is easy to implement. For instance, the following table shows the classification accuracy of individual patterns and their different combinations on the MUTAG dataset. The best performance is shown in **bold**. It is worth mentioning that our method has the following significant advantage compared to other methods:
>    * Our method can not only identify the contribution of each individual pattern but also find the optimal combination of patterns, leading to improved classification accuracy.
>
> | Pattern Combinations | Accuracy (%) | λ weights |
> |---------------------|--------------|-----------|
> | Paths only | 80.47±1.24 | 1.0 |
> | Trees only | 86.39±2.73 | 1.0 |
> | Cycles only | 89.24±1.76 | 1.0 |
> | Paths + Trees | 87.11±2.93 | 0.274/0.716 |
> | Paths + Cycles | 91.62±1.14 | 0.207/0.793 |
> | Trees + Cycles | 92.31±2.65 | 0.325/0.675 |
> | **All Patterns** | **94.87±2.26** | 0.095/0.046/0.654 |
>
>
> [1] Kakkad et al. A Survey on Explainability of Graph Neural Networks. 2023. https://arxiv.org/pdf/2306.01958
>
> **Response  to Weakness 3:**
>
> We agree that the discussion of node representation explanations deserves more attention, although our approach is based on subgraph structures (patterns) rather than from a node perspective. In the revision, we will add a detailed discussion of UNR-Explainer and its related node representation explanation approach and compare our pattern-based method with UNR-Explainer in terms of methodology and applications. Further, we will reduce redundant discussions of post-hoc explanations while maintaining the necessary background.
>
> **Response  to Weakness 4:**
>
>  Thanks for the insightful comments. Our work aims at representation learning which is usually an **unsupervised learning** problem, though we provided a supervised variant of PXGL-GNN. This means our methods PXGL-EGK and PXGL-GNN are not designed to explain the prediction in supervised learning. To explain the prediction, one should prefer sample or feature attribution methods. Nevertheless, it is still possible to combine our pattern-based explanation with other explanation methods to explain prediction in supervised learning.
>
> **Response  to Minor:**
>
> Thanks for pointing this out. We added the highlights.

---

> ### Author Response · Authors · 2024-11-28
> **Part II of Rebuttal**
>
> **Response  to Question 1:**
>
> There are two advantages.
>
> * Graph representation learning is usually an unsupervised learning task and graph patterns have practical meanings or functions in analyzing graph data. Therefore, it is important to understand what graph patterns are learned by a graph representation learning model. Our model can explicitly show the importance of different graph patterns and hence provide clear evidence of what graph patterns are important and discriminative in a given unlabeled dataset.
>
> * In addition, explicitly taking advantage of different graph patterns can improve the accuracy in downstream tasks such as graph classification, which has been demonstrated by our experiments and our Theorem 5.2. Please refer to the discussion following Theorem 5.2: adding diverse patterns reduces the generalization error bound.
>
> **Response  to Question 2:**
>
> Understanding important patterns in graph representations benefits real-world scenarios in several key aspects. First, it provides domain experts with interpretable insights about which structural features are crucial for specific properties, such as functional groups in molecules or community structures in social networks. Second, it helps ML practitioners design more focused and efficient feature extraction strategies by identifying the most informative patterns. Additionally, this understanding enables better model validation and refinement by comparing the importance of the learned pattern with domain knowledge.
>
> **Response  to Question 3:**
>
> Unfortunately, it is not feasible because a graph pattern is based on multiple nodes rather than a single node and our method is specialized in graph-level representation learning.
>
> **Response  to Question 4:**
>
> We appreciate your suggestion. We guess you mean assigning a score (e.g. classification accuracy) to the pattern we considered in our method. This is easy to implement in supervised learning, as we have shown by the table in Part I of Rebuttal. Given the importance indicator $\lambda$, we can just evaluate the classification accuracy of each of the top $k$ patterns separately. However, for unsupervised representation learning, there is no ground truth label, therefore we have to use only the weight parameters $\lambda$ as scores for explanation.
>
> We are also considering to construct a mapping from $\lambda$ to the classification accuracy, which will improve the efficiency of explicit explanation.
>
>
> **Thank you again. We are looking forward to your feedback.**

---

> ### Comment · Reviewer_T9g2 · 2024-12-02
> **Response to the Authors**
>
> Thank you for the rebuttal. From the beginning, I appreciate the novelty of the method. I agree with the authors that the proposed method differs from post-hoc explanation methods in model-level or local-level explanations since PXGL-GNN cannot pinpoint or generate the important subgraph. That's why the second paragraph has been confusing for me. Unfortunately, my concern has not been solved yet in the revised version, so I decided to maintain my score.

---

> > ### Author Response · Authors · 2024-12-02
> >
> > Thanks for your feedback and recognition of the novelty of our method. We will further revise the paper to improve the clarity and eliminate confusion.

---

> ### Author Response · Authors · 2024-12-03
>
> Dear Reviewer T9g2,
>
> We have thoroughly revised the Introduction, where the text with significant changes is highlighted in red. The first two pages of the paper are at the anonymous link: https://anonymous.4open.science/api/repo/ICLR-Rebuttal-86F4/file/4579_Explainable_Graph_Represe.pdf?v=5d6afadc
> It would be highly appreciated if you could take into consideration the new revision.
>
> Sincerely,
>
> Authors

---

### Official Review · Reviewer_YwKS · 2024-11-03

**Soundness:** 3
**Presentation:** 3
**Contribution:** 2
**Rating:** 6
**Confidence:** 3

**Summary:**

This paper proposes explainable AI (XAI) in the graph domain from the perspective of graph representations. Existing XAI methods in the graph domain mainly focus on model-level or instance-level explanations and lack exploration of representation-level explanations. This paper introduces two methods: 1) PXGL-EGK and 2) PXGL-GNN, which combine representations of graph patterns within the input graph. These methods provide explanations through weight parameters that combine pattern representations to construct the overall graph representation. The authors propose both supervised and unsupervised versions for each method. PXGL-GNN addresses several limitations of PXGL-EGK, such as ignoring node features, high dimensionality, time complexity, and limited expressiveness. The paper includes theoretical analysis on robustness, generality, and complexity, with experimental results and t-SNE visualizations demonstrating the superiority of the proposed methods over existing approaches.

**Strengths:**

1. The paper is well-written and easy to follow.

2. It explores representation-level explanations within the graph domain, an area that is not well-explored.

3. The explanation is intuitive and effectively highlights dominant graph patterns using weight parameters.

**Weaknesses:**

1. The theoretical analysis suggests that the robustness of the method depends on the number of layers \( L \). A performance comparison across different values of \( L \) would be beneficial, as setting \( L = 5 \) appears to be heuristic.

2. Since the representation of the input graph is directly influenced by the types of patterns used, the authors should demonstrate how different combinations of patterns affect the representation.

3. A hyperparameter analysis regarding the number of samples \( Q \) is necessary.

**Questions:**

1. Regarding datasets that contain node attributes, as the authors argue that PXGL-GNN can address the issue of ignoring node features present in PXGL-EGK, it would be beneficial to directly compare the two methods on a dataset with node attributes.

2. In the MUTAG dataset, each graph's label is determined by the presence of key tree-like substructures, such as \( NO_2 \) or \( NH_4 \). In other words, ring patterns do not influence the prediction of the input data. However, PXGL-GNN tends to capture dominant patterns within the input data, often assigning larger weights to cyclic structures. Given this tendency, how can PXGL-GNN outperform other baselines that focus on the critical substructures (i.e., \( NO_2 \) or \( NH_4 \)) needed to predict the label of the input graph?

---

> ### Author Response · Authors · 2024-11-28
> **Part I of Rebuttal**
>
> Dear Reviewer YwKS,
>
> We sincerely appreciate your comments that have improved the quality of our work a lot. Our responses to your comments are as follows.
>
> **Response  to Weakness 1:**
>
> Thanks for pointing out this. In the following table, we use three datasets to show the
> accuracy of Graph Classification of our PXGL-EGK model with different number of layers (L).
>
> | Model | L=1 | L=3 | L=5 | L=7 | L=9 |
> |-------|-----|-----|-----|-----|-----|
> | MUTAG | 81.44±1.29 | 86.73±2.78 | 94.87±2.26 | 91.25±1.14 | 89.66±1.15 |
> | PROTEINS | 62.17±1.53 | 67.22±1.16 | 78.23±2.46 | 73.21±1.98 | 71.07±1.63 |
> | DD | 75.36±1.21 | 79.35±1.20 | 86.54±1.95 | 82.36±1.24 | 82.17±1.54 |
>
> The results reveal that the model performs best at $L = 5$. With fewer layers, the model lacks sufficient capacity for representation; with more layers, the model is too complex and has overfitting performances. This is consistent with our theoretical analysis, since when the model is complex the gap between training error and the testing error becomes large.
>
> **Response  to Weakness 2:**
>
> Yes. The following table shows the classification accuracy given by PXGL-GNN with different combinations of patterns on the MUTAG dataset. We see that by including more patterns, the classification accuracy tends to be higher. This is consistent with our theoretical analysis of Theorem 5.2 (please refer to the text around equation (15)). It is worth noting that when numerous patterns are used, our method can automatically identify the importance of each pattern and provide an optimal combination of the patterns.
>
> Performance comparison of different pattern combinations on MUTAG dataset. The best performance is shown in **bold**.
>
> | Pattern Combinations | Accuracy (%) | λ weights |
> |---------------------|--------------|-----------|
> | Paths only | 80.47±1.24 | 1.0 |
> | Trees only | 86.39±2.73 | 1.0 |
> | Cycles only | 89.24±1.76 | 1.0 |
> | Paths + Trees | 87.11±2.93 | 0.274/0.716 |
> | Paths + Cycles | 91.62±1.14 | 0.207/0.793 |
> | Trees + Cycles | 92.31±2.65 | 0.325/0.675 |
> | **All Patterns** | **94.87±2.26** | 0.095/0.046/0.654 |
>
> **Response  to Weakness 3:**
>
> Thanks for pointing this out. The following table provides some results about the impact of $Q$ on the classification accuracy and time cost on the MUTAG dataset. We see that the time cost is roughly linear with $Q$ and the accuracy is not sensitive to $Q$ when it is larger than $5$.
>
> Table: Impact of sampling number Q on MUTAG dataset (20 epochs, 7 patterns)
>
> | Q | 3 | 5 | 7 | 10 | 15 |
> |---|---|---|---|----|----|
> | Accuracy (%) | 87.63±1.42 | 94.87±2.26 | 94.26±2.17 | 95.35±1.89 | 95.33±2.48 |
> | Training Time (s) | 636 | 877 | 1035 | 1563 | 2351 |

---

> > ### Author Response · Authors · 2024-11-28
> > **Part II of Rebuttal**
> >
> > **Response  to Question 1:**
> >
> >  The following table shows the comparison between PXGL-GNN and PXGL-EGK on the dataset PROTEINS with node attributes. We see that the difference is not significant. One possible reason is that the attributes are not discriminative compared to the adjacency matrices.
> >
> > Comparison between PXGL-EGK and PXGL-GNN on PROTEINS dataset with node attributes
> >
> > | Method | Accuracy (%) | NMI |
> > |--------|-------------|-----|
> > | PXGL-EGK | 0.721±0.028 | 0.321±0.019 |
> > | PXGL-GNN | 0.746±0.019 | 0.292±0.010 |
> >
> > **Response  to Question 2:**
> >
> > Thanks for your insightful question. For the MUTAG dataset, carbon rings with both NH2 or NO2 chemical groups are valid explanations for the GNN model to recognize a given molecule as mutagenic [1].  These functional groups are typically attached to aromatic or heteroaromatic rings, which are critical for determining chemical reactivity and mutagenicity. Our PXGL-GNN leverages one-hot encoded atom types (7 discrete labels, e.g., C, N, O) as node attributes (via widely used dgl.GINDataset data loader API),  enabling the model to distinguish atom types within cyclic structures via inherently considering the chemical context in which NO2 or NH2 groups exist. By capturing and aggregating carbon rings enriched with atom-type information, PXGL-GNN considers the chemical context of functional groups, enhancing its ability to classify mutagenic molecules.
> >
> >
> > In the following table, we show the performance of PXGL-GNN with tree pattern, cycle, and their combination on the MUTAG dataset. The MUTAG dataset suggests that the presence of carbon rings that have either NH2 or NO2 chemical groups can explain why a GNN model identifies a molecule as mutagenic, according to one study [1]. The original journal article emphasizes that both cycles and trees are important, and combining them leads to better results.
> >
> > Pattern-specific analysis on MUTAG dataset
> >
> > | Pattern Focus | Accuracy (%) | Important Substructures |
> > |--------------|--------------|------------------------|
> > | Trees only | 86.39±2.73 | NO2, NH2 groups |
> > | Cycles only | 89.24±1.76 | Ring structures |
> > | Combined | 92.31±2.65 | Both |
> >
> > [1] Agarwal C, Queen O, Lakkaraju H, et al. Evaluating explainability for graph neural networks[J]. Scientific Data, 2023, 10(1): 144.
> >
> > **We hope that these explanations and results can address your concerns. We are eager to receive your further feedback.**

---

> ### Comment · Reviewer_YwKS · 2024-12-02
>
> Thank you for your rebuttal. I think the authors have addressed the concerns I raised well. I am inclined to raise my score and hope the revised version will be included if the paper is accepted.

---

> > ### Author Response · Authors · 2024-12-03
> >
> > We sincerely thank you for your support. We will definitely include the revision and any further enhancements in the paper.

---

### Official Review · Reviewer_3Cyd · 2024-11-04

**Soundness:** 2
**Presentation:** 3
**Contribution:** 3
**Rating:** 6
**Confidence:** 4

**Summary:**

This paper proposes a graph representation learning method leveraging graph pattern analysis. It presents a representation learning approach applicable to both supervised and unsupervised learning, utilizing various graph kernels to provide interpretable pattern information for post hoc analysis, effectively demonstrating its utility in experiments.

**Strengths:**

- The notation is clearly defined, contributing to a well-articulated description of the proposed approach.
- The use of visualizations enhances the accessibility of pattern-based explanations.
- The paper effectively conveys the need to provide explanations based on graph patterns.

**Weaknesses:**

- The time complexity would be significantly higher than the time complexity analysis. However, it is difficult to say it is incorrect because details about kernel usage, such as the number of graphlets employed, and preprocessing requirements are missing. In particular, efficiently identifying graphlets with more than four nodes is a challenging task.
- While the theoretical analysis is included, it lacks the intuition and analysis for understanding the proposed work.
 - The introduction mentions post-hoc interpretability methods, but the connection to the proposed approach is unclear, as they appear to address entirely different tasks.
 - As the paper addresses explanations at the representation level, comparisons with methods like UNR-Explainer [1] and MotifExplainer [2] would strengthen the related work or introduction section.
 - The paper employs graph sampling, yet details and sensitivity analysis are omitted.
 - GNNs typically struggle with modeling paths, trees, and graphlets due to limited expressive power. It remains unclear if this aspect is effectively visualized and well-learned in the training analysis.



References:

[1] Kang, Hyunju, Geonhee Han, and Hogun Park. "UNR-Explainer: Counterfactual Explanations for Unsupervised Node Representation Learning Models." The Twelfth International Conference on Learning Representations, 2024.

[2] Yu, Zhaoning, and Hongyang Gao. "MotifExplainer: a motif-based graph neural network explainer." arXiv preprint arXiv:2202.00519, 2022.

**Questions:**

Please refer to the weaknesses above.

---

> ### Author Response · Authors · 2024-11-28
> **Part I of Rebuttal**
>
> Dear Reviewer 3Cyd,
>
> We appreciate your comments that have improved the quality of our work a lot. Our responses to weaknesses you mention as follows.
>
> **Response  to Weakness 1**
>
> This is an insightful comment. We didn't show the time complexities of kernel computation and subgraph (e.g. graphlet) sampling. In the updated manuscript, we modified the time complexities by adding the complexities of kernel computation and subgraph sampling. For convenience, we show some results as follows:
>
> Given a dataset with $N$ graphs (each has $n$ nodes and $e$ edges), we select $M$ different patterns and sample $Q$ subgraphs of each pattern. The time complexity of PXGL-EGK is $\mathcal{O}(N^2\sum_{m=1}^M\psi_i)$, where $\psi_i$ denotes the time complexity of the $m$-th graph kernel. For instance, the time complexities of the graphlet kernel, shortest path kernel, and Weisfeiler-Lehman Subtree kernel are $\mathcal{O}(n^k)$, $\mathcal{O}(n^4)$, and $\mathcal{O}(hn+he)$ respectively, where $k$ and $h$ are some kernel-specific hyperparameters.
> Regarding PXGL-GNN, suppose each representation learning function $F_m$ is an $L$-layer GCN, of which the width is linear with $d$. For both supervised and unsupervised learning, suppose the batch size and the number of iterations in the optimization are $B$ and $T$ respectively. Then, in supervised learning, the time complexity is $\mathcal{O}(TBMQL(ed+nd^2)+NQ\sum_{m=1}^M\vartheta_m)$, where $\vartheta_m$ denotes the time complexity of generating a sample of the $m$-th pattern. For instance, when the $m$-th pattern is graphlets with size $k\in\\{3,4,5\\}$, we have $\vartheta_m\leq nu^{k-1}$ [1], where $u$ denotes the maximum node degree of the graph. In unsupervised learning, the time complexity is $\mathcal{O}(TBMQL(ed+nd^2)+TB^2+NQ\sum_{m=1}^M\vartheta_m)$.
>
> Here we show the time costs of the sampling stage and training stage on the MUTAG dataset.
> | # Samples | # Patterns | Sampling Time (s) | Training Time (s) for one epoch |
> |-----------|------------|------------------|--------------------------------|
> | 10        | 7          | 95.71           | 73.00                         |
> | 15        | 7          | 189.12          | 108.00                        |
>
>
> [1] Shervashidze et al. Efficient graphlet kernels for large graph comparison. AISTATS 2009.
>
> **Response to Weakness 2:**
>
> Thanks for pointing it out. Our theoretical analysis focuses on demonstrating the robustness and generalization capabilities of our model PXGL-EGK, since robustness and generalization are two important aspects of any machine learning model.
>
> Our robust analysis shows that the representations produced by PXGL-EGK remain stable even when the input graphs are altered by noise under some mild conditions, such as the spectral norms of the weight matrices and the adjacency matrix are not too large. Specifically, PXGL-EGK is notably resilient to changes in the graph structure (like perturbations to node features) and increasing the minimum node degree tends to enhance its robustness.
>
> Additionally, our generalization analysis shows the upper bound of the difference between training error and testing error.  It is worth noting that in the proof (see (36)) of the theorem, we used an aggressive relaxation such that $\boldsymbol{\lambda}$ was not present in $\eta$. By keeping $\boldsymbol{\lambda}$, we can obtain
>
> \begin{equation}
> \eta = \frac{\tau}{\sqrt{n}} \rho^L \hat{\beta}_ W^{L-1} \beta_X (1 + \beta_ A)^L (1 + \alpha)^{-L} \left[ \hat{\beta}_ W \gamma_ {\Delta C} + \gamma_{C} \left(\hat{\beta}_ W\Vert\boldsymbol{\lambda}_ {\mathcal{D}}-\boldsymbol{\lambda}_ {\mathcal{D}^{\backslash i}}\Vert+ L \hat{\beta}_ {\Delta W}\Vert\boldsymbol{\lambda}_ {\mathcal{D}^{\backslash i}}\Vert\right) \right]
> \end{equation}
>
> Since $\Vert\boldsymbol{\lambda}_ {\mathcal{D}}\Vert_ 1=\Vert\boldsymbol{\lambda}_ {\mathcal{D}^{\backslash i}}\Vert_ 1=1$, when $M$ is larger, $\Vert\boldsymbol{\lambda}_ {\mathcal{D}}-\boldsymbol{\lambda}_ {\mathcal{D}^{\backslash i}}\Vert$ and $\lVert\boldsymbol{\lambda}_ {\mathcal{D}^{\backslash i}}\Vert$ are potentially smaller. This means that when we include more graph patterns, the generalization bound (linear with $\eta$ shown in Section 5.2) of our PXGL-GNN becomes tighter, which potentially leads to higher classification accuracy. This is actually supported by our experiments. For instance, the following table shows the influence of different combinations of patterns on the MUTAG dataset. We can see that more patterns do better.
>
> | Pattern Combinations | Accuracy (%) | λ weights |
> |---------------------|--------------|-----------|
> | Paths only | 80.47±1.24 | 1.0 |
> | Trees only | 86.39±2.73 | 1.0 |
> | Cycles only | 89.24±1.76 | 1.0 |
> | Paths + Trees | 87.11±2.93 | 0.274/0.716 |
> | Paths + Cycles | 91.62±1.14 | 0.207/0.793 |
> | Trees + Cycles | 92.31±2.65 | 0.325/0.675 |
> | **All Patterns** | **94.87±2.26** | 0.095/0.046/0.654 |
>
>
> We have added the above discussion to the updated paper.

---

> ### Author Response · Authors · 2024-11-28
> **Part II of Rebuttal**
>
> **Response to Weakness 3:**
>
> Yes, they are entirely different. Our PXGL-EGK introduces a novel concept in explainable graph learning at the representation level, which is a first in this field. This differs significantly from previous approaches to explainable graph learning. Our PXGL-EGK method pioneers representation-level explainability in graph learning. It seeks to answer a fundamental question: What specific information about a graph is captured within its representations.
>
> Existing model-level explainability methods in graph learning, such as GLG-Explainer [1] and GCFExplainer [2], aim to make the model's overall behavior more transparent. Instance-level methods, like GNNExplainer [3], provide explanations for specific predictions by detailing the reasoning behind particular classifications.
>
> [1] Steve Azzolin, et al. Global explainability of gnns via logic combination of learned concepts.
>
> [2] Zexi Huang, et al. Global counterfactual explainer for graph neural networks. ICDM, 2023.
>
> [3] Zhitao Ying et al. Gnnexplainer: Generating explanations for graph neural networks. NIPS 2019.
>
>
> **Response to Weakness 4:**
>
> Thanks for providing the related references. We included them in our updated manuscript and added the following discussion:
>
> * UNR-Explainer [1] identifies the top-k most important nodes in a graph to determine the most significant subgraph. It is a classic instance-level explainable graph learning method focused on node representation. However, this task is entirely different from our approach, as it addresses node-level representation rather than representation-level explainability. For this reason, we did not include a comparison.
>
> * MotifExplainer [2] identifies critical motifs (small subgraphs) in a graph. However, it does not address graph representation learning or representation-level explainability. Instead, it focuses on understanding what specific information about a graph is captured in its representations. Since this is fundamentally different from our work, we did not compare our method to theirs.
>
> [1] Kang, Hyunju, Geonhee Han, and Hogun Park. "UNR-Explainer: Counterfactual Explanations for Unsupervised Node Representation Learning Models." The Twelfth International Conference on Learning Representations, 2024.
>
> [2] Yu, Zhaoning, and Hongyang Gao. "MotifExplainer: a motif-based graph neural network explainer." arXiv preprint arXiv:2202.00519, 2022.
>
>
> **Response to Weakness 5:**
>
> Subgraph counting graph kernels involves sampling specific subgraph patterns and counting their occurrences. For example, random walk kernels [1] sample random walks, subtree kernels [2] sample trees, and graphlet kernels [3] sample graphlets. In our work, we directly use the subgraph sampling functions provided by these traditional graph kernels in the GraKel tool https://ysig.github.io/GraKeL/0.1a8/.
>
> Here we use the MUTAG dataset to show the sensitivity of accuracy and time cost to the number of samples $Q$ for each pattern. We see that the time cost is roughly linear with $Q$ and the accuracy is not sensitive to $Q$ when it is larger than $5$.
>
> **Table:** Impact of sampling number Q on MUTAG dataset (20 epochs, 7 patterns)
>
> | Q | 3 | 5 | 7 | 10 | 15 |
> |---|---|---|---|----|----|
> | Accuracy (%) | 87.63±1.42 | 94.87±2.26 | 94.26±2.17 | 95.35±1.89 | 95.33±2.48 |
> | Training Time (s) | 636 | 877 | 1035 | 1563 | 2351 |
>
>
>
> [1] Borgwardt et al. Protein function prediction via graph kernels. Bioinformatics, 2005.
>
> [2] Giovanni et al. A tree-based kernel for graphs. SIAM, 2012.
>
> [3] Shervashidze et al. Efficient graphlet kernels for large graph comparison. AI and statistics, 2009
>
>
>
> **Response to Weakness 6:**
>
>  You're right that GNNs face limitations in expressive power. This issue has been extensively studied in [1], which shows that GNNs can capture graph patterns when subgraphs are small. A more powerful alternative is to use graph transformers, which have significantly greater expressive capacity.
> As an extension of our work, we plan to incorporate graph transformers into our PXGL-EGK method to address these limitations.
>
>
> [1]B Zhang et al. Beyond weisfeiler-lehman: A quantitative framework for GNN expressiveness
>
>
> **Thank you again for your comments. We are looking forward to your feedback.**

---

> > ### Author Response · Authors · 2024-12-03
> >
> > Dear Reviewer 3Cyd,
> >
> > As the author-reviewer discussion is about to end, we'd like to know whether our responses addressed your concerns. We are eager to provide more information if needed in these last few hours.
> >
> > Sincerely,
> >
> > Authors

---

### Author Response · Authors · 2024-12-01

Dear Reviewers 3Cyd, YwKS, T9g2, and Un2e,

We thank you again for reviewing our work. Since the author-reviewer discussion period is going to end, we are eager to know whether our responses and revision have addressed your concerns or not and are looking forward to your feedback and further discussion.

Yours sincerely,

Authors of the paper

---

### Meta-Review · Area_Chair_iA58 · 2024-12-20

**Metareview:**

The paper introduces a pattern-level explanation method to analyze the contribution of different patterns to the representations learned by graph neural networks. While the reviewers recognized the merits of the approach, they also highlighted several areas for improvement. These include ambiguous terminology, insufficient coverage and comparison to related works, and unclear positioning of the paper's contributions. During the post-rebuttal discussion, these concerns remained unresolved, leading reviewers to hesitate in advocating for its acceptance.

**Additional Comments On Reviewer Discussion:**

The paper stands on the borderline. Hence, in the post-rebuttal phase, the reviewers were asked to put forward their final thoughts and whether they would champion acceptance or rejection. One reviewer concluded that the concerns remained unresolved despite the rebuttal, while the others refrained from championing acceptance.

---

### Decision · Program_Chairs · 2025-01-22

Reject